# BERNOULLI-LoRA: A THEORETICAL FRAMEWORK FOR RANDOMIZED LOW-RANK ADAPTATION

## ABSTRACT

Parameter-efficient fine-tuning (PEFT) has emerged as a crucial approach for adapting large foundational models to specific tasks, particularly as model sizes continue to grow exponentially. Among PEFT methods, Low-Rank Adaptation (LoRA) (Hu et al., 2022) stands out for its effectiveness and simplicity, expressing adaptations as a product of two low-rank matrices. While extensive empirical studies demonstrate LoRA's practical utility, theoretical understanding of such methods remains limited. Recent work on RAC-LoRA (Malinovsky et al., 2024) took initial steps toward rigorous analysis. In this work, we introduce Bernoulli-LoRA, a novel theoretical framework that unifies and extends existing LoRA approaches. Our method introduces a probabilistic Bernoulli mechanism for selecting which matrix to update. This approach encompasses and generalizes various existing update strategies while maintaining theoretical tractability. Under standard assumptions from non-convex optimization literature, we analyze several variants of our framework: Bernoulli-LoRA-GD, Bernoulli-LoRA-SGD, Bernoulli-LoRA-PAGE, and Bernoulli-LoRA-MVR, Bernoulli-LoRA-QGD, Bernoulli-LoRA-MARINA, Bernoulli-LoRA-EF21, establishing convergence guarantees for each variant. Additionally, we extend our analysis to convex non-smooth functions, providing convergence rates for both constant and adaptive (Polyak-type) stepsizes. Through extensive experiments on various tasks, we validate our theoretical findings and demonstrate the practical efficacy of our approach. This work is a step toward developing theoretically grounded yet practically effective PEFT methods.

## 1 INTRODUCTION

Fine-tuning adapts pre-trained models to new datasets, a central task in modern deep learning, particularly NLP (Peters et al., 2018; Devlin et al., 2019). However, full fine-tuning is computationally expensive for large models. Parameter-Efficient Fine-Tuning (PEFT) (He et al., 2021) addresses this by updating only a fraction of parameters (Richtárik & Takáč, 2016; Demidovich et al., 2023a), matching full fine-tuning performance with significantly lower costs (Radford et al., 2019; Brown et al., 2020; Han et al., 2024).

Leveraging the low intrinsic dimensionality of pre-trained models (Li et al., 2018; Aghajanyan et al., 2020), Low-Rank Adaptation (LoRA) (Hu et al., 2022) optimizes updates in a reduced subspace. It replaces large matrix updates with the product of two trainable low-rank matrices:

$$W = W^0 + \frac{\alpha}{r} BA,$$

where $W^0 \in \mathbb{R}^{m \times n}$ is fixed, and $B \in \mathbb{R}^{m \times r}, A \in \mathbb{R}^{r \times n}$ are trainable ($r \ll \min\{m, n\}$). While typically initialized with Gaussian $A$ and zero $B$, other strategies exist (Zhu et al., 2024; Hayou et al., 2024; Meng et al., 2024; Wang et al., 2025). Beyond improving efficiency (Cherniuk et al., 2023; Mao et al., 2025), LoRA mitigates catastrophic forgetting and enhances output diversity (Biderman et al., 2024).

To approach full fine-tuning performance, Xia et al. (2024) introduced Chain of LoRA (COLA). This framework iteratively builds higher-rank updates from a sequence of low-rank modules at no extra computational cost. By merging updates into fixed parameters, it yields:

$$W = W^0 + \frac{\alpha}{r} \sum_{t=0}^{T-1} B^t A^t.$$

Unlike standard LoRA, COLA uses sequential decompositions to efficiently approximate high-rank adaptations.

Recent theoretical works analyze LoRA from complementary angles. Jang et al. (2024) prove that sufficiently high-rank LoRA eliminates spurious local minima in the NTK regime. Kim et al. (2025) show that training typically converges to a low-rank global minimum or diverges toward high-rank solutions. In continuous-time settings, Xu et al. (2025) highlight the pivotal role of initialization in matrix factorization gradient flows, while Dayi & Chen (2025) position low-rank fine-tuning between lazy training and feature learning.

## 2 MOTIVATION

Theoretical advances above highlight what happens in specific regimes, but they leave open whether practical, discrete-time LoRA updates converge under realistic training noise and communication constraints. This gap motivates our framework: we seek general convergence guarantees for randomized low-rank adaptation with stochastic gradients, variance reduction, and federated communication savings. At the same time, despite their practical success, Low-Rank Adaptation (LoRA) and its variants like Chain of LoRA (COLA) still lack a unified and practically relevant convergence theory. LoRA's re-parameterization makes smooth loss functions non-smooth, creating significant theoretical hurdles (Sun et al., 2024). Second, existing COLA analysis ignores its core low-rank updates by focusing on full-rank optimization, thus failing to explain its efficiency (Xia et al., 2024). Consequently, most LoRA-based methods are heuristics without convergence guarantees, making them sensitive to hyperparameters (Khodak et al., 2021; Kuang et al., 2024). Malinovsky et al. (2024) even showed COLA can diverge and introduced RAC-LoRA, the first framework to establish convergence rates for LoRA-style updates. However, the RAC-LoRA framework is limited. It lacks optimal variance-reduced techniques for non-convex problems and fails to address advanced Federated Learning (FL) scenarios incorporating communication compression and error feedback (Alistarh et al., 2018; Wen et al., 2017; Horváth et al., 2022; Panferov et al., 2024). The need for distributed optimization like FL is driven by the challenge of training massive models (Brown et al., 2020; Kolesnikov et al., 2020; Goyal et al., 2017; You et al., 2019; Le Scao et al., 2023). Our work aims to bridge this gap by extending a theoretically sound LoRA framework to these vital, practical optimization settings. In the next section, we formalize the optimization problems we study.

## 3 PROBLEM STATEMENT

Supervised learning is an optimization problem that minimizes a loss function. We focus on this challenge in fine-tuning, using a general, model-agnostic formulation:

$$\min_{\Delta W \in \mathbb{R}^{m \times n}} f(W^0 + \Delta W). \tag{1}$$

Here, $W^0$ represents the pre-trained parameters, $\Delta W$ is the trainable adaptation, and $f$ is the empirical loss. Since $m \times n$ is very large, $\Delta W$ requires a simple, trainable structure.

Throughout the paper, we treat $W^0$ as a fixed pre-trained model and view $f$ as the fine-tuning loss that already encodes the effect of the pre-training and fine-tuning data distributions (including any mismatch between them). All of our convergence guarantees are therefore conditional on this given $W^0$ and the associated fine-tuning objective $f$. We do not model the representation-learning dynamics of the pre-training phase, nor do we analyze generalization; our focus is purely on the optimization behavior of low-rank LoRA-style updates when minimizing the fine-tuning loss.

For our stochastic methods, we consider these objective structures:

- **Finite-Sum Setting:** The objective is an average over $N$ data samples, used in methods like Bernoulli-LoRA-PAGE:

$$f(W^0 + \Delta W) = \frac{1}{N} \sum_{i=1}^{N} f_i(W^0 + \Delta W). \tag{2}$$

- **Expectation Setting:** The objective is an expectation over a data distribution $\mathcal{D}$, for methods like Bernoulli-LoRA-MVR:

$$f(W^0 + \Delta W) = \mathbb{E}_{\xi \sim \mathcal{D}} \left[ f_\xi(W^0 + \Delta W) \right]. \tag{3}$$

We also address the **distributed optimization setting** for our proposed Federated Learning (FL) algorithms (e.g., Fed-Bernoulli-LoRA-QGD). Here, the goal is to minimize a global objective averaged over $M$ clients:

$$f(W^0 + \Delta W) = \frac{1}{M} \sum_{l=1}^{M} f_l(W^0 + \Delta W), \tag{4}$$

where $f_l$ is the local loss for client $l$. The goal is to find $\Delta W$ that minimizes this global objective.

In practical applications, LoRA is often applied to many matrices across multiple layers (e.g., query, key, value, and feed-forward projections). Our analysis covers this case as well: one can view all LoRA-modified matrices as being stacked or arranged in block-diagonal form inside a single $W^0$ and $\Delta W$. Because we work with the Frobenius norm and inner product, our assumptions and convergence results extend verbatim to this concatenated/block-diagonal representation, following the same abstraction used in Hu et al. (2022); Sun et al. (2024); Malinovsky et al. (2024); Xia et al. (2024); Zhu et al. (2024).

## 4 CONTRIBUTIONS

LoRA-based methods are sensitive to hyperparameters (Khodak et al., 2021; Kuang et al., 2024) and require a stronger theoretical foundation. While Malinovsky et al. (2024) provided an initial framework with RAC-LoRA, we aim to further advance the theory and versatility of low-rank adaptation.

Low-rank PEFT updates two matrices, $A$ and $B$, either individually or alternating deterministically (Malinovsky et al., 2024; Xia et al., 2024; Zhu et al., 2024). Our main contribution, Bernoulli-LoRA, is a generic framework with a probabilistic update: at each step, a Bernoulli trial selects either $A$ or $B$ for optimization while the other is fixed. This randomized selection unifies and generalizes existing update strategies. Similar to COLA (Xia et al., 2024), our framework applies a sequence of these probabilistic low-rank updates.

Our analysis uses standard non-convex optimization assumptions, like $L$-smoothness. We instantiate Bernoulli-LoRA with several algorithms, from foundational gradient methods to advanced stochastic, variance-reduced, and federated learning variants. We establish rigorous convergence guarantees for each method. Our key contributions include:

◆ **Foundational Algorithmic Variants:** We establish the framework's properties with two fundamental schemes to understand the interplay between randomized selection and standard descent.
  – Bernoulli-LoRA-GD (Algorithm 2) uses full gradients, providing a foundational analysis of convergence in an idealized setting.
  – Bernoulli-LoRA-SGD (Algorithm 4) uses practical stochastic gradients, offering insights into the interplay of stochasticity and randomized adaptation for large-scale tasks.

◆ **Advanced Variance Reduction for Non-Convex Optimization:** To counter variance from stochastic gradients, we develop VR-enhanced variants, providing the first theoretical analysis of LoRA-type methods with advanced VR schemes in $L$-smooth non-convex settings.
  – Bernoulli-LoRA-PAGE (Algorithm 6) adapts the optimal and simple PAGE (Li et al., 2021) for the finite-sum setting (2).
  – Bernoulli-LoRA-MVR (Algorithm 5) uses Momentum Variance Reduction inspired by STORM (Cutkosky & Orabona, 2019) for the expectation setting, proving its effectiveness in our framework.

◆ **Communication-Efficient Federated Learning Extensions:** We extend Bernoulli-LoRA to FL, addressing communication overhead. We provide the first comprehensive analysis of LoRA-type methods integrated with established communication-saving techniques like quantization, gradient difference compression, and error feedback.
  – Fed-Bernoulli-LoRA-QGD (Algorithm 7) incorporates QSGD-style quantization (Alistarh et al., 2017; Wen et al., 2017; Horváth et al., 2022; Panferov et al., 2024) to compress gradients and reduce communication bandwidth.
  – Fed-Bernoulli-LoRA-MARINA (Algorithm 8) adapts the MARINA strategy (Gorbunov et al., 2021) to efficiently compress gradient differences.

– Fed-Bernoulli-LoRA-EF21 (Algorithm 9) integrates the EF21 error feedback mechanism (Richtárik et al., 2021) to stabilize training with contractive compressors.

◆ **Analysis for Non-Smooth Convex Functions:** We broaden our framework's applicability by providing the first theoretical analysis of LoRA-type methods for non-smooth convex optimization. We present a version of Bernoulli-LoRA-GD (Algorithm 3) and establish its convergence rates with different stepsize policies.

## 5 BERNOULLI-LORA FRAMEWORK

In this section, we introduce the Bernoulli-LoRA framework, a novel and generic approach for low-rank adaptation. The core idea is to perform a sequence of low-rank updates, where at each step, a probabilistic choice determines which of the two factor matrices ($A$ or $B$) is trained. This randomized mechanism, formalized in Algorithm 1, not only provides a flexible and unifying theoretical construct for existing LoRA-style methods but also allows for a rigorous convergence analysis.

At each iteration, one of the two low-rank matrices is sampled from a fixed distribution and remains frozen, while the other is trained to minimize the objective. This strategy prevents optimization from being confined to a fixed subspace, reducing the risk of converging to a suboptimal point. We formalize these two configurations as Left and Right sketch updates.

**Definition 1** (Left and Right Sketch Updates). *We define two complementary update rules based on which factor matrix is sampled from a fixed distribution and which is adjustable. The **Left Sketch** and **Right Sketch** updates are given, respectively, by:*

$$\Delta W = \frac{\alpha}{r} B_S \hat{A}, \quad \text{with } B_S \sim \mathcal{D}_B \text{ fixed and } \hat{A} \in \mathbb{R}^{r \times n} \text{ adjustable}, \tag{5}$$

$$\Delta W = \frac{\alpha}{r} \hat{B} A_S, \quad \text{with } A_S \sim \mathcal{D}_A \text{ fixed and } \hat{B} \in \mathbb{R}^{m \times r} \text{ adjustable}, \tag{6}$$

*where $\mathcal{D}_B$ and $\mathcal{D}_A$ are fixed distributions over $\mathbb{R}^{m \times r}$ and $\mathbb{R}^{r \times n}$ matrices.*

---

**Algorithm 1** Bernoulli-LoRA Framework

---

1: **Parameters:** pre-trained model $W^0 \in \mathbb{R}^{m \times n}$, rank $r \ll \min\{m,n\}$, scaling factor $\alpha > 0$, chain length $T$, sketch distributions $\mathcal{D}_S^B$ and $\mathcal{D}_S^A$, Bernoulli probability $p$.
2: **for** $t = 0, 1, \ldots, T-1$ **do**
3:     Sample $c^t \sim \text{Be}(p)$                                     Bernoulli random variable
4:     **if** $c^t = 1$ **then**
5:         Sample $B_S^t \sim \mathcal{D}_S^B$                                  (Left sketch)
6:         Using a chosen optimizer, approximately solve $\hat{A}^t \approx \arg\min_A f(W^t + \frac{\alpha}{r} B_S^t A)$.
7:         $W^{t+1} = W^t + \frac{\alpha}{r} B_S^t \hat{A}^t$.
8:     **else**
9:         Sample $A_S^t \sim \mathcal{D}_S^A$                                  (Right sketch)
10:        Using a chosen optimizer, approximately solve $\hat{B}^t \approx \arg\min_B f(W^t + \frac{\alpha}{r} B A_S^t)$.
11:        $W^{t+1} = W^t + \frac{\alpha}{r} \hat{B}^t A_S^t$.
12:     **end if**
13: **end for**

---

### 5.1 REFORMULATION AS A PROJECTED GRADIENT STEP

Building upon the work of Malinovsky et al. (2024) on their RAC-LoRA framework, the update steps in Algorithm 1 can be reformulated as projected gradient steps. The subproblems in lines 6 and 10 are typically solved approximately, for instance, by taking a single step of a suitable optimizer like Gradient Descent (GD) or its variants. More discussion can be found in Appendix E.

While RAC-LoRA employs a deterministic choice for which matrix to update, our Bernoulli-LoRA framework generalizes this concept by introducing a probabilistic selection at each step. This allows us to express the update for any of our proposed methods in a single, unified form:

$$W^{t+1} = W^t - \gamma \hat{G}^t, \tag{7}$$

| Setting | Method & Base Gradient Estimator $G^t$ | NC convergence rate | PŁ convergence rate |
|---|---|---|---|
| (1) | Bernoulli-LoRA-GD (Alg. 2) $G^t = \nabla f(W^t)$ | Thm. 1: $\frac{\Delta^0}{\gamma \lambda_{\min} T}$ | Thm. 9: $(1 - \gamma \mu \lambda_{\min})^T \Delta^0$ |
| (1) | Bernoulli-LoRA-SGD (Alg. 4) $G^t = g(W^t)$ | Thm. 2: $\frac{\Delta^0}{\gamma \lambda_{\min} T} + \frac{\gamma L C_1 \lambda_{\max}}{\lambda_{\min}}$ | Thm. 12: $(1 - \gamma \mu \lambda_{\min})^T \Delta^0 + \frac{\gamma L C_1 \lambda_{\max}}{\mu \lambda_{\min}}$ |
| (1)+(3) | Bernoulli-LoRA-MVR (Alg. 5) $G^t = \nabla f_{\xi^t}(W^t) + (1 - b)(G^{t-1} - \nabla f_{\xi^t}(W^{t-1}))$ | Thm. 3: $\frac{\Phi_1}{\gamma \lambda_{\min} T} + \frac{b \sigma^2 \lambda_{\max}}{(2 - b) \lambda_{\min}}$ (1) | Thm. 14: $(1 - \gamma \mu \lambda_{\min})^T \Phi_1 + \frac{b \sigma^2 \lambda_{\max}}{(2 - b) \mu \lambda_{\min}}$ (1) |
| (1)+(2) | Bernoulli-LoRA-PAGE (Alg. 6) $G^t = \begin{cases} \nabla f(W^t), & \text{w.p. } q \\ G^{t-1} + \nabla f_{i_t}(W^t) - \nabla f_{i_t}(W^{t-1}), & \text{w.p. } 1 - q \end{cases}$ | Thm. 4: $\frac{\Phi_2}{\gamma \lambda_{\min} T}$ (2) | Thm. 16: $(1 - \gamma \mu \lambda_{\min})^T \Phi_2$ (2) |
| (1)+(4) | Fed-Bernoulli-LoRA-QGD (Alg. 7) $G^t = \frac{1}{M} \sum_{l=1}^M \mathcal{Q}_l^t(\nabla f_l(W^t))$ | Thm. 5: $\frac{\Delta^0}{\gamma \lambda_{\min} T} + \frac{\gamma L \omega \Delta^* \lambda_{\max}}{M \lambda_{\min}}$ | Thm. 18: $(1 - \gamma \mu \lambda_{\min})^T \Delta^0 + \frac{\gamma L^2 \omega \lambda_{\max}}{M \mu \lambda_{\min}}$ |
| (1)+(4) | Fed-Bernoulli-LoRA-MARINA (Alg. 8) $G_l^t = \begin{cases} \nabla f_l(W^t), & \text{w.p. } q \\ G_l^{t-1} + \mathcal{Q}_l^t(\nabla f_l(W^t) - \nabla f_l(W^{t-1})), & \text{w.p. } 1 - q \end{cases}$ $G^t = \frac{1}{M} \sum_{l=1}^M G_l^t$ | Thm. 6: $\frac{\Phi_2}{\gamma \lambda_{\min} T}$ (2) | Thm. 20: $(1 - \gamma \mu \lambda_{\min})^T \Phi_2$ (2) |
| (1)+(4) | Fed-Bernoulli-LoRA-EF21 (Alg. 9) $G_l^t = G_l^{t-1} + \mathcal{C}_l^t(\nabla f_l(W^t) - G_l^{t-1})$ $G^t = \frac{1}{M} \sum_{l=1}^M G_l^t$ | Thm. 7: $\frac{\Phi_3}{\gamma \lambda_{\min} T}$ (3) | Thm. 22: $(1 - \gamma \mu \lambda_{\min})^T \Phi_3$ (3) |

(1) $\Phi_1 := \Delta^0 + \frac{\gamma}{b(2 - b)} \mathcal{G}^0$;

(2) $\Phi_2 := \Delta^0 + \frac{\gamma}{q} \mathcal{G}^0$;

(3) $\Phi_3 := \Delta^0 + \frac{\gamma}{1 - \sqrt{1 - \beta}} \hat{\mathcal{G}}^0$.

Table 1: Unified summary of the proposed methods, their base gradient estimators, and convergence rates for smooth non-convex ("NC") and PŁ settings. All methods follow the general update rule $W^{t+1} = W^t - \gamma \hat{G}^t$, where the projected estimator $\hat{G}^t$ is defined in (8). The table specifies the definition of the base gradient estimator $G^t$ for each method. Absolute constant factors are omitted. Notation: $\Delta^0 := f(W^0) - f^*$; $\mathcal{G}^0 := \|G^0 - \nabla f(W^0)\|_F^2$; $\hat{\mathcal{G}}^0 := \frac{1}{M} \sum_{l=1}^M \|G_l^0 - \nabla f_l(W^0)\|_F^2$; $T$ is the chain length; $\omega$ is the compression parameter; $\Delta^* := f^* - \frac{1}{M} \sum_{l=1}^M f_l^*$; $C_1$ is a constant from Asm. 4; $q$ is the probability of a full gradient computation; $\beta$ is the contractive compression parameter; $b$ is the momentum parameter; $\lambda_{\min} = \lambda_{\min}^p := p \lambda_{\min}^{H_B} + (1 - p) \lambda_{\min}^{H_A}$, and $\lambda_{\max} = \lambda_{\max}^p := p \lambda_{\max}^{H_B} + (1 - p) \lambda_{\max}^{H_A}$.

where $\hat{G}^t$ is the *projected gradient estimator*. It is formed by taking a *base gradient estimator* $G^t$ (e.g., a full gradient, a stochastic gradient, or a variance-reduced one) and projecting it based on the outcome of a Bernoulli trial:

$$\hat{G}^t = \begin{cases} H_B^t G^t, & \text{with probability } p \\ G^t H_A^t, & \text{with probability } 1 - p \end{cases}. \tag{8}$$

The specific choice of the base estimator $G^t$ defines the particular algorithm within the Bernoulli-LoRA family. We summarize our proposed methods and their convergence guarantees in Table 1 and describe them next.

## 6 CONVERGENCE RESULTS

The optimization dynamics of our framework depend on the spectral properties of the expected projection matrix (Section 5.1). To derive non-asymptotic guarantees, we rely on standard modeling abstractions used in the analysis of first-order methods (e.g., Lipschitz smoothness, PŁ condition). Our results are conditional on these assumptions, consistent with classical analyses of GD, SGD, and FL (Bottou et al., 2018; Bubeck, 2015; Nesterov, 2018; Khaled & Richtárik, 2023).

**Assumption 1.** *(Positive Expected Projection) Consider the projection matrices associated with the Left and Right Sketch updates:*

$$H_B := B_S(B_S^\top B_S)^\dagger B_S^\top \quad and \quad H_A := A_S^\top (A_S A_S^\top)^\dagger A_S,$$

*where $\dagger$ denotes the Moore-Penrose pseudoinverse. We assume that for the sampling distributions $\mathcal{D}_S^B$ and $\mathcal{D}_S^A$, the smallest eigenvalues of the expected projection matrices are strictly positive:*

$$\lambda_{\min}^{H_B} = \lambda_{\min}(\mathbb{E}[H_B]) > 0 \quad and \quad \lambda_{\min}^{H_A} = \lambda_{\min}(\mathbb{E}[H_A]) > 0.$$

**Remark 1** (On the practicality of Assumption 1)**.** *At first sight Assumption 1 may look restrictive: every single projector has eigenvalues in $\{0,1\}$, so $\lambda_{\min}(H_B) = \lambda_{\min}(H_A) = 0$ whenever $r < m$ or $r < n$. Crucially, we never require* individual *projectors to be positive definite, only their* expectation *over the sketch distribution. Intuitively, each step updates along a low-dimensional subspace, but the random subspaces collectively "cover" all directions over time. In fact, the assumption is mild: as shown in Section D, it is satisfied with $\mathbb{E}[H_B] = \frac{r}{m}I_m$, $\mathbb{E}[H_A] = \frac{r}{n}I_n$ for standard choices such as Gaussian, i.i.d. uniform, Kaiming-uniform, and SVD-based orthonormal sketches widely used in practice (Xia et al., 2024; Mao et al., 2025; Zhu et al., 2024; Hayou et al., 2024; Li et al., 2025; Kopiczko et al., 2023).*

**Assumption 2.** *(Lower Bounded Function) The objective function $f$ has a finite infimum $f^* \in \mathbb{R}$.*

Following classical literature (Nemirovski et al., 2009; Beck, 2017; Duchi, 2018; Lan, 2020; Drusvyatskiy, 2020; Nesterov, 2018), we seek an $\varepsilon$-suboptimal solution for convex (or PŁ) objectives, satisfying

$$\mathbb{E}\left[f(\hat{W}) - f(W^*)\right] \le \varepsilon, \tag{9}$$

where $W^*$ minimizes $f$. For smooth non-convex problems, we aim for an $\varepsilon$-stationary point $\hat{W}$ such that

$$\mathbb{E}\left[\left\|\nabla f(\hat{W})\right\|_{\mathrm{F}}^2\right] \le \varepsilon^2. \tag{10}$$

We quantify algorithmic efficiency via iteration complexity. To establish convergence rates, we use the standard assumption of gradient Lipschitz continuity (Bubeck, 2015; Nesterov, 2018; Beck, 2017; Demidovich et al., 2023b; Khaled & Richtárik, 2023; Bottou et al., 2018; Sun, 2020).

**Assumption 3.** *(Lipschitz Smooth Gradient) A function $f$ is differentiable, and there exists a constant $L > 0$ such that*

$$\|\nabla f(W) - \nabla f(V)\|_{\mathrm{F}} \le L \|W - V\|_{\mathrm{F}},$$

*for all $W, V \in \mathbb{R}^{m \times n}$.*

To unify our analysis, we define a probability-weighted eigenvalue $\lambda_{\min(\max)}^p := p\lambda_{\min(\max)}^{H_B} + (1 - p)\lambda_{\min(\max)}^{H_A}$. Let $\widetilde{W}^T$ be an iterate drawn randomly from the sequence $\{W^0, W^1, \ldots, W^{T-1}\}$, with the specific sampling distribution depending on the method.

We begin by presenting the convergence result for the foundational Bernoulli-LoRA-GD method.

**Theorem 1** (Smooth Non-Convex Setting)**.** *Let Assumptions 1, 2, and 3 hold, and let the stepsize satisfy $0 < \gamma \le \frac{1}{L}$. Then the iterates of Bernoulli-LoRA-GD (Algorithm 2), with matrices $\hat{A}^t$ and $\hat{B}^t$ computed according to Lemma 10, satisfy*

$$\mathbb{E}\left[\left\|\nabla f(\widetilde{W}^T)\right\|_{\mathrm{F}}^2\right] \le \frac{2\Delta^0}{\gamma \lambda_{\min}^p T},$$

*where $\Delta^0 := f(W^0) - f^*$.*

While insightful, full-gradient methods are often impractical for large-scale problems. We therefore extend our analysis to the stochastic setting, where the gradient is replaced by an unbiased estimator $g(W)$. For this, we use the general *expected smoothness* assumption.

**Assumption 4** (Expected Smoothness (Khaled & Richtárik, 2023))**.** *The stochastic gradient estimator $g(W)$ satisfies*

$$\mathbb{E}\left[\|g(W)\|_{\mathrm{F}}^2\right] \le 2A_1\left(f(W) - f^*\right) + B_1 \cdot \|\nabla f(W)\|_{\mathrm{F}}^2 + C_1,$$

*for some constants $A_1, B_1, C_1 \ge 0$ and all $W \in \mathbb{R}^{m \times n}$.*

The following theorem establishes the convergence for Bernoulli-LoRA-SGD. Its proof is in Appendix H.2.

**Theorem 2.** *Let Assumptions 2, 3, and 4 hold, and let the stepsize satisfy*

$$0 < \gamma \leq \min\left\{\frac{1}{\sqrt{LA_1\lambda^p_{\max}T}}, \frac{1}{LB_1}\left(\frac{\lambda^p_{\max}}{\lambda^p_{\min}}\right)^{-1}\right\}.$$

*Then the iterates generated by* Bernoulli-LoRA-SGD *(Algorithm 4) satisfy*

$$\mathbb{E}\left[\left\|\nabla f(\widetilde{W}^T)\right\|_F^2\right] \leq \frac{6\Delta^0}{\gamma\lambda^p_{\min}T} + \gamma LC_1 \cdot \frac{\lambda^p_{\max}}{\lambda^p_{\min}},$$

*where $\Delta^0 := f(W^0) - f^*$.*

To analyze our variance-reduced methods, we consider a specific bounded variance assumption.

**Assumption 5** (Bounded Variance (Nemirovski et al., 2009))**.** *There exists a constant $\sigma > 0$ such that, for all $W \in \mathbb{R}^{m \times n}$,*

$$\mathbb{E}\left[\nabla f_\xi(W)\right] = \nabla f(W), \qquad \mathbb{E}\left[\left\|\nabla f_\xi(W) - \nabla f(W)\right\|_F^2\right] \leq \sigma^2.$$

The next result establishes convergence for Bernoulli-LoRA-MVR.

**Theorem 3.** *Let Assumptions 1, 2, 3, and 5 hold, and let the stepsize satisfy $0 < \gamma \leq \frac{1}{L\left(1+\sqrt{\frac{2\lambda^p_{\max}(1-b)^2}{b}}\right)}$. Then the iterates of* Bernoulli-LoRA-MVR *(Algorithm 5) satisfy*

$$\mathbb{E}\left[\left\|\nabla f(\widetilde{W}^T)\right\|_F^2\right] \leq \frac{2\Delta^0}{\gamma\lambda^p_{\min}T} + \left(\frac{\mathcal{G}^0}{bT} + \frac{2b\sigma^2}{2-b}\right) \cdot \frac{\lambda^p_{\max}}{\lambda^p_{\min}},$$

*where $\Delta^0 := f(W^0) - f^*$ and $\mathcal{G}^0 := \left\|G^0 - \nabla f(W^0)\right\|_F^2$.*

For the finite-sum setting, we analyze Bernoulli-LoRA-PAGE, with its convergence detailed in the following theorem and proven in Appendix H.4.

**Theorem 4.** *Let Assumptions 1, 2, and 3 hold, and let the stepsize satisfy $0 < \gamma \leq \frac{1}{L\left(1+\sqrt{\frac{1-q}{q}\lambda^p_{\max}}\right)}$.*

*Then the iterates of* Bernoulli-LoRA-PAGE *(Algorithm 6) satisfy*

$$\mathbb{E}\left[\left\|\nabla f(\widetilde{W}^T)\right\|_F^2\right] \leq \frac{2\Delta^0}{\gamma\lambda^p_{\min}T} + \frac{\mathcal{G}^0}{qT} \cdot \frac{\lambda^p_{\max}}{\lambda^p_{\min}},$$

*where $\Delta^0 := f(W^0) - f^*$ and $\mathcal{G}^0 := \left\|G^0 - \nabla f(W^0)\right\|_F^2$.*

We now shift to our Federated Learning variants. The following theorem provides convergence guarantees for Fed-Bernoulli-LoRA-QGD, with the proof available in Appendix I.1.

**Theorem 5.** *Let Assumptions 1, 2, 3, and 11 hold, and let the stepsize satisfy $0 < \gamma \leq \min\left\{\frac{1}{L\sqrt{\frac{\omega}{M}\lambda^p_{\max}T}}, \frac{1}{L}\left(\frac{\lambda^p_{\max}}{\lambda^p_{\min}}\right)^{-1}\right\}$. Then the iterates of* Fed-Bernoulli-LoRA-QGD *(Algorithm 7) satisfy*

$$\mathbb{E}\left[\left\|\nabla f(\widetilde{W}^T)\right\|_F^2\right] \leq \frac{6\Delta^0}{\gamma\lambda^p_{\min}T} + \frac{2\gamma L\omega\Delta^*}{M} \cdot \frac{\lambda^p_{\max}}{\lambda^p_{\min}},$$

*where $\Delta^0 := f(W^0) - f^*$.*

Next, we present the convergence result for Fed-Bernoulli-LoRA-MARINA. The proof can be found in Appendix I.2.

**Theorem 6.** *Let Assumptions 1, 2, and 3 hold, and let the stepsize satisfy $0 < \gamma \leq \frac{1}{L\left(1+\sqrt{\lambda^p_{\max}\frac{1-q}{q}\cdot\frac{\omega}{M}}\right)}$. Then the iterates of* Fed-Bernoulli-LoRA-MARINA *(Algorithm 8) satisfy*

$$\mathbb{E}\left[\left\|\nabla f(\widetilde{W}^T)\right\|_F^2\right] \leq \frac{2\Delta^0}{\gamma\lambda^p_{\min}T} + \frac{\mathcal{G}^0}{qT} \cdot \frac{\lambda^p_{\max}}{\lambda^p_{\min}},$$

*where $\Delta^0 := f(W^0) - f^*$ and $\mathcal{G}^0 := \left\|G^0 - \nabla f(W^0)\right\|_F^2$.*

The convergence of Fed-Bernoulli-LoRA-EF21 is established below, with a detailed proof in Appendix I.3.

**Theorem 7.** *Let Assumptions 1, 2, and 3 hold, and let the stepsize satisfy* $0 < \gamma \leq \frac{1}{L\left(1+\frac{\sqrt{\lambda_{\max}^p(1-\beta)}}{1-\sqrt{1-\beta}}\right)}$.

*Then the iterates of* Fed-Bernoulli-LoRA-EF21 *(Algorithm 9) satisfy*

$$\mathbb{E}\left[\left\|\nabla f(\widetilde{W}^T)\right\|_{\mathrm{F}}^2\right] \leq \frac{2\Delta^0}{\gamma\lambda_{\min}^p T} + \frac{2\hat{\mathcal{G}}^0}{\beta T} \cdot \frac{\lambda_{\max}^p}{\lambda_{\min}^p},$$

*where* $\Delta^0 := f(W^0) - f^*$ *and* $\hat{\mathcal{G}}^0 := \frac{1}{M}\sum_{l=1}^M \left\|G_l^0 - \nabla f_l(W^0)\right\|_{\mathrm{F}}^2$.

To obtain stronger, linear convergence rates, we introduce the Polyak–Łojasiewicz condition, a common generalization of strong convexity.

**Assumption 6** (Polyak–Łojasiewicz condition (Polyak, 1963; Lojasiewicz, 1963))**.** *There exists* $\mu > 0$ *such that*

$$\tfrac{1}{2}\|\nabla f(W)\|_{\mathrm{F}}^2 \geq \mu\left(f(W) - f^*\right).$$

The next theorem states the convergence of Bernoulli-LoRA-SGD under this condition. It is proven in Appendix H.2.

**Theorem 8.** *Let Assumptions 2, 3, 4, and 6 hold, and let the stepsize satisfy*
$0 < \gamma \leq \min\left\{\frac{\mu\lambda_{\min}^p}{2LA_1\lambda_{\max}^p}, \frac{2}{\mu\lambda_{\min}^p}, \frac{1}{LB_1}\left(\frac{\lambda_{\max}^p}{\lambda_{\min}^p}\right)^{-1}\right\}$. *Then the iterates of* Bernoulli-LoRA-SGD *(Algorithm 4) satisfy*

$$\mathbb{E}\left[f(W^T) - f^*\right] \leq \left(1 - \frac{\gamma\mu\lambda_{\min}^p}{2}\right)^T \Delta^0 + \frac{\gamma LC_1}{\mu} \cdot \frac{\lambda_{\max}^p}{\lambda_{\min}^p},$$

*where* $\Delta^0 := f(W^0) - f^*$.

All other PŁ-condition results are relegated to the Appendix.

## 7 EXPERIMENTS

To validate our theoretical findings, we conducted numerical experiments across multiple machine learning tasks.

### 7.1 LINEAR REGRESSION WITH NON-CONVEX REGULARIZATION.

We begin with a controlled linear regression problem with non-convex regularization, split into pre-training and fine-tuning phases. We use $\widetilde{(\cdot)}$ for pre-training quantities and $\hat{(\cdot)}$ for fine-tuning. During the **pre-training phase**, we solve $\min_{x\in\mathbb{R}^n}\left\{\widetilde{f}(x) := \frac{1}{2\widetilde{m}}\left\|\widetilde{D}x - \widetilde{b}\right\|_2^2 + \widetilde{\lambda}\sum_{j=1}^d \frac{x_j^2}{1+x_j^2}\right\}$, where $\widetilde{D} \in \mathbb{R}^{\widetilde{m}\times n}$, $\widetilde{b} \in \mathbb{R}^{\widetilde{m}}$, $\widetilde{m} = 9 \times 10^4$, and $n = 4096$. We set $\widetilde{\lambda} = \left\|\widetilde{D}\right\|_2 \approx 18.2$, giving $\widetilde{L} \approx 54.7$. We optimize until $\|\nabla f(\widetilde{x}^*)\|^2 \leq 10^{-8}$ to obtain $\widetilde{x}^*$. For the **fine-tuning phase**, we use $\widetilde{x}^*$ as the initialization and then solve $\min_{x\in\mathbb{R}^n}\left\{\hat{f}(x) := \frac{1}{2\hat{m}}\left\|\hat{D}x - \hat{b}\right\|_2^2 + \hat{\lambda}\sum_{j=1}^d \frac{x_j^2}{1+x_j^2}\right\}$, where $\hat{D} \in \mathbb{R}^{\hat{m}\times n}$, $\hat{b} \in \mathbb{R}^{\hat{m}}$, and $\hat{m} = 10^4$. We keep $n = 4096$ and set $\hat{\lambda} = \left\|\hat{D}\right\|_2 \approx 4101.7$, yielding $\hat{L} \approx 12305.3$. This second phase uses a dataset with notably different characteristics to mirror realistic domain shifts.

**Stochastic setting.** We consider the stochastic setting, comparing RAC-LoRA-SGD, Bernoulli-LoRA-SGD, and Bernoulli-LoRA-PAGE. In all experiments, we use a batch size of 100, which corresponds to 1% of the data.

Figure 1 shows that Bernoulli-LoRA-PAGE successfully reduces variance and converges to the target tolerance, whereas all SGD variants stall at a certain accuracy. This underscores the practical advantage of Bernoulli-LoRA-PAGE over the baseline RAC-LoRA-SGD in the stochastic setting from an optimization standpoint.

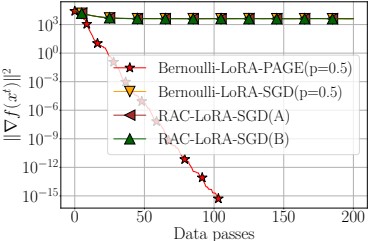

Figure 1: Comparison of RAC-LoRA-SGD, Bernoulli-LoRA-SGD and Bernoulli-LoRA-PAGE on linear regression fine-tuning. Curves with $p = 0.01, 0.2, \ldots$ indicate Bernoulli-LoRA sampling parameters. RAC-LoRA-SGD(A) trains $B$ after resampling $A$, while RAC-LoRA-SGD(B) does the reverse. All methods use $\gamma = c/\hat{L}$ with $c$ tuned individually.

## 7.2 MLP ON MNIST

In this section, we evaluate Bernoulli-LoRA against established baselines in parameter-efficient fine-tuning, following the setup of Malinovsky et al. (2024).

**Methodology.** We first pre-train a three-layer MLP on MNIST digits 0–4 (LeCun et al., 1998), then adapt it with various LoRA-type methods to classify digits 5–9. Only unseen classes are used for evaluation. All adaptations use rank $r = 1$ and train for 50 epochs with AdamW (Loshchilov, 2017) ($\beta_1 = 0.9$, $\beta_2 = 0.999$, $\epsilon = 10^{-8}$), a fixed learning rate of $2 \times 10^{-4}$, and batch size 128. Each method is run 20 times using different seeds, and Table 2 reports the median accuracy (with standard deviation). For Bernoulli-LoRA, we show the best median accuracy among all tested settings.

| Method | $\mathcal{D}_A$ | $\mathcal{D}_B$ | Acc. (test) | Train Params |
|---|---|---|---|---|
| FPFT | - | - | 99.5 | 54,700 |
| LoRA | Gaussian | Zero | $85.69 \pm 1.60$ | 1K |
| LoRA | Zero | Gaussian | $89.82 \pm 0.90$ | 1K |
| COLA | Gaussian | Zero | $93.32 \pm 0.50$ | 1K |
| COLA | Zero | Gaussian | $96.55 \pm 0.20$ | 1K |
| AsymmLoRA | Gaussian | Zero | $64.04 \pm 6.90$ | 133 |
| AsymmLoRA | Zero | Gaussian | $74.52 \pm 7.20$ | 912 |
| RAC-LoRA | Gaussian | Zero | $93.02 \pm 0.50$ | 133 |
| RAC-LoRA | Zero | Gaussian | $96.49 \pm 0.20$ | 912 |
| Bernoulli-LoRA [2] | Zero[1] | Gaussian | $96.46 \pm 0.17$ | $\approx 904$ |

[1] Although Bernoulli-LoRA prescribes probabilistic selection from the first iteration, a deterministic assignment of fixed and trainable matrices at initialization yielded better performance.
[2] Achieved with $p = 0.99$, giving an expected trainable parameter count $p \cdot 912 + (1 - p) \cdot 133 \approx 904$. Here, 912 and 133 are the parameter counts for matrices $A$ and $B$, respectively.

Table 2: Performance on MNIST classification using an MLP with rank $r$ and scaling $\alpha = 1$. For AsymmLoRA and RAC-LoRA, only the zero-initialized matrix is trained.

**Discussion.** From Table 2, standard LoRA attains roughly 86% of full-parameter fine-tuning (FPFT) accuracy, indicating room for improvements via chaining. COLA improves upon vanilla LoRA, though both lack formal convergence guarantees. AsymmLoRA approximates LoRA in practice (Sun et al., 2024) but similarly lacks convergence analysis, whereas RAC-LoRA and Bernoulli-LoRA both boost accuracy and have theoretical backing. Notably, Bernoulli-LoRA matches RAC-LoRA in generalization and also guarantees convergence. An additional benefit is that RAC-LoRA and Bernoulli-LoRA each train only one matrix per LoRA block, whereas COLA needs two. In RAC-LoRA, either $A$ or $B$ is trained deterministically; in Bernoulli-LoRA, the choice is probabilistic, yielding an expected $pmr + (1 - p)rn$ trainable parameters. This advantage is especially valuable in resource-constrained settings such as Federated Learning.

Detailed configurations, hardware specs, and dataset descriptions are provided in Appendix J.

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

# A  APPENDIX

## CONTENTS

# B BASIC FACTS AND USEFUL INEQUALITIES

**Tower property.** For any random variables $X$ and $Y$, we have
$$\mathbb{E}\left[\mathbb{E}\left[X \mid Y\right]\right] = \mathbb{E}\left[X\right]. \tag{11}$$

**Cauchy-Bunyakovsky-Schwarz inequality.** For any random variables $X$ and $Y$, we have
$$\left|\mathbb{E}\left[XY\right]\right| \leq \sqrt{\mathbb{E}\left[X^2\right]\mathbb{E}\left[Y^2\right]}. \tag{12}$$

**Variance decomposition.** For any random vector $X \in \mathbb{R}^d$ and any non-random $c \in \mathbb{R}^d$, we have
$$\mathbb{E}\left[\|X - c\|_2^2\right] = \mathbb{E}\left[\|X - \mathbb{E}\left[X\right]\|_2^2\right] + \|\mathbb{E}\left[X\right] - c\|_2^2. \tag{13}$$

**Jensen's inequality.** For any random vector $X \in \mathbb{R}^d$ and any convex function $g : \mathbb{R}^d \mapsto \mathbb{R}$, we have
$$g(\mathbb{E}\left[X\right]) \leq \mathbb{E}\left[g(X)\right]. \tag{14}$$

## C  NOTATION

For matrices $W \in \mathbb{R}^{m \times n}$, where $m$ and $n$ denote the input and output dimensions respectively, we employ the Frobenius norm $\|\cdot\|_{\mathrm{F}}$, defined as $\|W\|_{\mathrm{F}} = \sqrt{\mathrm{Tr}(W^\top W)}$, where $\mathrm{Tr}(\cdot)$ denotes the matrix trace. The inner product between two matrices $A$ and $B$ is denoted by $\langle A, B \rangle = \mathrm{Tr}(A^\top B)$. In our low-rank adaptation framework, $B \in \mathbb{R}^{m \times r}$ and $A \in \mathbb{R}^{r \times n}$ represent the factors of rank $r \ll \min\{m, n\}$. We use $\mathcal{O}(\cdot)$ to hide absolute constants. We denote $\Delta^0 := f(W^0) - f^*$, $\mathcal{G}^0 := \|G^0 - \nabla f(W^0)\|_{\mathrm{F}}^2$ and $\hat{\mathcal{G}}^0 := \frac{1}{M} \sum_{l=1}^{M} \|G_l^0 - \nabla f_l(W^0)\|_{\mathrm{F}}^2$. For differentiable functions $f$, the gradient $\nabla f(W) \in \mathbb{R}^{m \times n}$ is computed with respect to the trace inner product, while for non-smooth functions, the subgradient $\partial f(W) \in \mathbb{R}^{m \times n}$ is similarly defined. The superscript $\dagger$ denotes the Moore-Penrose pseudoinverse.

# D    DISCUSSION ON POSITIVE EXPECTED PROJECTION (ASSUMPTION 1)

Recall that in our Bernoulli-LoRA framework, at each iteration we update only one of the low-rank factors ($A$ or $B$) while the other is treated as a fixed "sketch" sampled from a prescribed distribution. The resulting updates can be written as projected gradient steps with respect to the full parameter matrix $W$:

$$W^{t+1} = W^t - \gamma \hat{G}^t,$$

where the projected estimator $\hat{G}^t$ has the form

$$\hat{G}^t = \begin{cases} H_B^t G^t, & \text{(left sketch)}, \\ G^t H_A^t, & \text{(right sketch)}, \end{cases}$$

and $H_B^t$ and $H_A^t$ are projection matrices defined by the current sketch. In particular, for a left sketch we use

$$H_B := B \left(B^\top B\right)^\dagger B^\top \in \mathbb{R}^{m \times m}, \tag{15}$$

with $B \in \mathbb{R}^{m \times r}$, and for a right sketch we use

$$H_A := A^\top \left(A A^\top\right)^\dagger A \in \mathbb{R}^{n \times n}, \tag{16}$$

with $A \in \mathbb{R}^{r \times n}$. Here $\dagger$ denotes the Moore–Penrose pseudoinverse. Both $H_B$ and $H_A$ are orthogonal projectors onto the column spaces of $B$ and $A^\top$, respectively:

$$H_B^2 = H_B, \quad H_B^\top = H_B, \quad \text{Tr}(H_B) = \text{rank}(H_B) \le r,$$

$$H_A^2 = H_A, \quad H_A^\top = H_A, \quad \text{Tr}(H_A) = \text{rank}(H_A) \le r.$$

Our convergence guarantees are derived under Assumption 1, which requires the smallest eigenvalues of the *expected* projection matrices to be strictly positive:

$$\lambda_{\min}\left(\mathbb{E}\left[H_B\right]\right) > 0, \qquad \lambda_{\min}\left(\mathbb{E}\left[H_A\right]\right) > 0.$$

At first glance this may appear restrictive: any single projector has eigenvalues in $\{0,1\}$, so $\lambda_{\min}(H_B) = 0$ and $\lambda_{\min}(H_A) = 0$ whenever $r < m$ or $r < n$. However, the key point is that we never require *individual* projectors to be positive definite. Instead, we only require that the *average* projection (over the random sketches) be positive definite. Intuitively, this means that while each update acts in a low-dimensional subspace, the sequence of random subspaces collectively "covers" all directions over time.

In this section we show that Assumption 1 is satisfied for several widely used sketch distributions, including Gaussian, i.i.d. uniform, Kaiming-uniform and random orthonormal initializations. Our strategy is to exploit symmetry: for many random matrix ensembles the expected projection commutes with a large group of orthogonal transformations, which forces it to be a scalar multiple of the identity. The scalar is then determined by the rank/trace constraint.

## D.1    ROTATIONAL AND SIGNED-PERMUTATION SYMMETRIES

We begin with a classical result: if a matrix commutes with every orthogonal matrix, it must be a scalar multiple of the identity.

**Lemma 1** (Rotational invariance implies scalar matrix). *Let $M \in \mathbb{R}^{n \times n}$ be a matrix satisfying*

$$M = Q M Q^\top \quad \text{for all orthonormal matrices } Q \in \mathbb{R}^{n \times n}. \tag{17}$$

*Then $M = \alpha I_n$ for some scalar $\alpha \in \mathbb{R}$.*

*Proof.* Condition (17) is equivalent to $QM = MQ$ for all orthogonal $Q$, i.e., $M$ commutes with every orthogonal transformation. In particular, $M$ commutes with all rotations.

Since $M$ is a real symmetric matrix (indeed, $M = QMQ^\top$ for all orthogonal $Q$ implies $M^\top = M$), it admits an orthonormal eigenbasis. Let $v$ be an eigenvector of $M$ with eigenvalue $\lambda$, and normalize $v$ to unit length:

$$u_1 := \frac{v}{\|v\|}.$$

Then $M u_1 = \lambda u_1$.

Take any other unit vector $u$ on the sphere $S^{n-1}$. There exists an orthogonal matrix $Q \in \mathbb{R}^{n \times n}$ such that $u = Q u_1$ (geometrically, $Q$ is a rotation sending $u_1$ to $u$). Using $QM = MQ$,

$$Mu = M(Q u_1) = Q M u_1 = Q(\lambda u_1) = \lambda(Q u_1) = \lambda u.$$

Thus *every* unit vector $u$ is an eigenvector of $M$ with the same eigenvalue $\lambda$.

Now let $x \in \mathbb{R}^n$ be arbitrary and non-zero, and write $x = \|x\| \, u_x$ with $u_x := x / \|x\|$ a unit vector. Then
$$Mx = M(\|x\| \, u_x) = \|x\| \, Mu_x = \|x\| \, (\lambda u_x) = \lambda x.$$
So every vector $x$ is an eigenvector with eigenvalue $\lambda$, which implies $M = \lambda I_n$. Setting $\alpha = \lambda$ completes the proof. $\qquad\square$

For many random initializations we do not have full rotational invariance, but we *do* have invariance under row permutations and independent sign flips. The corresponding group is the set of all signed permutation matrices
$$G_n := \left\{ Q \in \mathbb{R}^{n \times n} : Q = PD, \ P \text{ permutation}, \ D = \operatorname{diag}(\pm 1, \ldots, \pm 1) \right\}.$$
The following lemma shows that invariance under $G_n$ is already enough to force a scalar matrix.

**Lemma 2** (Signed-permutation invariance implies scalar matrix). *Let $M \in \mathbb{R}^{n \times n}$ satisfy*
$$QMQ^\top = M \quad \text{for all } Q \in G_n. \tag{18}$$
*Then $M = \alpha I_n$ for some $\alpha \in \mathbb{R}$.*

*Proof.* We write $M = (m_{ij})$ to mean that $m_{ij}$ is the entry of $M$ in row $i$ and column $j$.

**Step 1: sign-flip invariance forces $M$ to be diagonal.** First consider only those $Q \in G_n$ that are pure sign-flip matrices, i.e.,
$$Q = D = \operatorname{diag}(q_{11}, \ldots, q_{nn}), \qquad q_{ii} \in \{\pm 1\}.$$
These are orthogonal and belong to $G_n$ (with $P = I_n$). For such $Q$, the $(i,j)$-entry of $QMQ^\top$ is
$$\left( QMQ^\top \right)_{ij} = \sum_{k,\ell} q_{ik} m_{k\ell} q_{j\ell} = q_{ii} m_{ij} q_{jj},$$
because $Q$ is diagonal. By (18),
$$q_{ii} q_{jj} m_{ij} = m_{ij} \quad \text{for all } i,j \text{ and all } (q_{11}, \ldots, q_{nn}) \in \{\pm 1\}^n. \tag{19}$$
Fix any $i \neq j$. We are free to choose $q_{ii}$ and $q_{jj}$ independently. Let $q_{ii} = 1$, $q_{jj} = -1$ and $q_{kk} = 1$ for all $k \notin \{i,j\}$. Then (19) yields
$$(-1) \, m_{ij} = m_{ij} \quad \Longrightarrow \quad m_{ij} = 0.$$
Since $i \neq j$ was arbitrary, all off-diagonal entries vanish, and $M$ must be diagonal:
$$M = \operatorname{diag}(m_{11}, \ldots, m_{nn}).$$

**Step 2: permutation invariance forces all diagonal entries to coincide.** Next consider permutation matrices $P \in G_n$, i.e., matrices with exactly one entry equal to 1 in each row and column (and all other entries 0). Each $P$ is orthogonal and belongs to $G_n$ (with $D = I_n$), so by (18),
$$PMP^\top = M. \tag{20}$$
Let $\pi$ be the permutation of $\{1, \ldots, n\}$ represented by $P$, so that $Pe_j = e_{\pi(j)}$ for the standard basis vectors. One checks that
$$\left( PMP^\top \right)_{ii} = m_{\pi(i)\pi(i)},$$
so (20) implies
$$m_{ii} = m_{\pi(i)\pi(i)} \quad \text{for all } i \text{ and all permutations } \pi.$$
This is only possible if all diagonal entries are equal to a common value $\alpha \in \mathbb{R}$:
$$m_{11} = \cdots = m_{nn} = \alpha.$$
Therefore $M = \alpha I_n$.

**Step 3: consistency with general signed permutations.** In the argument above we only used two special subgroups of $G_n$: pure sign flips ($P = I_n$) and pure permutations ($D = I_n$). Since both are contained in $G_n$, the assumption (18) applies to them. Once we have shown that $M = \alpha I_n$, it is immediate that $QMQ^\top = M$ holds for all $Q = PD \in G_n$:
$$QMQ^\top = (PD)(\alpha I_n)(D^\top P^\top) = \alpha PDD^\top P^\top = \alpha I_n = M.$$
This completes the proof. $\qquad\square$

We will use Lemma 1 in the Gaussian case (where full rotational invariance holds) and Lemma 2 in the uniform and Kaiming cases (where we have signed-permutation invariance).

## D.2 GAUSSIAN INITIALIZATION

Gaussian sketches are a standard choice in LoRA-style methods; see, for example, Xia et al. (2024); Mao et al. (2025). The next lemma shows that for Gaussian initialization, the expected projection matrices are isotropic and their eigenvalues are exactly $r/m$ and $r/n$.

**Lemma 3** (Expected projections for Gaussian sketches). *Let* $r \leq \min\{m,n\}$ *and consider two random matrices:*

◆ $B \in \mathbb{R}^{m \times r}$ *with entries i.i.d.* $\mathcal{N}(0,1)$,

◆ $A \in \mathbb{R}^{r \times n}$ *with entries i.i.d.* $\mathcal{N}(0,1)$.

*Define* $H_B$ *and* $H_A$ *as in* (15) *and* (16). *Then*
$$\mathbb{E}[H_B] = \frac{r}{m} I_m, \qquad \mathbb{E}[H_A] = \frac{r}{n} I_n,$$
*which implies*
$$\lambda_{\min}(\mathbb{E}[H_B]) = \frac{r}{m}, \qquad \lambda_{\min}(\mathbb{E}[H_A]) = \frac{r}{n}.$$

*Proof.* We first prove the statement for $H_B$, then explain the analogous argument for $H_A$.

**Step 1:** $\mathbb{E}[H_B]$ **is a scalar multiple of the identity.** Let $B \in \mathbb{R}^{m \times r}$ with i.i.d. $\mathcal{N}(0,1)$ entries, and let $Q \in \mathbb{R}^{m \times m}$ be an arbitrary orthogonal matrix. By rotational invariance of the standard Gaussian distribution,
$$QB \overset{d}{=} B.$$
Consider the projector built from $QB$:
$$\begin{aligned} H_{QB} &:= (QB)\left((QB)^\top QB\right)^\dagger (QB)^\top \\ &= QB\left(B^\top Q^\top QB\right)^\dagger B^\top Q^\top \\ &= QB\left(B^\top B\right)^\dagger B^\top Q^\top \\ &= Q\left(B(B^\top B)^\dagger B^\top\right)Q^\top \\ &= QH_B Q^\top. \end{aligned}$$
Since $QB$ and $B$ are identically distributed, $H_{QB}$ and $H_B$ have the same distribution and hence the same expectation:
$$\mathbb{E}[H_{QB}] = \mathbb{E}[H_B].$$
Using $H_{QB} = QH_B Q^\top$ and linearity of expectation,
$$\mathbb{E}[H_B] = \mathbb{E}[H_{QB}] = \mathbb{E}[QH_B Q^\top] = Q\,\mathbb{E}[H_B]\,Q^\top \quad \text{for all orthogonal } Q \in \mathbb{R}^{m \times m}.$$
By Lemma 1, a matrix commuting with all orthogonal matrices must be a scalar multiple of the identity. Hence there exists $\alpha \in \mathbb{R}$ such that
$$\mathbb{E}[H_B] = \alpha I_m.$$

**Step 2: determine** $\alpha$ **via the rank/trace.** For any realization of $B$ with full column rank (which holds almost surely, since $B$ has i.i.d. continuous entries and $r \leq m$), the matrix $H_B$ is the orthogonal projector onto the $r$-dimensional column space of $B$. Thus
$$\mathrm{rank}(H_B) = r, \qquad \mathrm{Tr}(H_B) = r.$$
Taking expectations and using linearity of the trace,
$$\mathrm{Tr}(\mathbb{E}[H_B]) = \mathbb{E}[\mathrm{Tr}(H_B)] = r.$$
Since $\mathbb{E}[H_B] = \alpha I_m$, we also have
$$\mathrm{Tr}(\mathbb{E}[H_B]) = \mathrm{Tr}(\alpha I_m) = \alpha m.$$
Equating the two expressions yields $\alpha m = r$ and hence
$$\mathbb{E}[H_B] = \frac{r}{m} I_m.$$
Because $\mathbb{E}[H_B]$ is a scalar multiple of the identity, all of its eigenvalues are equal to $r/m$, so in particular $\lambda_{\min}(\mathbb{E}[H_B]) = r/m$.

**Step 3: the case of $H_A$.** Now let $A \in \mathbb{R}^{r \times n}$ with i.i.d. $\mathcal{N}(0,1)$ entries and define

$$H_A = A^\top \left(AA^\top\right)^\dagger A \in \mathbb{R}^{n \times n}.$$

Note that $A^\top \in \mathbb{R}^{n \times r}$ also has i.i.d. $\mathcal{N}(0,1)$ entries. Repeating the same argument as above with $A^\top$ in place of $B$ (and with ambient dimension $n$ instead of $m$) gives

$$\mathbb{E}\left[H_A\right] = \frac{r}{n} I_n,$$

and all eigenvalues of $\mathbb{E}\left[H_A\right]$ are equal to $r/n$. This completes the proof. $\qquad\square$

### D.3 I.I.D. UNIFORM INITIALIZATION ON $[-a,a]$

We now consider sketches whose entries are i.i.d. uniform on an interval $[-a,a]$, where $a > 0$. This initialization strategy is employed, for instance, in AsymmLoRA (Zhu et al., 2024). This setting covers both simple uniform initializations and serves as a stepping stone to Kaiming-uniform initialization.

Our analysis relies on three ingredients:

◆ *equivariance* of $H_B$ under left multiplication by an orthogonal matrix,

◆ *equivariance* of $H_A$ under right multiplication by an orthogonal matrix,

◆ *signed-permutation invariance* of the distribution of the sketch matrix.

**Lemma 4** (Equivariance of $H_B$ and $H_A$ under orthogonal transforms). *Let $B \in \mathbb{R}^{m \times r}$ with $\mathrm{rank}(B) = r$ and $A \in \mathbb{R}^{r \times n}$ with $\mathrm{rank}(A) = r$.*

*(i) For any orthogonal matrix $Q \in \mathbb{R}^{m \times m}$, define*

$$H_{QB} := (QB)\left((QB)^\top QB\right)^\dagger (QB)^\top.$$

*Then*

$$H_{QB} = Q\,H_B\,Q^\top, \tag{21}$$

*where $H_B$ is defined in (15).*

*(ii) For any orthogonal matrix $R \in \mathbb{R}^{n \times n}$, define*

$$H_{AR} := (AR)^\top \left((AR)(AR)^\top\right)^\dagger (AR).$$

*Then*

$$H_{AR} = R^\top H_A R, \tag{22}$$

*where $H_A$ is defined in (16).*

*Proof.* We prove the two parts separately.

**(i) Equivariance of $H_B$ under left orthogonal transforms.** Recall that $Q \in \mathbb{R}^{m \times m}$ is orthogonal, so $Q^\top Q = QQ^\top = I_m$. We compute

$$(QB)^\top QB = B^\top Q^\top QB = B^\top B.$$

Hence the inner Gram matrix is unchanged and

$$\left((QB)^\top QB\right)^\dagger = \left(B^\top B\right)^\dagger.$$

Substituting into the definition of $H_{QB}$, we obtain

$$H_{QB} = QB \left(B^\top B\right)^\dagger B^\top Q^\top$$
$$= Q \left(B \left(B^\top B\right)^\dagger B^\top\right) Q^\top$$
$$= Q H_B Q^\top,$$

which proves (21).

**(ii) Equivariance of $H_A$ under right orthogonal transforms.** Now let $R \in \mathbb{R}^{n \times n}$ be orthogonal, so $R^\top R = RR^\top = I_n$. We first compute the Gram matrix for $AR$:

$$(AR)(AR)^\top = ARR^\top A^\top = AA^\top.$$

Thus

$$\left((AR)(AR)^\top\right)^\dagger = \left(AA^\top\right)^\dagger.$$

Using the definition of $H_{AR}$, we have

$$
\begin{aligned}
H_{AR} &= (AR)^{\top} \left( (AR)(AR)^{\top} \right)^{\dagger} (AR) \\
&= R^{\top} A^{\top} \left( A A^{\top} \right)^{\dagger} AR \\
&= R^{\top} \left( A^{\top} \left( A A^{\top} \right)^{\dagger} A \right) R \\
&= R^{\top} H_A R,
\end{aligned}
$$

which establishes (22). This completes the proof. $\qquad\square$

**Lemma 5** (Signed-permutation invariance of i.i.d. uniform sketches for $B$ and $A$). *Let $a > 0$ and consider:*

*(i) A random matrix $B_S \in \mathbb{R}^{m \times r}$ with i.i.d. entries $(B_S)_{ij} \sim \text{Unif}\left([-a,a]\right)$.*

*(ii) A random matrix $A_S \in \mathbb{R}^{r \times n}$ with i.i.d. entries $(A_S)_{ij} \sim \text{Unif}\left([-a,a]\right)$.*

*Let $G_m$ and $G_n$ denote the groups of $m \times m$ and $n \times n$ signed permutation matrices, respectively:*
$$
G_m := \left\{ Q \in \mathbb{R}^{m \times m} : Q = PD, \ P \text{ permutation}, \ D = \text{diag}\left(\pm 1, \ldots, \pm 1\right) \right\},
$$
$$
G_n := \left\{ R \in \mathbb{R}^{n \times n} : R = P'D', \ P' \text{ permutation}, \ D' = \text{diag}\left(\pm 1, \ldots, \pm 1\right) \right\}.
$$
*Then:*

*(i) For any $Q \in G_m$, the random matrix $QB_S$ has the same distribution as $B_S$.*

*(ii) For any $R \in G_n$, the random matrix $A_S R$ has the same distribution as $A_S$.*

*Proof.* We again treat the two cases separately.

**(i) Left signed-permutation invariance for $B_S$.** Write $Q = PD$ with $P$ a permutation matrix and $D = \text{diag}\left(\pm 1, \ldots, \pm 1\right)$. Left-multiplying $B_S$ by $P$ permutes its rows. Since the entries of $B_S$ are i.i.d., each row has the same joint distribution, and permuting rows does not change the joint distribution of the matrix. Thus $PB_S$ has the same distribution as $B_S$.

Next, left-multiplication by $D$ flips the sign of some rows. More precisely, if $D = \text{diag}\left(d_1, \ldots, d_m\right)$ with $d_i \in \{\pm 1\}$, then the $i$-th row of $DB_S$ is $d_i$ times the $i$-th row of $B_S$. For a single scalar random variable $X \sim \text{Unif}\left([-a,a]\right)$, we have
$$
-X \sim \text{Unif}\left([-a,a]\right),
$$
so flipping signs leaves the marginal distribution of each entry unchanged, and independence across entries is preserved (since the sign pattern is deterministic here). Therefore $DB_S$ has the same distribution as $B_S$.

Combining the two transformations, we see that
$$
QB_S = P(DB_S)
$$
is obtained from $B_S$ by a sequence of operations (row permutations and sign flips) that each leave the joint distribution invariant. Hence $QB_S$ has the same distribution as $B_S$ for any $Q \in G_m$.

**(ii) Right signed-permutation invariance for $A_S$.** The argument for $A_S$ is analogous, but now signed permutations act on the *columns* rather than the rows. Let $R \in G_n$ and write $R = P'D'$ with $P'$ a permutation matrix and $D' = \text{diag}\left(\pm 1, \ldots, \pm 1\right)$.

Right-multiplying $A_S$ by $P'$ permutes its columns. Since the entries of $A_S$ are i.i.d., each column has the same joint distribution, and permuting columns preserves the joint distribution of the matrix. Thus $A_S P'$ has the same distribution as $A_S$.

Right-multiplying by $D'$ flips the sign of some columns: if $D' = \text{diag}\left(d_1', \ldots, d_n'\right)$ with $d_j' \in \{\pm 1\}$, then the $j$-th column of $A_S D'$ is $d_j'$ times the $j$-th column of $A_S$. As above, each sign flip preserves the marginal $\text{Unif}\left([-a,a]\right)$ distribution of every entry, and independence across entries is preserved, so $A_S D'$ has the same distribution as $A_S$.

Combining these, we have
$$
A_S R = A_S(P'D') = (A_S P')D',
$$

which is obtained from $A_S$ by a sequence of column permutations and column-wise sign flips, each of which leaves the joint distribution invariant. Therefore $A_S R$ has the same distribution as $A_S$ for any $R \in G_n$.

This proves both claims. □

Combining Lemmas 4 and 5, we can derive the expected projections in closed form.

**Lemma 6** (Expected projections for uniform sketches). *Let $r \leq \min\{m,n\}$ and consider two random matrices:*

◆ $B_S \in \mathbb{R}^{m \times r}$ *with entries i.i.d.* $\mathrm{Unif}\left([-a,a]\right)$,

◆ $A_S \in \mathbb{R}^{r \times n}$ *with entries i.i.d.* $\mathrm{Unif}\left([-a,a]\right)$.

*Define*
$$H_B := B_S \left(B_S^\top B_S\right)^\dagger B_S^\top \in \mathbb{R}^{m \times m},$$
$$H_A := A_S^\top \left(A_S A_S^\top\right)^\dagger A_S \in \mathbb{R}^{n \times n}.$$
*Assume $B_S$ and $A_S$ have full rank $r$ almost surely. Then*
$$\mathbb{E}\left[H_B\right] = \frac{r}{m} I_m, \qquad \mathbb{E}\left[H_A\right] = \frac{r}{n} I_n,$$
*and hence*
$$\lambda_{\min}\left(\mathbb{E}\left[H_B\right]\right) = \frac{r}{m}, \qquad \lambda_{\min}\left(\mathbb{E}\left[H_A\right]\right) = \frac{r}{n}.$$

*Proof.* We first treat $H_B$. For any $Q \in G_m$, Lemma 5(i) gives $Q B_S \overset{d}{=} B_S$, and Lemma 4(i) gives $H_{Q B_S} = Q H_B Q^\top$. Since $Q B_S$ and $B_S$ have the same distribution, we obtain
$$\mathbb{E}\left[H_B\right] = \mathbb{E}\left[H_{Q B_S}\right] = \mathbb{E}\left[Q H_B Q^\top\right] = Q \mathbb{E}\left[H_B\right] Q^\top \quad \text{for all } Q \in G_m.$$
Thus $\mathbb{E}\left[H_B\right]$ commutes with every signed permutation matrix $Q \in G_m$. By Lemma 2, there exists $\alpha \in \mathbb{R}$ such that
$$\mathbb{E}\left[H_B\right] = \alpha I_m.$$
To determine $\alpha$, note that for any realization with $\mathrm{rank}(B_S) = r$, $H_B$ is an orthogonal projector of rank $r$, so $\mathrm{Tr}\left(H_B\right) = r$. Taking expectations and using linearity of the trace,
$$\mathrm{Tr}\left(\mathbb{E}\left[H_B\right]\right) = \mathbb{E}\left[\mathrm{Tr}\left(H_B\right)\right] = r.$$
On the other hand,
$$\mathrm{Tr}\left(\mathbb{E}\left[H_B\right]\right) = \mathrm{Tr}\left(\alpha I_m\right) = \alpha m,$$
so $\alpha m = r$ and hence
$$\mathbb{E}\left[H_B\right] = \frac{r}{m} I_m.$$

The argument for $H_A$ is analogous, now working in ambient dimension $n$. Specifically, $A_S^\top \in \mathbb{R}^{n \times r}$ has i.i.d. $\mathrm{Unif}\left([-a,a]\right)$ entries. For any $R \in G_n$, Lemma 5(ii) gives $A_S R \overset{d}{=} A_S$, and Lemma 4(ii) yields $H_{AR} = R^\top H_A R$. Therefore
$$\mathbb{E}\left[H_A\right] = \mathbb{E}\left[H_{AR}\right] = \mathbb{E}\left[R^\top H_A R\right] = R^\top \mathbb{E}\left[H_A\right] R \quad \text{for all } R \in G_n.$$
By Lemma 2 applied in $\mathbb{R}^{n \times n}$, we must have $\mathbb{E}\left[H_A\right] = \beta I_n$ for some $\beta \in \mathbb{R}$. As before, $\mathrm{rank}(A_S) = r$ almost surely, so $H_A$ is a rank-$r$ projector and $\mathrm{Tr}\left(H_A\right) = r$ almost surely, implying
$$\mathrm{Tr}\left(\mathbb{E}\left[H_A\right]\right) = \mathbb{E}\left[\mathrm{Tr}\left(H_A\right)\right] = r.$$
On the other hand, $\mathrm{Tr}\left(\mathbb{E}\left[H_A\right]\right) = \mathrm{Tr}\left(\beta I_n\right) = \beta n$, so $\beta = r/n$ and hence
$$\mathbb{E}\left[H_A\right] = \frac{r}{n} I_n.$$
This completes the proof. □

### D.4 KAIMING-UNIFORM INITIALIZATION

In this section, we consider the widely used Kaiming-uniform initializer, implemented in PyTorch as `nn.init.kaiming_uniform_`. Kaiming-uniform (He) initialization (**?**) underlies the default linear-layer initialization in PyTorch and is therefore inherited by many practical LoRA implementations that keep the framework defaults for adapter weights (e.g. Hayou et al., 2024; **?**; Kopiczko

et al., 2023). This initializer samples each entry of a weight matrix independently from a symmetric uniform distribution on an interval $[-b,b]$, where the bound $b > 0$ depends on the fan-in and the activation function. In particular, the entries are i.i.d., continuous, and symmetric about zero.

Let $B_S \in \mathbb{R}^{m \times r}$ and $A_S \in \mathbb{R}^{r \times n}$ be initialized with Kaiming-uniform. Then $B_S$ and $A_S$ satisfy exactly the same symmetry properties as in the uniform $[-a,a]$ case:

◆ The entries are i.i.d. and symmetric around zero, so the distribution is invariant under row permutations and sign flips (i.e. under $G_m$ or $G_n$).

◆ With probability one, $\mathrm{rank}(B_S) = r$ and $\mathrm{rank}(A_S) = r$ (since the entries are drawn from a continuous distribution).

Therefore the proof of Lemma 6 applies verbatim.

**Lemma 7** (Expected projections for Kaiming-uniform sketches). *Let $r \leq \min\{m,n\}$ and consider two random matrices:*

◆ $B_S \in \mathbb{R}^{m \times r}$ *with entries initialized by Kaiming-uniform,*

◆ $A_S \in \mathbb{R}^{r \times n}$ *with entries initialized by Kaiming-uniform.*

*Define $H_B$ and $H_A$ as in* (15) *and* (16)*. Then*

$$\mathbb{E}\left[H_B\right] = \frac{r}{m} I_m, \qquad \mathbb{E}\left[H_A\right] = \frac{r}{n} I_n,$$

*and hence*

$$\lambda_{\min}\left(\mathbb{E}\left[H_B\right]\right) = \frac{r}{m}, \qquad \lambda_{\min}\left(\mathbb{E}\left[H_A\right]\right) = \frac{r}{n}.$$

*Proof.* Because Kaiming-uniform draws each entry independently from a symmetric uniform distribution $[-b,b]$, the distribution of $B_S$ is invariant under any signed permutation of rows: permuting rows leaves the joint law unchanged, and multiplying any row by $-1$ preserves the marginal law of each entry (by symmetry). Thus $QB_S \overset{d}{=} B_S$ for all $Q \in G_m$, and the same holds for $A_S^\top$ with $G_n$.

The rest of the argument is exactly as in Lemma 6: by combining Lemma **??** with signed-permutation invariance, we conclude that $\mathbb{E}[H_B] = \alpha I_m$ and $\mathbb{E}[H_A] = \beta I_n$ for some scalars $\alpha, \beta \in \mathbb{R}$. Since $H_B$ and $H_A$ are rank-$r$ projectors almost surely, $\mathrm{Tr}(H_B) = r$ and $\mathrm{Tr}(H_A) = r$, and the trace identities

$$\mathrm{Tr}\left(\mathbb{E}\left[H_B\right]\right) = \alpha m = r, \qquad \mathrm{Tr}\left(\mathbb{E}\left[H_A\right]\right) = \beta n = r$$

imply $\alpha = r/m$ and $\beta = r/n$. This yields the stated formulas. □

In summary, for Gaussian, i.i.d. uniform, and Kaiming-uniform sketch distributions, the expected projection matrices are isotropic:

$$\mathbb{E}\left[H_B\right] = \frac{r}{m} I_m, \qquad \mathbb{E}\left[H_A\right] = \frac{r}{n} I_n,$$

and Assumption 1 holds with $\lambda_{\min}^H = \min\{r/m, r/n\} > 0$. This shows that the positive expected projection condition is naturally satisfied by a broad class of standard initialization schemes used in LoRA and its variants.

### D.5 RANDOM ORTHONORMAL SKETCHES VIA SVD

We now consider the initialization where a dense random matrix $W \in \mathbb{R}^{m \times n}$ with i.i.d. entries $W_{ij} \sim \mathrm{Unif}\left([-a,a]\right)$ is first sampled, and then orthonormal sketches are obtained from its singular vectors. Specifically, let $W = U\Sigma V^\top$ be an SVD with singular values arranged in strictly decreasing order, and set

$$B_S(W) := U_{[:,1:r]} \in \mathbb{R}^{m \times r},$$

$$A_S(W) := V_{[:,1:r]}^\top \in \mathbb{R}^{r \times n}.$$

By construction,

$$B_S(W)^\top B_S(W) = I_r, \qquad A_S(W) A_S(W)^\top = I_r.$$

In particular, when we plug $B_S(W)$ and $A_S(W)$ into the general projector definitions

$$H_B := B_S(W) \left( B_S(W)^\top B_S(W) \right)^\dagger B_S(W)^\top \in \mathbb{R}^{m \times m},$$

$$H_A := A_S(W)^\top \left( A_S(W) A_S(W)^\top \right)^\dagger A_S(W) \in \mathbb{R}^{n \times n},$$

the pseudo-inverse is simply the identity (because $B_S(W)^\top B_S(W) = A_S(W) A_S(W)^\top = I_r$), so

$$H_B(W) = B_S(W) B_S(W)^\top = U_{[:,1:r]} U_{[:,1:r]}^\top \in \mathbb{R}^{m \times m},$$

$$H_A(W) = A_S(W)^\top A_S(W) = V_{[:,1:r]} V_{[:,1:r]}^\top \in \mathbb{R}^{n \times n}.$$

Both $H_B(W)$ and $H_A(W)$ are orthogonal projectors of rank $r$, with eigenvalues $\{1\}$ on the chosen $r$-dimensional subspace and $\{0\}$ on its orthogonal complement.

This type of initialization (taking $U_{[:,1:r]}$ or $V_{[:,1:r]}$ from the SVD of a dense random matrix) appears, for example, in the experimental studies by Zhu et al. (2024), and is closely related to the orthonormal constructions used in OLoRA (**?**).

Our first goal is to understand how the sketch projectors $H_B(W)$ and $H_A(W)$ transform when we apply signed permutations to the rows or columns of $W$.

**Lemma 8** (Equivariance of SVD-based left and right sketches under signed permutations)**.** *Let $W \in \mathbb{R}^{m \times n}$ be any matrix with SVD $W = U \Sigma V^\top$, where $\Sigma = \mathrm{diag}(\sigma_1, \ldots, \sigma_d)$ with strictly decreasing singular values $\sigma_1 > \cdots > \sigma_d > 0$ (here $d = \mathrm{rank}(W)$). Define*

$$B_S(W) := U_{[:,1:r]} \in \mathbb{R}^{m \times r}, \qquad H_B(W) := B_S(W) B_S(W)^\top,$$

$$A_S(W) := V_{[:,1:r]}^\top \in \mathbb{R}^{r \times n}, \qquad H_A(W) := A_S(W)^\top A_S(W).$$

*Then:*

*(i) For any signed permutation $Q \in G_m$, consider an SVD of $QW$ with the singular values ordered in the same descending fashion. Up to column-wise sign flips, the left singular vectors of $QW$ are $QU$, and the corresponding left-sketch projector satisfies*

$$H_B(QW) = Q H_B(W) Q^\top. \tag{23}$$

*(ii) For any signed permutation $R \in G_n$, consider an SVD of $WR$ with the singular values ordered in the same descending fashion. Up to column-wise sign flips, the right singular vectors of $WR$ are $R^\top V$, and the corresponding right-sketch projector satisfies*

$$H_A(WR) = R^\top H_A(W) R. \tag{24}$$

*Proof.* We prove (i) and (ii) separately.

**(i) Left sketches: effect of $Q \in G_m$ acting on rows.** Since $Q \in G_m$ is orthogonal, $QW$ admits the factorization

$$QW = (QU) \Sigma V^\top,$$

where $QU$ is also orthogonal. The singular values of $QW$ are the same as those of $W$, namely $\sigma_1, \ldots, \sigma_d$, and by assumption they are strictly ordered: $\sigma_1 > \cdots > \sigma_d > 0$.

Consider an SVD of $QW$ with singular values written in descending order:

$$QW = U' \Sigma V'^\top,$$

where $U'$ and $V'$ are orthogonal. The uniqueness properties of the SVD when all singular values are distinct imply that $U'$ and $V'$ are determined by $QU$ and $V$ up to sign flips of individual singular vectors. More precisely, there exists a diagonal orthogonal matrix $R = \mathrm{diag}(\pm 1, \ldots, \pm 1) \in \mathbb{R}^{d \times d}$ such that

$$U' = QUR, \qquad V' = VR.$$

(If some singular values were repeated, $R$ could mix singular vectors within blocks corresponding to equal singular values; the strict-ordering assumption rules this out.)

Let $R_{1:r}$ denote the leading $r \times r$ diagonal block of $R$. Then the first $r$ left singular vectors of $QW$ can be written as

$$B_S(QW) = U'_{[:,1:r]} = QU_{[:,1:r]} R_{1:r}.$$

The corresponding projector is

$$H_B(QW) = B_S(QW)B_S(QW)^\top$$

$$= \left(QU_{[:,1:r]}R_{1:r}\right)\left(QU_{[:,1:r]}R_{1:r}\right)^\top$$

$$= QU_{[:,1:r]}R_{1:r}R_{1:r}^\top U_{[:,1:r]}^\top Q^\top$$

$$= QU_{[:,1:r]}U_{[:,1:r]}^\top Q^\top$$

$$= QH_B(W)Q^\top,$$

since $R_{1:r}R_{1:r}^\top = I_r$. This proves (23).

**(ii) Right sketches: effect of $R \in G_n$ acting on columns.** Now consider $WR$ with $R \in G_n$ orthogonal. Using the SVD of $W$, we have

$$WR = U\Sigma V^\top R = U\Sigma\left(R^\top V\right)^\top.$$

Since $R^\top V$ is orthogonal, this is an SVD of $WR$ with left singular matrix $U$ and right singular matrix $\widetilde{V} := R^\top V$. The singular values remain $\sigma_1, \ldots, \sigma_d$, strictly ordered.

Let

$$WR = \widetilde{U}\Sigma\widetilde{V}^\top$$

be any SVD of $WR$ with singular values in descending order. By the same uniqueness argument, there exists a diagonal orthogonal matrix $S = \mathrm{diag}\left(\pm 1, \ldots, \pm 1\right) \in \mathbb{R}^{d \times d}$ such that

$$\widetilde{U} = US, \qquad \widetilde{V} = \widetilde{V}_0 S = (R^\top V)S.$$

Let $V_r = V_{[:,1:r]}$ and $S_{1:r}$ be the leading $r \times r$ block of $S$. Then the first $r$ right singular vectors of $WR$ are given by the first $r$ columns of $\widetilde{V}$:

$$\widetilde{V}_{[:,1:r]} = (R^\top VS)_{[:,1:r]}$$

$$= R^\top V_{[:,1:r]}S_{1:r}.$$

Recalling that $A_S(W) = V_r^\top$, the right-sketch matrix for $WR$ is

$$A_S(WR) = \widetilde{V}_{[:,1:r]}^\top$$

$$= S_{1:r}^\top V_{[:,1:r]}^\top R$$

$$= S_{1:r}V_r^\top R,$$

where we used that $S_{1:r}$ is diagonal with entries $\pm 1$, so $S_{1:r}^\top = S_{1:r}$.

The corresponding right-sketch projector is

$$H_A(WR) = A_S(WR)^\top A_S(WR)$$

$$= \left(S_{1:r}V_r^\top R\right)^\top \left(S_{1:r}V_r^\top R\right)$$

$$= R^\top V_r S_{1:r}^\top S_{1:r}V_r^\top R$$

$$= R^\top V_r V_r^\top R$$

$$= R^\top H_A(W)R,$$

since $S_{1:r}^\top S_{1:r} = I_r$. This proves (24) and completes the proof. $\qquad\square$

We now combine this equivariance with the signed-permutation invariance of the i.i.d. uniform matrix $W$ to obtain closed-form expressions for the expected projectors.

**Lemma 9** (Expected projections for SVD-based uniform orthonormal sketches). *Let $W \in \mathbb{R}^{m \times n}$ have i.i.d. entries $W_{ij} \sim \mathrm{Unif}\left([-a,a]\right)$, and let $H_B(W)$ and $H_A(W)$ be defined as above from an SVD $W = U\Sigma V^\top$ with strictly decreasing singular values. Then*

$$\mathbb{E}\left[H_B(W)\right] = \frac{r}{m}I_m, \qquad \mathbb{E}\left[H_A(W)\right] = \frac{r}{n}I_n,$$

*and hence*

$$\lambda_{\min}\left(\mathbb{E}\left[H_B(W)\right]\right) = \frac{r}{m} > 0, \qquad \lambda_{\min}\left(\mathbb{E}\left[H_A(W)\right]\right) = \frac{r}{n} > 0.$$

*Proof.* We treat $H_B(W)$ and $H_A(W)$ in turn.

**Left-sketch projector $H_B(W)$.** The rows of $W$ are i.i.d. vectors in $\mathbb{R}^n$ with continuous, symmetric entries. For any signed permutation $Q \in G_m$, left-multiplication by $Q$ permutes and flips the signs of rows, so

$$QW \stackrel{d}{=} W \quad \text{for all } Q \in G_m.$$

By Lemma 8(i),

$$H_B(QW) = QH_B(W)Q^\top.$$

Since $QW$ and $W$ are identically distributed, $H_B(QW)$ and $H_B(W)$ are identically distributed, and hence

$$\mathbb{E}\left[H_B(W)\right] = \mathbb{E}\left[H_B(QW)\right] = \mathbb{E}\left[QH_B(W)Q^\top\right] = Q\,\mathbb{E}\left[H_B(W)\right]Q^\top \quad \text{for all } Q \in G_m.$$

Thus $\mathbb{E}\left[H_B(W)\right]$ commutes with every signed permutation matrix in $G_m$, and by Lemma 2 there exists $\alpha \in \mathbb{R}$ such that

$$\mathbb{E}\left[H_B(W)\right] = \alpha I_m.$$

To determine $\alpha$, recall that $B_S(W)$ has orthonormal columns, so $H_B(W) = B_S(W)B_S(W)^\top$ is a rank-$r$ projector with

$$\mathrm{Tr}\left(H_B(W)\right) = r$$

for every realization. Taking expectations and using linearity of the trace,

$$\mathrm{Tr}\left(\mathbb{E}\left[H_B(W)\right]\right) = \mathbb{E}\left[\mathrm{Tr}\left(H_B(W)\right)\right] = r.$$

On the other hand,

$$\mathrm{Tr}\left(\mathbb{E}\left[H_B(W)\right]\right) = \mathrm{Tr}\left(\alpha I_m\right) = \alpha m,$$

so $\alpha m = r$ and hence $\alpha = r/m$. Therefore

$$\mathbb{E}\left[H_B(W)\right] = \frac{r}{m}\,I_m.$$

**Right-sketch projector $H_A(W)$.** The columns of $W$ are also i.i.d. vectors in $\mathbb{R}^m$ with continuous, symmetric entries. For any signed permutation $R \in G_n$, right-multiplication by $R$ permutes and flips the signs of columns, so

$$WR \stackrel{d}{=} W \quad \text{for all } R \in G_n.$$

By Lemma 8(ii),

$$H_A(WR) = R^\top H_A(W)R.$$

Since $WR$ and $W$ have the same distribution, the random matrices $H_A(WR)$ and $H_A(W)$ are identically distributed. Hence

$$\mathbb{E}\left[H_A(W)\right] = \mathbb{E}\left[H_A(WR)\right] = \mathbb{E}\left[R^\top H_A(W)R\right] = R^\top \mathbb{E}\left[H_A(W)\right]R \quad \text{for all } R \in G_n.$$

Applying Lemma 2 (now in dimension $n$) shows that there exists $\beta \in \mathbb{R}$ such that

$$\mathbb{E}\left[H_A(W)\right] = \beta I_n.$$

Again, $A_S(W)$ has orthonormal rows, so $H_A(W) = A_S(W)^\top A_S(W)$ is a rank-$r$ projector and

$$\mathrm{Tr}\left(H_A(W)\right) = r$$

for every realization. Taking expectations,

$$\mathrm{Tr}\left(\mathbb{E}\left[H_A(W)\right]\right) = \mathbb{E}\left[\mathrm{Tr}\left(H_A(W)\right)\right] = r.$$

But $\mathrm{Tr}\left(\mathbb{E}\left[H_A(W)\right]\right) = \mathrm{Tr}\left(\beta I_n\right) = \beta n$, hence $\beta n = r$ and therefore $\beta = r/n$. Thus

$$\mathbb{E}\left[H_A(W)\right] = \frac{r}{n}\,I_n.$$

This completes the proof. $\square$

**Remark 2.** *Each individual projector $H_B(W)$ (resp. $H_A(W)$) is rank-deficient, with eigenvalues $\{1\}$ on an $r$-dimensional subspace and $\{0\}$ on its orthogonal complement. The lemma above concerns the* expectation *of these projectors over the randomness of $W$. Because the subspace spanned by the leading singular vectors is random and, in distribution, symmetric under signed permutations, the expectation $\mathbb{E}\left[H_B(W)\right]$ (resp. $\mathbb{E}\left[H_A(W)\right]$) becomes a full-rank, isotropic matrix $(r/m)I_m$ (resp. $(r/n)I_n$). This is exactly analogous to the classical fact that if $u$ is a random unit vector in $\mathbb{R}^d$, then $\mathbb{E}\left[uu^\top\right] = (1/d)I_d$ even though $uu^\top$ has rank 1 for every realization.*

# E  REFORMULATION AS A PROJECTED GRADIENT STEP

Following the approach of Malinovsky et al. (2024), let's consider the update for the trainable matrix $\hat{A}^t$ in the Left Sketch case. Taking a single GD step on the subproblem corresponds to minimizing a quadratic approximation of the objective. This yields the solution for $\hat{A}^t$:

$$\hat{A}^t = -\eta \left( \left( B_S^t \right)^\top B_S^t \right)^\dagger \left( B_S^t \right)^\top \nabla f(W^t),$$

where $\eta$ is a learning rate for the subproblem and $\dagger$ denotes the Moore-Penrose pseudoinverse. Substituting this into the update for $W^{t+1}$ gives:

$$W^{t+1} = W^t + \frac{\alpha}{r} B_S^t \hat{A}^t = W^t - \frac{\alpha\eta}{r} B_S^t \left( \left( B_S^t \right)^\top B_S^t \right)^\dagger \left( B_S^t \right)^\top \nabla f(W^t)$$

$$= W^t - \gamma H_B^t \nabla f(W^t),$$

where we define the effective stepsize $\gamma := \frac{\alpha\eta}{r}$ and the projection matrix $H_B^t :=$ $B_S^t \left( \left( B_S^t \right)^\top B_S^t \right)^\dagger \left( B_S^t \right)^\top$. A similar derivation for the Right Sketch case gives the update:

$$W^{t+1} = W^t - \gamma \nabla f(W^t) H_A^t,$$

where $H_A^t := \left( A_S^t \right)^\top \left( A_S^t \left( A_S^t \right)^\top \right)^\dagger A_S^t$. This reformulation reveals that both Left and Right sketch updates are equivalent to applying a standard gradient-based update, but projected onto a randomly chosen low-rank subspace.

# F    Core Algorithmic Variants

**Bernoulli-LoRA-GD.**    The simplest instantiation of our framework is Bernoulli-LoRA-GD (Algorithm 2). This method serves as a foundational building block and a starting point for more elaborate variants. It uses the full gradient of the objective function as its base estimator, i.e., $G^t = \nabla f(W^t)$. While impractical for large-scale deep learning, its analysis provides crucial insights into the convergence behavior of the Bernoulli-LoRA mechanism under idealized, deterministic conditions.

**Bernoulli-LoRA-SGD.**    Stochastic Gradient Descent (SGD) (Robbins & Monro, 1951) is a highly effective and widely utilized algorithm for training a variety of machine learning models. The latest advancements in deep learning training methods are all based on different variations of SGD (Sun, 2020). Its advantage over GD is that it uses stochastic gradients for updates, rather than relying on full gradients. Within our framework, we develop Bernoulli-LoRA-SGD, where the base estimator $G^t$ is a general unbiased stochastic gradient of $f$ at $W^t$.

**Bernoulli-LoRA-PAGE.**    Several optimal algorithms exist for addressing non-convex optimization problems, such as SPIDER (Fang et al., 2018) and SARAH (Pham et al., 2020). However, their optimality is supported by a known lower bound that applies only in the small data setting. In contrast, ProbAbilistic Gradient Estimator (PAGE) (Li et al., 2021) stands out for its simplicity, ease of implementation, and ability to achieve optimal convergence in non-convex optimization. PAGE alternates between a full gradient update with probability $q_t$ and a low-cost gradient adjustment with probability $1 - q_t$. Bernoulli-LoRA-PAGE is a new method based on PAGE within our Bernoulli-LoRA framework.

**Bernoulli-LoRA-MVR.**    VR methods outperform SGD in reaching first-order critical points but often require finely tuned learning rates and large batch sizes to be effective. To overcome these challenges, Momentum Variance Reduction (MVR) (Cutkosky & Orabona, 2019) was introduced for server-only stochastic non-convex optimization. MVR uses a modified momentum technique to reduce variance without relying on large batch sizes. Several works employ this powerful approach (Tyurin & Richtárik, 2023; Karagulyan et al., 2024). We propose Bernoulli-LoRA-MVR, where the base estimator $G^t$ is updated using the MVR rule: a combination of the current stochastic gradient and a momentum term that incorporates the difference between past estimators and gradients.

# G    Extensions for Federated Learning

Sun et al. (2024) identified instability in LoRA, arising from the mismatch between local clients simultaneously optimizing two low-rank matrices and the central server aggregating them independently. Factors such as data heterogeneity, multi-step local updates, and the amplification of additive noise applied to gradients for ensuring differential privacy (DP) significantly impact the process. Additionally, the final performance is highly sensitive to hyperparameter choices. Their proposed solution centers on keeping the randomly initialized non-zero matrices fixed while exclusively fine-tuning the zero-initialized ones. Based on this asymmetric approach, Malinovsky et al. (2024) proposed a distributed method Fed-RAC-LoRA.

We develop the theory further by incorporating compression, VR and EF techniques into FL methods for LoRA within the novel Bernoulli-LoRA framework.

The effectiveness of a distributed training method is primarily measured by its communication complexity, defined as the product of the required communication rounds and the communication volume per round. Following common practice, we assume client-to-server communication is the main bottleneck and exclude server-to-client communication from our analysis.

**Fed-Bernoulli-LoRA-QGD.**    A key challenge for distributed methods lies in the high communication cost of gradient updates. Lossy compression techniques, such as QSGD (Alistarh et al., 2017), address this by enabling clients to send quantized gradients. We design Fed-Bernoulli-LoRA-QGD based on QSGD. The clients send compressed versions of their gradients. The base estimator $G^t$ is formed by averaging the compressed local gradients received from all clients.

**Fed-Bernoulli-LoRA-MARINA.** MARINA (Gorbunov et al., 2021) is a communication-efficient method for non-convex distributed learning on heterogeneous datasets that uses a novel gradient difference compression strategy. Its biased gradient estimator underpins its strong theoretical and practical performance, with proven communication complexity bounds surpassing all prior first-order methods. We propose Fed-Bernoulli-LoRA-MARINA, where each client's local estimator $G_l^t$ is updated either with a full local gradient (with probability $q$) or by adding a compressed gradient difference to its previous estimator. The server's base estimator $G^t$ is the average of these local estimators.

**Fed-Bernoulli-LoRA-EF21.** Error Feedback (EF) (Seide et al., 2014; Stich et al., 2018; Alistarh et al., 2018; Richtárik et al., 2021) is a widely adopted technique for stabilizing training with contractive compressors. We propose Fed-Bernoulli-LoRA-EF21, based on the modern EF21. Here, each client updates its local estimator $G_l^t$ by adding a compressed version of the difference between the current local gradient and the previous local estimator. The server's base estimator $G^t$ is again the average of the clients' estimators.

# H PROOFS FOR CORE ALGORITHMIC VARIANTS

## H.1 ANALYSIS OF BERNOULLI-LORA-GD

---

**Algorithm 2** Bernoulli-LoRA-GD

---

1: **Parameters:** pre-trained model $W^0 \in \mathbb{R}^{m \times n}$, rank $r \ll \min\{m,n\}$, scaling factor $\alpha > 0$, stepsize $\gamma_t$ chain length $T$, sketch distribution $\mathcal{D}_S^B$ or $\mathcal{D}_S^A$, Bernoulli probability $p$
2: **for** $t = 0, 1, \ldots, T-1$ **do**
3:    Sample $c^t \sim \mathrm{Be}(p)$               Bernoulli random variable
4:    **if** $c^t = 1$ **then**
5:      Sample $B_S^t \sim \mathcal{D}_S^B$                 Left sketch
6:      $\hat{A}^t = -\eta \left( (B_S^t)^\top B_S^t \right)^\dagger (B_S^t)^\top \nabla f(W^t)$
7:      $W^{t+1} = W^t + \frac{\alpha}{r} B_S^t \hat{A}^t$
8:    **else**
9:      Sample $A_S^t \sim \mathcal{D}_S^A$                 Right sketch
10:     $\hat{B}^t = -\eta \nabla f(W^t) (A_S^t)^\top \left( A_S^t (A_S^t)^\top \right)^\dagger$
11:     $W^{t+1} = W^t + \frac{\alpha}{r} \hat{B}^t A_S^t$
12:    **end if**
13: **end for**

---

The following lemma establishes that the Bernoulli-LoRA update can be reformulated as a standard projected gradient descent step, providing a crucial foundation for our subsequent convergence analysis.

**Lemma 10.** *Consider the updates $\hat{A}^t$ and $\hat{B}^t$ from Algorithm 2 computed as solutions to the following optimization problems:*

$$\hat{A}^t \quad := \quad \arg\min_A \left\{ f(W^t) + \frac{\alpha}{r} \left\langle \nabla f(W^t), B_S^t A \right\rangle_\mathrm{F} + \frac{\alpha^2}{2\gamma r^2} \left\| B_S^t A \right\|_\mathrm{F}^2 \right\},$$

$$\hat{B}^t \quad := \quad \arg\min_B \left\{ f(W^t) + \frac{\alpha}{r} \left\langle \nabla f(W^t), B A_S^t \right\rangle_\mathrm{F} + \frac{\alpha^2}{2\gamma r^2} \left\| B A_S^t \right\|_\mathrm{F}^2 \right\}. \tag{25}$$

*Then the Left and Right sketch updates can be expressed as a gradient descent step:*
$$W^{t+1} = W^t - \gamma G^t, \tag{26}$$
*where $G^t$ is defined by*

$$G^t = \begin{cases} H_B^t \nabla f(W^t), & \text{with probability } p \\ \nabla f(W^t) H_A^t, & \text{with probability } 1-p \end{cases} \tag{27}$$

*with projection matrices $H_A^t$ and $H_B^t$ given by:*

$$H_A^t := \left( A_S^t \right)^\top \left( A_S^t \left( A_S^t \right)^\top \right)^\dagger A_S^t \quad \text{and} \quad H_B^t := B_S^t \left( \left( B_S^t \right)^\top B_S^t \right)^\dagger \left( B_S^t \right)^\top, \tag{28}$$

*where † denotes the Moore-Penrose pseudoinverse.*

*Proof.* Following Algorithm 2, at each iteration we randomly select either the Left sketch (with probability $p$) or the Right sketch (with probability $1-p$). We analyze both cases separately and then combine them into a unified update rule.

**Left Sketch Analysis.** When the Left sketch is selected, the update takes the form:

$$W^{t+1} = W^t + \frac{\alpha}{r} B_S^t \hat{A}^t. \tag{29}$$

Minimizing the right-hand side with respect to $\hat{A}^t$ yields:

$$\frac{\alpha}{r}\left(B_S^t\right)^\top \nabla f(W^t) + \frac{\alpha^2}{\gamma r^2}\left(B_S^t\right)^\top B_S^t \hat{A}^t = 0;$$

$$\left(B_S^t\right)^\top B_S^t \hat{A}^t = -\frac{\gamma r}{\alpha}\left(B_S^t\right)^\top \nabla f(W^t);$$

$$\hat{A}^t = -\frac{\gamma r}{\alpha}\left(\left(B_S^t\right)^\top B_S^t\right)^\dagger \left(B_S^t\right)^\top \nabla f(W^t). \quad (30)$$

This leads to the Left sketch update:

$$W^{t+1} = W^t + \frac{\alpha}{r}B_S^t \hat{A}^t$$

$$= W^t - \gamma B_S^t \left(\left(B_S^t\right)^\top B_S^t\right)^\dagger \left(B_S^t\right)^\top \nabla f(W^t)$$

$$= W^t - \gamma H_B^t \nabla f(W^t), \quad (31)$$

where $H_B^t := B_S^t \left(\left(B_S^t\right)^\top B_S^t\right)^\dagger \left(B_S^t\right)^\top$ is a projection matrix.

**Right Sketch Analysis.** For the Right sketch, we follow a similar approach. The update rule is:

$$W^{t+1} = W^t + \frac{\alpha}{r}\hat{B}^t A_S^t. \quad (32)$$

First, observe that:

$$\left\|\hat{B}^t A_S^t\right\|_F^2 = \left\langle \hat{B}^t A_S^t, \hat{B}^t A_S^t\right\rangle_F = \left\langle A_S^t, \left(\hat{B}^t\right)^\top \hat{B}^t A_S^t\right\rangle_F. \quad (33)$$

For the linear term from (25):

$$\frac{\alpha}{r}\left\langle \nabla f(W^t), \hat{B}^t A_S^t\right\rangle_F = \frac{\alpha}{r}\text{Tr}\left(\left(\nabla f(W^t)\right)^\top \hat{B}^t A_S^t\right), \quad (34)$$

with gradient $\nabla f(W^t)\left(A_S^t\right)^\top$ with respect to $\hat{B}^t$. Using the matrix calculus identity $\nabla_X \|X\|_F^2 = 2X$, the gradient of the quadratic term is:

$$\frac{\alpha^2}{\gamma r^2}\hat{B}^t A_S^t \left(A_S^t\right)^\top. \quad (35)$$

Setting the total gradient to zero and solving for $\hat{B}^t$:

$$\hat{B}^t = -\frac{\gamma r}{\alpha}\nabla f(W^t)\left(A_S^t\right)^\top \left(A_S^t \left(A_S^t\right)^\top\right)^\dagger, \quad (36)$$

which yields the Right sketch update:

$$W^{t+1} = W^t + \frac{\alpha}{r}\hat{B}^t A_S^t$$

$$= W^t - \gamma \nabla f(W^t)\left(A_S^t\right)^\top \left(A_S^t \left(A_S^t\right)^\top\right)^\dagger A_S^t$$

$$= W^t - \gamma \nabla f(W^t)H_A^t, \quad (37)$$

where $H_A^t := \left(A_S^t\right)^\top \left(A_S^t \left(A_S^t\right)^\top\right)^\dagger A_S^t$ is a projection matrix.

**Combined Update Rule.** Combining equations (31) and (37), we obtain the unified update:

$$W^{t+1} = W^t - \gamma G^t, \quad (38)$$

where $G^t$ takes the form given in the lemma statement, completing the proof. $\square$

With these assumptions in place, we can now state our main convergence result for RAC-LoRA with Gradient Descent updates.

### H.1.1 Convergence for Smooth Non-Convex Functions

**Theorem** 1. *Let Assumptions 1, 3, and 2 hold, and let the stepsize satisfy $0 < \gamma \leq \frac{1}{L}$. Then the iterates of* Bernoulli-LoRA-GD *(Algorithm 2), with matrices $\hat{A}^t$ and $\hat{B}^t$ computed according to Lemma 10, satisfy*

$$\mathbb{E}\left[\left\|\nabla f(\widetilde{W}^T)\right\|_F^2\right] \leq \frac{2(f(W^0) - f^*)}{\gamma \lambda_{\min}^p T}, \quad (39)$$

where $\lambda_{\min}^p := p\lambda_{\min}^{H_B} + (1-p)\lambda_{\min}^{H_A}$ and $\widetilde{W}^T$ is drawn uniformly at random from the iterate sequence $\{W^0, W^1, \ldots, W^{T-1}\}$.

*Proof.* From Lemma 10, we know that Bernoulli-LoRA updates can be expressed as

$$W^{t+1} = W^t - \gamma G^t, \tag{40}$$

where $G^t$ takes the form

$$G^t = \begin{cases} H_B^t \nabla f(W^t), & \text{with probability } p \\ \nabla f(W^t) H_A^t, & \text{with probability } 1-p \end{cases} \tag{41}$$

with projection matrices $H_A^t$ and $H_B^t$ as defined in the lemma.

To analyze the convergence, we first compute the conditional expectation and second moment of $G^t$:

$$\mathbb{E}\left[G^t \mid W^t, H^t\right] = pH_B^t \nabla f(W^t) + (1-p)\nabla f(W^t) H_A^t,$$

$$\mathbb{E}\left[\|G^t\|_F^2 \mid W^t, H^t\right] = p\|H_B^t \nabla f(W^t)\|_F^2 + (1-p)\|\nabla f(W^t) H_A^t\|_F^2, \tag{42}$$

where we defined $H^t := \{H_A^t, H_B^t\}$.

We begin by establishing several key auxiliary bounds. For the Left sketch term:

$$-\gamma p \left\langle \nabla f(W^t), H_B^t \nabla f(W^t)\right\rangle_F + \frac{L\gamma^2}{2} p \left\|H_B^t \nabla f(W^t)\right\|_F^2$$

$$= -\gamma p \left\langle \nabla f(W^t), H_B^t \nabla f(W^t)\right\rangle_F + \frac{L\gamma^2}{2} p \left\langle H_B^t \nabla f(W^t), H_B^t \nabla f(W^t)\right\rangle_F$$

$$= -\gamma p \left\langle \nabla f(W^t), H_B^t \nabla f(W^t)\right\rangle_F + \frac{L\gamma^2}{2} p \left\langle \nabla f(W^t), \left(H_B^t\right)^\top H_B^t \nabla f(W^t)\right\rangle_F$$

$$= p\left(-\gamma \left\langle \nabla f(W^t), H_B^t \nabla f(W^t)\right\rangle_F + \frac{L\gamma^2}{2} \left\langle \nabla f(W^t), H_B^t \nabla f(W^t)\right\rangle_F\right)$$

$$\overset{\gamma \leq 1/L}{\leq} -\frac{\gamma}{2} p \left\langle \nabla f(W^t), H_B^t \nabla f(W^t)\right\rangle_F. \tag{43}$$

For any projection matrix $H_A^t$, we have:

$$\left\langle \nabla f(W^t) H_A^t, \nabla f(W^t) H_A^t\right\rangle_F = \text{Tr}\left(\left(H_A^t\right)^\top \left(\nabla f(W^t)\right)^\top \nabla f(W^t) H_A^t\right)$$

$$= \text{Tr}\left(\left(\nabla f(W^t)\right)^\top \nabla f(W^t) H_A^t \left(H_A^t\right)^\top\right)$$

$$= \text{Tr}\left(\left(\nabla f(W^t)\right)^\top \nabla f(W^t) H_A^t\right)$$

$$= \left\langle \nabla f(W^t), \nabla f(W^t) H_A^t\right\rangle_F. \tag{44}$$

Therefore:

$$-\gamma(1-p) \left\langle \nabla f(W^t), \nabla f(W^t) H_A^t\right\rangle_F + \frac{L\gamma^2}{2}(1-p) \left\|\nabla f(W^t) H_A^t\right\|_F^2$$

$$= -\gamma(1-p) \left\langle \nabla f(W^t), \nabla f(W^t) H_A^t\right\rangle_F + \frac{L\gamma^2}{2}(1-p) \left\langle \nabla f(W^t) H_A^t, \nabla f(W^t) H_A^t\right\rangle_F$$

$$= -\gamma(1-p) \left\langle \nabla f(W^t), \nabla f(W^t) H_A^t\right\rangle_F + \frac{L\gamma^2}{2}(1-p) \left\langle \nabla f(W^t), \nabla f(W^t) H_A^t\right\rangle_F$$

$$\overset{\gamma \leq 1/L}{\leq} -\frac{\gamma}{2}(1-p) \left\langle \nabla f(W^t), \nabla f(W^t) H_A^t\right\rangle_F. \tag{45}$$

Using the Lipschitz gradient condition and the above bounds:

$$
\begin{aligned}
\mathbb{E}\left[f(W^{t+1}) \mid W^t, H^t\right] &\leq f(W^t) + \mathbb{E}\left[\left\langle \nabla f(W^t), W^{t+1} - W^t \right\rangle_{\mathrm{F}} \mid W^t, H^t\right] \\
&\quad + \frac{L}{2}\mathbb{E}\left[\left\|W^{t+1} - W^t\right\|_{\mathrm{F}}^2 \mid W^t, H^t\right] \\
&= f(W^t) - \gamma\left\langle \nabla f(W^t), \mathbb{E}\left[G^t \mid W^t, H^t\right]\right\rangle_{\mathrm{F}} + \frac{L\gamma^2}{2}\mathbb{E}\left[\left\|G^t\right\|_{\mathrm{F}}^2 \mid W^t, H^t\right] \\
&= f(W^t) - \gamma p\left\langle \nabla f(W^t), H_B^t \nabla f(W^t)\right\rangle_{\mathrm{F}} - \gamma(1-p)\left\langle \nabla f(W^t), \nabla f(W^t) H_A^t\right\rangle_{\mathrm{F}} \\
&\quad + \frac{L\gamma^2}{2}p\left\|H_B^t \nabla f(W^t)\right\|_{\mathrm{F}}^2 + \frac{L\gamma^2}{2}(1-p)\left\|\nabla f(W^t) H_A^t\right\|_{\mathrm{F}}^2 \\
&\overset{(43),(45)}{\leq} f(W^t) - \frac{\gamma}{2}\left(p\left\langle \nabla f(W^t), H_B^t \nabla f(W^t)\right\rangle_{\mathrm{F}} + (1-p)\left\langle \nabla f(W^t), \nabla f(W^t) H_A^t\right\rangle_{\mathrm{F}}\right).
\end{aligned}
$$
$$\tag{46}$$

For the first term:

$$
\begin{aligned}
-\left\langle \nabla f(W^t), \mathbb{E}\left[H_B^t\right] \nabla f(W^t)\right\rangle_{\mathrm{F}} &= -\mathrm{Tr}\left(\left(\nabla f(W^t)\right)^\top \mathbb{E}\left[H_B^t\right] \nabla f(W^t)\right) \\
&\leq -\lambda_{\min}\left(\mathbb{E}\left[H_B^t\right]\right)\mathrm{Tr}\left(\left(\nabla f(W^t)\right)^\top \nabla f(W^t)\right) \\
&= -\lambda_{\min}^{H_B}\left\|\nabla f(W^t)\right\|_{\mathrm{F}}^2.
\end{aligned}
$$
$$\tag{47}$$

Similarly, for the second term:

$$
\begin{aligned}
-\left\langle \nabla f(W^t), \nabla f(W^t)\mathbb{E}\left[H_A^t\right]\right\rangle_{\mathrm{F}} &= -\mathrm{Tr}\left(\left(\nabla f(W^t)\right)^\top \nabla f(W^t)\mathbb{E}\left[H_A^t\right]\right) \\
&= -\mathrm{Tr}\left(\mathbb{E}\left[H_A^t\right]\left(\nabla f(W^t)\right)^\top \nabla f(W^t)\right) \\
&\leq -\lambda_{\min}^{H_A}\left\|\nabla f(W^t)\right\|_{\mathrm{F}}^2.
\end{aligned}
$$
$$\tag{48}$$

Therefore:

$$
\begin{aligned}
\mathbb{E}\left[f(W^{t+1}) \mid W^t\right] &= \mathbb{E}\left[\mathbb{E}\left[f(W^{t+1}) \mid W^t, H^t\right] \mid W^t\right] \\
&\leq f(W^t) - \frac{\gamma}{2}\left(p\left\langle \nabla f(W^t), \mathbb{E}\left[H_B^t\right] \nabla f(W^t)\right\rangle_{\mathrm{F}} + (1-p)\left\langle \nabla f(W^t), \nabla f(W^t)\mathbb{E}\left[H_A^t\right]\right\rangle_{\mathrm{F}}\right) \\
&\leq f(W^t) - \frac{\gamma}{2}\left(p\lambda_{\min}^{H_B} + (1-p)\lambda_{\min}^{H_A}\right)\left\|\nabla f(W^t)\right\|_{\mathrm{F}}^2 \\
&= f(W^t) - \frac{\gamma}{2}\lambda_{\min}^p\left\|\nabla f(W^t)\right\|_{\mathrm{F}}^2,
\end{aligned}
$$
$$\tag{49}$$

where $\lambda_{\min}^p := p\lambda_{\min}^{H_B} + (1-p)\lambda_{\min}^{H_A}$. Further,

$$
\mathbb{E}\left[\mathbb{E}\left[f(W^{t+1}) \mid W^t, H^t\right] \mid W^t\right] - f^\star \leq f(W^t) - f^\star - \frac{\gamma}{2}\lambda_{\min}^p\left\|\nabla f(W^t)\right\|_{\mathrm{F}}^2.
$$
$$\tag{50}$$

Taking the sum over $t = 0, \ldots, T-1$ and using the tower property of expectation:

$$
\mathbb{E}\left[f(W^T) - f^\star\right] \leq f(W^0) - f^\star - \frac{\gamma}{2}\lambda_{\min}^p\sum_{t=0}^{T-1}\mathbb{E}\left[\left\|\nabla f(W^t)\right\|_{\mathrm{F}}^2\right].
$$
$$\tag{51}$$

By rearranging terms, we get:

$$
\frac{\gamma}{2}\lambda_{\min}^p\sum_{t=0}^{T-1}\mathbb{E}\left[\left\|\nabla f(W^t)\right\|_{\mathrm{F}}^2\right] \leq f(W^0) - f^\star.
$$
$$\tag{52}$$

Finally, dividing both sides by $\frac{\gamma T}{2}\lambda_{\min}^p$ yields:

$$
\mathbb{E}\left[\left\|\nabla f(\widetilde{W}^T)\right\|_{\mathrm{F}}^2\right] \leq \frac{2(f(W^0) - f^\star)}{\gamma\lambda_{\min}^p T},
$$
$$\tag{53}$$

where $\widetilde{W}^T$ is chosen uniformly at random from $\{W^0, W^1, \ldots, W^{T-1}\}$, completing the proof.

$\square$

### H.1.2 Convergence under Polyak-Łojasiewicz Condition

**Theorem 9.** *Let Assumptions 1, 2, 3, and 6 hold, and let the stepsize satisfy $0 < \gamma \leq \frac{1}{L}$. Then the iterates of* Bernoulli-LoRA-GD *(Algorithm 2), with matrices $\hat{A}^t$ and $\hat{B}^t$ computed according to Lemma 10, satisfy*

$$\mathbb{E}\left[f(W^T) - f^*\right] \leq \left(1 - \gamma\mu\lambda_{\min}^p\right)^T \left(f(W^0) - f^*\right),$$

*where $\lambda_{\min}^p := p\lambda_{\min}^{H_B} + (1-p)\lambda_{\min}^{H_A}$.*

*Proof.* We begin our analysis from a key inequality derived in the proof of Theorem 1:

$$\mathbb{E}\left[f(W^{t+1}) \mid W^t\right] \leq f(W^t) - \frac{\gamma}{2}\lambda_{\min}^p \left\|\nabla f(W^t)\right\|_{\mathrm{F}}^2. \tag{54}$$

By invoking the Polyak-Łojasiewicz condition (Assumption 6), which states that $\frac{1}{2}\left\|\nabla f(W)\right\|_{\mathrm{F}}^2 \geq \mu\left(f(W) - f^*\right)$, we can further bound the right-hand side of the inequality (54):

$$\mathbb{E}\left[f(W^{t+1}) \mid W^t\right] \leq f(W^t) - \gamma\lambda_{\min}^p \left(\mu\left(f(W^t) - f^*\right)\right).$$

Subtracting the optimal function value $f^*$ from both sides, we get a recursive relationship for the expected suboptimality gap:

$$\mathbb{E}\left[f(W^{t+1}) - f^* \mid W^t\right] \leq \left(f(W^t) - f^*\right) - \gamma\mu\lambda_{\min}^p \left(f(W^t) - f^*\right)$$
$$= \left(1 - \gamma\mu\lambda_{\min}^p\right)\left(f(W^t) - f^*\right).$$

By taking the full expectation over all randomness up to iteration $t$ and applying the tower property, we obtain:

$$\mathbb{E}\left[f(W^{t+1}) - f^*\right] \leq \left(1 - \gamma\mu\lambda_{\min}^p\right)\mathbb{E}\left[f(W^t) - f^*\right].$$

Unrolling this recursion from $t = T - 1$ down to $t = 0$ yields the final linear convergence result:

$$\mathbb{E}\left[f(W^T) - f^*\right] \leq \left(1 - \gamma\mu\lambda_{\min}^p\right)^T \left(f(W^0) - f^*\right).$$

This completes the proof. $\qquad\square$

### H.1.3 Convergence for Non-Smooth Convex Functions

---

**Algorithm 3** Bernoulli-LoRA-GD (Non-smooth setting)

---

1: **Parameters:** pre-trained model $W^0 \in \mathbb{R}^{m \times n}$, rank $r \ll \min\{m,n\}$, scaling factor $\alpha > 0$, stepsize $\gamma_t$ chain length $T$, sketch distribution $\mathcal{D}_S^B$ or $\mathcal{D}_S^A$, Bernoulli probability $p$
2: **for** $t = 0, 1, \ldots, T - 1$ **do**
3:     Sample $c^t \sim \mathrm{Be}(p)$                                              Bernoulli random variable
4:     **if** $c^t = 1$ **then**
5:         Sample $B_S^t \sim \mathcal{D}_S^B$                                     Left sketch
6:         $\hat{A}^t = \arg\min_A \left\{f(W^t) + \frac{\alpha}{r}\langle \partial f(W^t), B_S^t A\rangle_{\mathrm{F}} + \frac{\alpha^2}{2\gamma_t r^2}\left\|B_S^t A\right\|_{\mathrm{F}}^2\right\}$
7:         $W^{t+1} = W^t + \frac{\alpha}{r}B_S^t \hat{A}^t$
8:     **else**
9:         Sample $A_S^t \sim \mathcal{D}_S^A$                                    Right sketch
10:        $\hat{B}^t = \arg\min_B \left\{f(W^t) + \frac{\alpha}{r}\langle \partial f(W^t), BA_S^t\rangle_{\mathrm{F}} + \frac{\alpha^2}{2\gamma_t r^2}\left\|BA_S^t\right\|_{\mathrm{F}}^2\right\}$
11:        $W^{t+1} = W^t + \frac{\alpha}{r}\hat{B}^t A_S^t$
12:     **end if**
13: **end for**

---

Our analysis relies on the following standard assumptions that are widely used in non-smooth optimization theory:

**Assumption 7.** *The function $f$ has at least one minimizer, denoted by $W^*$.*

**Assumption 8.** *The function $f$ is convex.*

**Assumption 9** (Lipschitz continuity). *The function $f$ is $L_0$-Lipschitz continuous. That is, there exists $L_0 > 0$ such that*

$$|f(W) - f(V)| \leq L_0 \left\|W - V\right\|_{\mathrm{F}}, \quad \forall W, V \in \mathbb{R}^{m \times n}. \tag{55}$$

The combination of convexity and Lipschitz continuity represents a standard framework in non-smooth optimization (Vorontsova et al., 2021; Nesterov, 2013; Bubeck, 2015; Beck, 2017; Duchi,

2018; Lan, 2020; Drusvyatskiy, 2020). Notably, the $L_0$-Lipschitz continuity implies uniformly bounded subgradients (Beck, 2017), a property that plays a crucial role in our analysis:

$$\|\partial f(W)\|_{\mathrm{F}} \leq L_0, \quad \forall W \in \mathbb{R}^{m \times n}. \tag{56}$$

This boundedness of subgradients ensures the stability of our optimization process and enables us to establish rigorous convergence guarantees.

The following lemma establishes that the Bernoulli-LoRA update in the non-smooth case can also be reformulated as a subgradient descent step, which plays a central role in our convergence analysis for non-smooth objectives.

**Lemma 11.** *Consider the updates $\hat{A}^t$ and $\hat{B}^t$ from Algorithm 3 computed as solutions to the following optimization problems:*

$$\hat{A}^t := \arg\min_A \left\{ f(W^t) + \frac{\alpha}{r} \left\langle \partial f\left(W^t\right), B_S^t A \right\rangle_{\mathrm{F}} + \frac{\alpha^2}{2\gamma_t r^2} \left\| B_S^t A \right\|_{\mathrm{F}}^2 \right\},$$

$$\hat{B}^t := \arg\min_B \left\{ f(W^t) + \frac{\alpha}{r} \left\langle \partial f\left(W^t\right), B A_S^t \right\rangle_{\mathrm{F}} + \frac{\alpha^2}{2\gamma_t r^2} \left\| B A_S^t \right\|_{\mathrm{F}}^2 \right\}. \tag{57}$$

*Then the Left and Right sketch updates can be expressed as a subgradient descent step:*

$$W^{t+1} = W^t - \gamma_t G^t, \tag{58}$$

*where $G^t$ is defined by*

$$G^t = \begin{cases} H_B^t \partial f\left(W^t\right), & \text{with probability } p \\ \partial f\left(W^t\right) H_A^t, & \text{with probability } 1-p \end{cases} \tag{59}$$

*with projection matrices $H_A^t$ and $H_B^t$ given by:*

$$H_A^t := \left(A_S^t\right)^{\top} \left(A_S^t \left(A_S^t\right)^{\top}\right)^{\dagger} A_S^t \quad \text{and} \quad H_B^t := B_S^t \left(\left(B_S^t\right)^{\top} B_S^t\right)^{\dagger} \left(B_S^t\right)^{\top}, \tag{60}$$

*where $\dagger$ denotes the Moore-Penrose pseudoinverse.*

*Proof.* The proof follows a similar structure to that of Lemma 10, with subgradients replacing gradients throughout the analysis. We examine both sketch types separately before combining them into a unified update rule.

**Left Sketch Analysis.** When the Left sketch is selected, the update takes the form:

$$W^{t+1} = W^t + \frac{\alpha}{r} B_S^t \hat{A}^t. \tag{61}$$

The matrix $\hat{A}^t$ is defined as the solution to the optimization problem:

$$\hat{A}^t := \arg\min_A \left\{ f(W^t) + \frac{\alpha}{r} \left\langle \partial f\left(W^t\right), B_S^t A \right\rangle_{\mathrm{F}} + \frac{\alpha^2}{2\gamma_t r^2} \left\| B_S^t A \right\|_{\mathrm{F}}^2 \right\}. \tag{62}$$

By computing the gradient of the objective with respect to $A$ and setting it to zero, we obtain:

$$\frac{\alpha}{r} \left(B_S^t\right)^{\top} \partial f\left(W^t\right) + \frac{\alpha^2}{\gamma_t r^2} \left(B_S^t\right)^{\top} B_S^t \hat{A}^t = 0;$$

$$\hat{A}^t = -\frac{\gamma_t r}{\alpha} \left(\left(B_S^t\right)^{\top} B_S^t\right)^{\dagger} \left(B_S^t\right)^{\top} \partial f\left(W^t\right). \tag{63}$$

Substituting this expression back into the update equation yields the Left sketch update:

$$\begin{aligned} W^{t+1} &= W^t + \frac{\alpha}{r} B_S^t \hat{A}^t \\ &= W^t - \gamma_t B_S^t \left(\left(B_S^t\right)^{\top} B_S^t\right)^{\dagger} \left(B_S^t\right)^{\top} \partial f\left(W^t\right) \\ &= W^t - \gamma_t H_B^t \partial f\left(W^t\right). \end{aligned} \tag{64}$$

**Right Sketch Analysis.** For the Right sketch, we follow an analogous approach. The update rule takes the form:

$$W^{t+1} = W^t + \frac{\alpha}{r} \hat{B}^t A_S^t. \tag{65}$$

Applying similar optimization steps but now with respect to matrix $B$, we obtain:

$$\hat{B}^t = -\frac{\gamma_t r}{\alpha} \partial f\left(W^t\right) \left(A_S^t\right)^{\top} \left(A_S^t \left(A_S^t\right)^{\top}\right)^{\dagger}, \tag{66}$$

which leads to the Right sketch update:

$$
\begin{aligned}
W^{t+1} &= W^t + \frac{\alpha}{r} \hat{B}^t A_S^t \\
&= W^t - \gamma_t \partial f\left(W^t\right) \left(A_S^t\right)^\top \left(A_S^t \left(A_S^t\right)^\top\right)^\dagger A_S^t \\
&= W^t - \gamma_t \partial f\left(W^t\right) H_A^t.
\end{aligned}
\tag{67}
$$

**Combined Update Rule.** By combining equations (64) and (67), we arrive at the unified update rule:
$$
W^{t+1} = W^t - \gamma_t G^t,
\tag{68}
$$
where $G^t$ takes the form specified in the lemma statement, thus completing the proof. $\qquad\square$

**Assumption 10.** *Consider a projection matrix $H$ generated through either Left Sketch (Definition 5) or Right Sketch (Definition 6). For the sampling distributions $\mathcal{D}_S^{\hat{B}}$ and $\mathcal{D}_S^A$, the expected projection matrix $H$ satisfies*
$$
\mathbb{E}[H] = \alpha I,
\tag{69}
$$
*where a constant $\alpha > 0$.*

**Theorem 10.** *Let Assumptions 1, 7, 8, 9, and 10 hold. Let us define the following quantities: $\overline{W}^T := \frac{1}{T}\sum_{t=0}^{T-1} W^t$ as the averaged iterate, $R_0^2 := \left\|W^0 - W^*\right\|_{\mathrm{F}}^2$ as the initial distance to optimum. Consider the sequence $\{W^t\}$ produced by* Bernoulli-LoRA *(Algorithm 3) with updates of $\hat{A}^t$ and $\hat{B}^t$ computed according to Lemma 11.*

*1. (Constant stepsize). If the stepsize is constant, i.e., $\gamma_t := \gamma > 0$, then*
$$
\mathbb{E}\left[f(\overline{W}^T) - f(W^*)\right] \leq \frac{R_0^2}{2\gamma\alpha T} + \frac{\gamma L_0^2}{2}.
\tag{70}
$$
*Moreover, with the optimal stepsize $\gamma_* = \sqrt{\frac{(R^0)^2}{T\alpha L_0^2}}$, we obtain:*
$$
\mathbb{E}\left[f(\overline{W}^T) - f(W^*)\right] \leq \frac{R^0 L_0}{\sqrt{\alpha T}}.
\tag{71}
$$

*2. (Polyak stepsize). If the stepsize is chosen adaptively as*
$$
\gamma_t = \frac{(f(W^t) - f(W^*))}{\|\partial f(W^t)\|_{\mathrm{F}}^2},
\tag{72}
$$
*then*
$$
\mathbb{E}\left[f(\overline{W}^T) - f(W^*)\right] \leq \frac{R^0 L_0}{\sqrt{\alpha T}}.
\tag{73}
$$

*Proof.* From Lemma 11, we know that Bernoulli-LoRA updates in the non-smooth setting can be expressed as
$$
W^{t+1} = W^t - \gamma_t G^t,
\tag{74}
$$
where $G^t$ takes the form
$$
G^t = \begin{cases} H_B^t \partial f(W^t), & \text{with probability } p \\ \partial f(W^t) H_A^t, & \text{with probability } 1 - p \end{cases}
\tag{75}
$$
with projection matrices $H_A^t$ and $H_B^t$ as defined in the lemma.

To analyze the convergence, we first compute the conditional expectation and second moment of $G^t$:
$$
\mathbb{E}\left[G^t \mid W^t, H^t\right] = p H_B^t \partial f(W^t) + (1 - p)\partial f(W^t) H_A^t,
\tag{76}
$$
$$
\mathbb{E}\left[\|G^t\|_{\mathrm{F}}^2 \mid W^t, H^t\right] = p\left\|H_B^t \partial f(W^t)\right\|_{\mathrm{F}}^2 + (1 - p)\left\|\partial f(W^t) H_A^t\right\|_{\mathrm{F}}^2,
\tag{77}
$$
where we defined $H^t := \{H_A^t, H_B^t\}$.

By the definition of subgradient, we have:
$$
f(W^*) \geq f(W^t) + \left\langle \partial f(W^t), W^* - W^t \right\rangle_{\mathrm{F}},
\tag{78}
$$
which implies:
$$
\left\langle \partial f(W^t), W^t - W^* \right\rangle_{\mathrm{F}} \geq f(W^t) - f(W^*).
\tag{79}
$$
Let us establish key auxiliary bounds. First, for the inner product terms:
$$
-2\gamma_t \mathbb{E}\left[\left\langle G^t, W^t - W^* \right\rangle_{\mathrm{F}} \mid W^t, H^t\right] \overset{(76)}{=} -2\gamma_t p \left\langle H_B^t \partial f(W^t), W^t - W^* \right\rangle_{\mathrm{F}}
$$
$$
-2\gamma_t(1 - p) \left\langle \partial f(W^t) H_A^t, W^t - W^* \right\rangle_{\mathrm{F}}.
\tag{80}
$$

For projection matrices, we have the following properties:

$$
\begin{aligned}
\left\| \partial f(W^t) H_A^t \right\|_{\mathrm{F}}^2 &= \left\langle \partial f(W^t) H_A^t, \partial f(W^t) H_A^t \right\rangle_{\mathrm{F}} \\
&= \mathrm{Tr} \left( \left( H_A^t \right)^\top \left( \partial f(W^t) \right)^\top \partial f(W^t) H_A^t \right) \\
&= \mathrm{Tr} \left( \left( \nabla f(W^t) \right)^\top \nabla f(W^t) H_A^t \left( H_A^t \right)^\top \right) \\
&= \mathrm{Tr} \left( \left( \partial f(W^t) \right)^\top \partial f(W^t) H_A^t \right) \\
&= \left\langle \partial f(W^t), \partial f(W^t) H_A^t \right\rangle_{\mathrm{F}},
\end{aligned}
\tag{81}
$$

and similarly, one can show that

$$
\left\| H_B^t \partial f(W^t) \right\|_{\mathrm{F}}^2 = \left\langle \partial f(W^t), H_B^t \partial f(W^t) \right\rangle_{\mathrm{F}}.
\tag{82}
$$

This allows us to express the second moment term as:

$$
\begin{aligned}
\gamma_t^2 \mathbb{E} \left[ \left\| G^t \right\|_{\mathrm{F}}^2 \mid W^t, H^t \right] &\overset{(77)}{=} \gamma_t^2 p \left\| H_B^t \partial f(W^t) \right\|_{\mathrm{F}}^2 + \gamma_t^2 (1-p) \left\| \partial f(W^t) H_A^t \right\|_{\mathrm{F}}^2 \\
&\overset{(81),(82)}{=} \gamma_t^2 p \left\langle \partial f(W^t), H_B^t \partial f(W^t) \right\rangle_{\mathrm{F}} + \gamma_t^2 (1-p) \left\langle \partial f(W^t), \partial f(W^t) H_A^t \right\rangle_{\mathrm{F}}.
\end{aligned}
\tag{83}
$$

Combining these bounds, we can analyze the distance to the optimal solution:

$$
\begin{aligned}
\mathbb{E} \left[ \left\| W^{t+1} - W^* \right\|_{\mathrm{F}}^2 \mid W^t, H^t \right] &= \mathbb{E} \left[ \left\| W^t - \gamma_t G^t - W^* \right\|_{\mathrm{F}}^2 \mid W^t, H^t \right] \\
&= \left\| W^t - W^* \right\|_{\mathrm{F}}^2 - 2\gamma_t \mathbb{E} \left[ \left\langle G^t, W^t - W^* \right\rangle_{\mathrm{F}} \mid W^t, H^t \right] \\
&\quad + \gamma_t^2 \mathbb{E} \left[ \left\| G^t \right\|_{\mathrm{F}}^2 \mid W^t, H^t \right] \\
&\overset{(80),(83)}{=} \left\| W^t - W^* \right\|_{\mathrm{F}}^2 - 2\gamma_t p \left\langle H_B^t \partial f(W^t), W^t - W^* \right\rangle_{\mathrm{F}} \\
&\quad - 2\gamma_t (1-p) \left\langle \partial f(W^t) H_A^t, W^t - W^* \right\rangle_{\mathrm{F}} + \gamma_t^2 p \left\langle \partial f(W^t), H_B^t \partial f(W^t) \right\rangle_{\mathrm{F}} \\
&\quad + \gamma_t^2 (1-p) \left\langle \partial f(W^t), \partial f(W^t) H_A^t \right\rangle_{\mathrm{F}}.
\end{aligned}
\tag{84}
$$

For the expected projection matrices (see Assumption 10), we have:

$$
\begin{aligned}
\left\langle \partial f(W^t), \mathbb{E} \left[ H_B^t \right] \partial f(W^t) \right\rangle_{\mathrm{F}} &= \mathrm{Tr} \left( \left( \partial f(W^t) \right)^\top \mathbb{E} \left[ H_B^t \right] \partial f(W^t) \right) \\
&= \alpha \, \mathrm{Tr} \left( \left( \partial f(W^t) \right)^\top \partial f(W^t) \right) \\
&= \alpha \left\| \partial f(W^t) \right\|_{\mathrm{F}}^2,
\end{aligned}
\tag{85}
$$

and similarly,

$$
\left\langle \partial f(W^t), \partial f(W^t) \mathbb{E} \left[ H_A^t \right] \right\rangle_{\mathrm{F}} = \alpha \left\| \partial f(W^t) \right\|_{\mathrm{F}}^2.
\tag{86}
$$

Taking expectation of both sides of (84) again, we get

$$
\mathbb{E} \left[ \left\| W^{t+1} - W^* \right\|_{\mathrm{F}}^2 \mid W^t \right] = \mathbb{E} \left[ \mathbb{E} \left[ \left\| W^{t+1} - W^* \right\|_{\mathrm{F}}^2 \mid W^t, H^t \right] \mid W^t \right]
\tag{87}
$$

$$
= \left\| W^t - W^* \right\|_{\mathrm{F}}^2 - 2\gamma_t p \left\langle \mathbb{E} \left[ H_B^t \right] \partial f \left( W^t \right), W^t - W^* \right\rangle_{\mathrm{F}}
\tag{88}
$$

$$
- 2\gamma_t (1-p) \left\langle \partial f \left( W^t \right) \mathbb{E} \left[ H_A^t \right], W^t - W^* \right\rangle_{\mathrm{F}}
$$

$$
+ \gamma_t^2 p \left\langle \partial f \left( W^t \right), \mathbb{E} \left[ H_B^t \right] \partial f \left( W^t \right) \right\rangle_{\mathrm{F}} + \gamma_t^2 (1-p) \left\langle \partial f \left( W^t \right), \partial f \left( W^t \right) \mathbb{E} \left[ H_A^t \right] \right\rangle_{\mathrm{F}}
$$

$$
\overset{(85),(86)}{=} \left\| W^t - W^* \right\|_{\mathrm{F}}^2 - 2\gamma_t p \alpha \left\langle \partial f \left( W^t \right), W^t - W^* \right\rangle_{\mathrm{F}}
\tag{89}
$$

$$
- 2\gamma_t (1-p) \alpha \left\langle \partial f \left( W^t \right), W^t - W^* \right\rangle_{\mathrm{F}} + \gamma_t^2 \alpha \left\| \partial f \left( W^t \right) \right\|_{\mathrm{F}}^2
$$

$$
= \left\| W^t - W^* \right\|_{\mathrm{F}}^2 - 2\gamma_t \alpha \left\langle \partial f \left( W^t \right), W^t - W^* \right\rangle_{\mathrm{F}} + \gamma_t^2 \alpha \left\| \partial f \left( W^t \right) \right\|_{\mathrm{F}}^2
$$

$$
\overset{(79)}{=} \left\| W^t - W^* \right\|_{\mathrm{F}}^2 - 2\gamma_t \alpha \left( f(W^t) - f(W^*) \right) + \gamma_t^2 \alpha \left\| \partial f \left( W^t \right) \right\|_{\mathrm{F}}^2.
\tag{90}
$$

By Assumption 9, subgradients are uniformly bounded (see (Beck, 2017)):

$$
\left\| \partial f(W) \right\|_{\mathrm{F}} \leq L_0 \quad \forall W \in \mathbb{R}^{m \times n}.
\tag{91}
$$

Now we analyze both stepsize strategies separately.

**1. (Constant stepsize).** Let us first consider using a fixed stepsize $\gamma_t := \gamma > 0$. Taking expectation of both sides of (87) again, applying tower property (11) and using the bound (91), we obtain:

$$\mathbb{E}\left[\left\|W^{t+1} - W^*\right\|_{\mathrm{F}}^2\right] \leq \mathbb{E}\left[\left\|W^t - W^*\right\|_{\mathrm{F}}^2\right] - 2\gamma\alpha\mathbb{E}\left[f(W^t) - f(W^*)\right] + \gamma^2\alpha L_0^2. \tag{92}$$

Rearranging terms in (92):

$$2\gamma\alpha\mathbb{E}\left[f(W^t) - f(W^*)\right] \leq \mathbb{E}\left[\left\|W^t - W^*\right\|_{\mathrm{F}}^2\right] - \mathbb{E}\left[\left\|W^{t+1} - W^*\right\|_{\mathrm{F}}^2\right] + \gamma^2\alpha L_0^2. \tag{93}$$

Summing inequality (93) for $t = 0, \ldots, T-1$:

$$\begin{aligned}
2\gamma\alpha\sum_{t=0}^{T-1}\mathbb{E}\left[f(W^t) - f(W^*)\right] &\leq \sum_{t=0}^{T-1}\left(\mathbb{E}\left[\left\|W^t - W^*\right\|_{\mathrm{F}}^2\right] - \mathbb{E}\left[\left\|W^{t+1} - W^*\right\|_{\mathrm{F}}^2\right]\right) \\
&\quad + T\gamma^2\alpha L_0^2 \\
&= \mathbb{E}\left[\left\|W^0 - W^*\right\|_{\mathrm{F}}^2\right] - \mathbb{E}\left[\left\|W^T - W^*\right\|_{\mathrm{F}}^2\right] + T\gamma^2\alpha L_0^2 \\
&\leq \left\|W^0 - W^*\right\|_{\mathrm{F}}^2 + T\gamma^2\alpha L_0^2, \tag{94}
\end{aligned}$$

where the last inequality follows from the non-negativity of $\left\|W^T - W^*\right\|_{\mathrm{F}}^2$.

For the averaged iterate $\overline{W}^T := \frac{1}{T}\sum_{t=0}^{T-1} W^t$, by convexity of $f$ we have:

$$\begin{aligned}
\mathbb{E}\left[f(\overline{W}^T) - f(W^*)\right] &\leq \frac{1}{T}\sum_{t=0}^{T-1}\mathbb{E}\left[f(W^t) - f(W^*)\right] \\
&\overset{(94)}{\leq} \frac{\left\|W^0 - W^*\right\|_{\mathrm{F}}^2}{2\gamma\alpha T} + \frac{\gamma L_0^2}{2} \\
&= \frac{(R^0)^2}{2\gamma\alpha T} + \frac{\gamma L_0^2}{2}, \tag{95}
\end{aligned}$$

where we denoted $(R^0)^2 := \left\|W^0 - W^*\right\|_{\mathrm{F}}^2$.

To optimize this bound, we minimize it with respect to $\gamma$. The optimal stepsize $\gamma_*$ solves:

$$\begin{aligned}
\gamma_* &= \arg\min_{\gamma > 0}\left(\frac{(R^0)^2}{2\gamma\alpha T} + \frac{\gamma L_0^2}{2}\right) \\
&= \sqrt{\frac{(R^0)^2}{T\alpha L_0^2}}. \tag{96}
\end{aligned}$$

Substituting $\gamma_*$ back into (95), we obtain the optimal convergence rate:

$$\mathbb{E}\left[f(\overline{W}^T) - f(W^*)\right] \leq \frac{R^0 L_0}{\sqrt{\alpha T}}. \tag{97}$$

**2. (Polyak stepsize).** For this strategy, we choose the stepsize adaptively based on the current function value:

$$\begin{aligned}
\gamma_t &= \arg\min_{\gamma > 0}\left\{\left\|W^t - W^*\right\|_{\mathrm{F}}^2 - 2\gamma\alpha\left(f(W^t) - f(W^*)\right) + \gamma^2\alpha\left\|\partial f\left(W^t\right)\right\|_{\mathrm{F}}^2\right\} \\
&= \frac{(f(W^t) - f(W^*))}{\left\|\partial f(W^t)\right\|_{\mathrm{F}}^2}. \tag{98}
\end{aligned}$$

Substituting this stepsize into inequality (87):

$$\begin{aligned}
\mathbb{E}\left[\left\|W^{t+1} - W^*\right\|_{\mathrm{F}}^2 \mid W^t\right] &= \mathbb{E}\left[\mathbb{E}\left[\left\|W^{t+1} - W^*\right\|_{\mathrm{F}}^2 \mid W^t, H^t\right] \mid W^t\right] \\
&\leq \left\|W^t - W^*\right\|_{\mathrm{F}}^2 - 2\gamma_t\alpha\left(f(W^t) - f(W^*)\right) + \gamma_t^2\alpha\left\|\partial f\left(W^t\right)\right\|_{\mathrm{F}}^2 \\
&\overset{(98)}{=} \left\|W^t - W^*\right\|_{\mathrm{F}}^2 - \frac{\alpha\left(f(W^t) - f(W^*)\right)^2}{\left\|\partial f(W^t)\right\|_{\mathrm{F}}^2} \\
&\overset{(91)}{\leq} \left\|W^t - W^*\right\|_{\mathrm{F}}^2 - \frac{\alpha\left(f(W^t) - f(W^*)\right)^2}{L_0^2}. \tag{99}
\end{aligned}$$

Taking expectation of both sides of (99) again and applying the tower property

$$\mathbb{E}\left[\left\|W^{t+1} - W^*\right\|_{\mathrm{F}}^2\right] \leq \mathbb{E}\left[\left\|W^t - W^*\right\|_{\mathrm{F}}^2\right] - \frac{\alpha \mathbb{E}\left[(f(W^t) - f(W^*))^2\right]}{L_0^2} \tag{100}$$

Since $f$ is convex, by Jensen's inequality (14) and the Cauchy-Bunyakovsky-Schwarz inequality (12) with $X := f(W^t) - f(W^*)$ and $Y := 1$, we have

$$\mathbb{E}\left[f_i(\overline{W}^T) - f(W^*)\right] \overset{(14)}{\leq} \mathbb{E}\left[\frac{1}{T}\sum_{t=0}^{T-1} f(W^t) - f(W^*)\right]$$

$$\leq \frac{1}{T}\sum_{t=0}^{T-1} \mathbb{E}\left[f(W^t) - f(W^*)\right]$$

$$\overset{(12)}{\leq} \frac{1}{T}\sum_{t=0}^{T-1} \sqrt{\mathbb{E}\left[(f(W^t) - f(W^*))^2\right]}$$

$$\leq \sqrt{\frac{1}{T}\sum_{t=0}^{T-1} \mathbb{E}\left[(f(W^t) - f(W^*))^2\right]}$$

$$\overset{(100)}{\leq} \frac{R^0 L_0}{\sqrt{\alpha T}}, \tag{101}$$

which matches the optimal rate achieved by the constant stepsize strategy with optimal tuning. $\square$

## H.2 ANALYSIS OF BERNOULLI-LORA-SGD

---

**Algorithm 4** Bernoulli-LoRA-SGD

---

1: **Parameters:** pre-trained model $W^0 \in \mathbb{R}^{m \times n}$, rank $r \ll \min\{m,n\}$, scaling factor $\alpha > 0$, chain length $T$, sketch distribution $\mathcal{D}_S^B$ or $\mathcal{D}_S^A$, Bernoulli probability $p$
2: **for** $t = 0, 1, \ldots, T-1$ **do**
3:     Sample $c^t \sim \text{Be}(p)$                                                              Bernoulli random variable
4:     **if** $c^t = 1$ **then**
5:         Sample $B_S^t \sim \mathcal{D}_S^B$                                                         Left sketch
6:         $\hat{A}^t = -\eta \left( (B_S^t)^\top B_S^t \right)^\dagger (B_S^t)^\top g(W^t)$
7:         $W^{t+1} = W^t + \frac{\alpha}{r} B_S^t \hat{A}^t$
8:     **else**
9:         Sample $A_S^t \sim \mathcal{D}_S^A$                                                         Right sketch
10:         $\hat{B}^t = -\eta g(W^t)(A_S^t)^\top \left( A_S^t (A_S^t)^\top \right)^\dagger$
11:         $W^{t+1} = W^t + \frac{\alpha}{r} \hat{B}^t A_S^t$
12:     **end if**
13: **end for**

---

Earlier findings were derived utilizing full gradient computations. Nonetheless, this method proves impractical in deep learning applications, where obtaining full gradients is rarely feasible. Our focus moves to a framework that employs Stochastic Gradient Descent (SGD) while incorporating a more flexible and generalized data sampling strategy, enabling greater adaptability in the selection and utilization of data throughout the training process. General sampling techniques for strongly convex functions have been thoroughly examined in (Gower et al., 2019). For broader convex optimization problems, Khaled et al. (2023) provide a comprehensive study of how SGD performs under different sampling strategies. In non-convex scenarios, the works of Khaled & Richtárik (2023) and (Demidovich et al., 2023b) investigate the effects of generalized sampling methods on SGD 's convergence and efficiency, offering valuable insights into its adaptability for diverse machine learning applications. In this section we focus on Bernoulli-LoRA-SGD, a method, designed in the scope of Bernoulli-LoRA framework, based on the classical SGD algorithm.

For convergence analysis, we notice the gradient step in Algorithm 4 is equivalent to the following update

$$W^{t+1} = W^t - \gamma \hat{G}^t, \quad \text{where} \quad \hat{G}^t = \begin{cases} H_B^t G^t, & \text{with probability } p \\ G^t H_A^t, & \text{with probability } 1-p \end{cases}, \quad (102)$$

where $G^t = g(W^t)$ is an unbiased stochastic gradient, which satisfies Assumption 4.

### H.2.1 CONVERGENCE FOR SMOOTH NON-CONVEX FUNCTIONS

**Theorem 11.** *Let Assumptions 2, 3, and 4 hold, and stepsize satisfy*

$$0 < \gamma \leq \min \left\{ \frac{1}{\sqrt{LA_1 \lambda_{\max}^p T}}, \frac{1}{LB_1} \left( \frac{\lambda_{\max}^p}{\lambda_{\min}^p} \right)^{-1} \right\}.$$

*Then iterates generated by Bernoulli-LoRA-SGD (Algorithm 4) satisfy*

$$\mathbb{E}\left[ \left\| \nabla f(\widetilde{W}^T) \right\|_F^2 \right] \leq \frac{6(f(W^0) - f^*)}{\gamma \lambda_{\min}^p T} + \gamma LC_1 \frac{\lambda_{\max}^p}{\lambda_{\min}^p},$$

*where* $\lambda_{\min}^p := p\lambda_{\min}^{H_B} + (1-p)\lambda_{\min}^{H_A}$, $\lambda_{\max}^p := p\lambda_{\max}^{H_B} + (1-p)\lambda_{\max}^{H_A}$, *and* $\widetilde{W}^T$ *is chosen at random from* $\{W^0, W^1, \ldots, W^{T-1}\}$ *with probabilities* $\{\frac{w_t}{\mathcal{W}_{T-1}}\}_{t=0}^{T-1}$, *where* $w_t = \frac{w_{t-1}}{(1+\gamma^2 LA_1 \lambda_{\max}^p)}$, $\mathcal{W}_{T-1} = \sum_{t=0}^{T-1} w_t$, *and* $w^{-1} > 0$.

*Proof.* We start with smoothness of function $f$:

$$
\begin{aligned}
f(W^{t+1}) &\leq f(W^t) + \langle \nabla f(W^t), W^{t+1} - W^t \rangle + \frac{L}{2} \left\| W^{t+1} - W^t \right\|_F^2 \\
&\stackrel{(102)}{=} f(W^t) - \gamma \langle \nabla f(W^t), \hat{G}^t \rangle + \frac{\gamma^2 L}{2} \left\| \hat{G}^t \right\|_F^2.
\end{aligned} \tag{103}
$$

Taking a conditional expectation by $W^t$, we bound the second and the third terms from inequality (103):

$$
\begin{aligned}
\mathbb{E}\left[ \langle \nabla f(W^t), \hat{G}^t \rangle | W^t \right] &= \langle \nabla f(W^t), \mathbb{E}\left[ \hat{G}^t | W^t \right] \rangle \\
&\stackrel{(102)}{=} p \langle \nabla f(W^t), \mathbb{E}\left[ H_B^t G^t | W^t \right] \rangle + (1-p) \langle \nabla f(W^t), \mathbb{E}\left[ G^t H_A^t | W^t \right] \rangle \\
&\stackrel{(*)}{=} p \langle \nabla f(W^t), \mathbb{E}\left[ H_B^t | W^t \right] \mathbb{E}\left[ G^t | W^t \right] \rangle + (1-p) \langle \nabla f(W^t), \mathbb{E}\left[ G^t | W^t \right] \mathbb{E}\left[ H_A^t | W^t \right] \rangle \\
&= p \langle \nabla f(W^t), \mathbb{E}\left[ H_B^t | W^t \right] \nabla f(W^t) \rangle + (1-p) \langle \nabla f(W^t), \nabla f(W^t) \mathbb{E}\left[ H_A^t | W^t \right] \rangle \\
&\geq \underbrace{\left( p \lambda_{\min}(\mathbb{E}\left[ H_B^t \right]) + (1-p) \lambda_{\min}(\mathbb{E}\left[ H_A^t \right]) \right)}_{:= \lambda_{\min}^p} \left\| \nabla f(W^t) \right\|_F^2 \\
&= \lambda_{\min}^p \left\| \nabla f(W^t) \right\|_F^2,
\end{aligned} \tag{104}
$$

where in $(*)$ we used that $H_B^t$, $H_A^t$ and $G^t$ are independent. Now we bound the third term:

$$
\begin{aligned}
\mathbb{E}\left[ \left\| \hat{G}^t \right\|_F^2 | W^t \right] &\stackrel{(102)}{=} p \mathbb{E}\left[ \left\| H_B^t G^t \right\|_F^2 | W^t \right] + (1-p) \mathbb{E}\left[ \left\| G^t H_A^t \right\|_F^2 | W^t \right] \\
&= p \mathbb{E}\left[ \langle H_B^t G^t, H_B^t G^t \rangle | W^t \right] + (1-p) \mathbb{E}\left[ \langle G^t H_A^t, G^t H_A^t \rangle | W^t \right] \\
&\stackrel{(**)}{=} p \mathbb{E}\left[ \langle G^t, H_B^t G^t \rangle | W^t \right] + (1-p) \mathbb{E}\left[ \langle G^t, G^t H_A^t \rangle | W^t \right],
\end{aligned}
$$

where in $(**)$ we used property of projection matrices $H_B^t$, $H_B^t$. By the independence of $H_B^t$, $H_A^t$, $G^t$, we obtain

$$
\begin{aligned}
\mathbb{E}\left[ \left\| \hat{G}^t \right\|_F^2 | W^t \right] &= p \mathbb{E}\left[ \langle G^t, \mathbb{E}\left[ H_B^t | W^t \right] G^t \rangle | W^t \right] + (1-p) \mathbb{E}\left[ \langle G^t, G^t \mathbb{E}\left[ H_A^t | W^t \right] \rangle | W^t \right] \\
&\leq p \lambda_{\max}(\mathbb{E}\left[ H_B^t | W^t \right]) \mathbb{E}\left[ \left\| G^t \right\|_F^2 | W^t \right] + (1-p) \lambda_{\max}(\mathbb{E}\left[ H_A^t | W^t \right]) \mathbb{E}\left[ \left\| G^t \right\|_F^2 | W^t \right] \\
&= \underbrace{\left( p \lambda_{\max}(\mathbb{E}\left[ H_B^t | W^t \right]) + (1-p) \lambda_{\max}(\mathbb{E}\left[ H_A^t | W^t \right]) \right)}_{:= \lambda_{\max}^p} \mathbb{E}\left[ \left\| G^t \right\|_F^2 | W^t \right] \\
&= \lambda_{\max}^p \mathbb{E}\left[ \left\| G^t \right\|_F^2 | W^t \right].
\end{aligned} \tag{105}
$$

Plugging (104) and (105) into (103), we obtain

$$
\begin{aligned}
\mathbb{E}\left[ f(W^{t+1}) | W^t \right] &\leq f(W^t) - \gamma \mathbb{E}\left[ \langle \nabla f(W^t), \hat{G}^t \rangle | W^t \right] + \frac{\gamma^2 L}{2} \mathbb{E}\left[ \left\| \hat{G}^t \right\|_F^2 | W^t \right] \\
&\leq f(W^t) - \gamma \lambda_{\min}^p \left\| \nabla f(W^t) \right\|_F^2 + \frac{\gamma^2 \lambda_{\max}^p L}{2} \mathbb{E}\left[ \left\| G^t \right\|_F^2 | W^t \right].
\end{aligned}
$$

By Assumption 4,

$$
\begin{aligned}
\mathbb{E}\left[ f(W^{t+1}) - f^* | W^t \right] &\leq f(W^t) - \gamma \mathbb{E}\left[ \langle \nabla f(W^t), \hat{G}^t \rangle | W^t \right] + \frac{\gamma^2 L}{2} \mathbb{E}\left[ \left\| \hat{G}^t \right\|_F^2 | W^t \right] \\
&\leq f(W^t) - f^* - \gamma \lambda_{\min}^p \left\| \nabla f(W^t) \right\|_F^2 \\
&\quad + \frac{\gamma^2 \lambda_{\max}^p L}{2} \left( 2A_1(f(W^t) - f^*) + B_1 \left\| \nabla f(W^t) \right\|_F^2 + C_1 \right) \\
&\leq \left( 1 + \gamma^2 \lambda_{\max}^p L A_1 \right) \left( f(W^t) - f^* \right) - \gamma \lambda_{\min}^p \left( 1 - \frac{\gamma L B_1 \lambda_{\max}^p}{2 \lambda_{\min}^p} \right) \left\| \nabla f(W^t) \right\|_F^2 \\
&\quad + \frac{\gamma^2 \lambda_{\max}^p L C_1}{2}.
\end{aligned}
$$

Taking mathematical expectation and selecting a stepsize as $0 < \gamma \leq \frac{1}{LB_1}\left(\frac{\lambda_{\max}^p}{\lambda_{\min}^p}\right)^{-1}$, we get

$$
\begin{aligned}
\mathbb{E}\left[f(W^{t+1}) - f^*\right] &\leq \left(1 + \gamma^2 \lambda_{\max}^p LA_1\right)\mathbb{E}\left[f(W^t) - f^*\right] \\
&\quad - \frac{\gamma \lambda_{\min}^p}{2}\mathbb{E}\left[\left\|\nabla f(W^t)\right\|_F^2\right] + \frac{\gamma^2 \lambda_{\max}^p LC_1}{2}.
\end{aligned} \tag{106}
$$

Defining $\delta^t := \mathbb{E}\left[f(W^t) - f^*\right]$, $r^t := \mathbb{E}\left[\left\|\nabla f(W^t)\right\|_F^2\right]$ for every $t \geq 0$, we have

$$
\delta^{t+1} \leq \left(1 + \gamma^2 \lambda_{\max}^p LA_1\right)\delta^t - \frac{\gamma \lambda_{\min}^p}{2}r^t + \frac{\gamma^2 \lambda_{\max}^p LC_1}{2}.
$$

Fixing $w^{-1} > 0$ and defining $w_t = \frac{w_{t-1}}{1 + \gamma^2 LA_1 \lambda_{\max}^p}$ for all $t \geq 0$, we have

$$
\begin{aligned}
\frac{1}{2}\lambda_{\min}^p w_t r^t &\leq \frac{w_t}{\gamma}\left(1 + \gamma^2 \lambda_{\max}^p LA_1\right)\delta^t - \frac{w_t}{\gamma}\delta^{t+1} + \frac{1}{2}\gamma LC_1 \lambda_{\max}^p w_t \\
&= \frac{w_{t-1}\delta^t}{\gamma} - \frac{w_t \delta^{t+1}}{\gamma} + \frac{1}{2}\gamma LC_1 \lambda_{\max}^p w_t.
\end{aligned}
$$

Summing over $t$ from 0 to $T-1$, we have

$$
\sum_{t=0}^{T-1} w_t r^t \leq \frac{2w_{-1}\delta^0}{\gamma \lambda_{\min}^p} - \frac{2w_{T-1}\delta^T}{\gamma \lambda_{\min}^p} + \gamma LC_1 \frac{\lambda_{\max}^p}{\lambda_{\min}^p}\sum_{t=0}^{T-1} w_t.
$$

Defining $\mathcal{W}_{T-1} = \sum_{t=0}^{T-1} w_t$, we acquire

$$
\sum_{t=0}^{T-1}\frac{w_t}{\mathcal{W}^{T-1}}r^t \leq \frac{2w_{-1}\delta^0}{\gamma \lambda_{\min}^p \mathcal{W}_{T-1}} + \gamma LC_1 \frac{\lambda_{\max}^p}{\lambda_{\min}^p}.
$$

Using the next chain of inequalities

$$
W_{T-1} = \sum_{t=0}^{T-1} w_t \geq T \min_{0 \leq t \leq T-1} w_t = Tw_{T-1} = \frac{Tw_{-1}}{(1 + \gamma^2 \lambda_{\max}^p LA_1)^T},
$$

we have

$$
\sum_{t=0}^{T-1}\frac{w_t}{\mathcal{W}^{T-1}}r^t \leq \frac{2(1 + \gamma^2 \lambda_{\max}^p LA_1)^T}{\gamma T \lambda_{\min}^p}(f(W^0) - f^*) + \gamma LC_1 \frac{\lambda_{\max}^p}{\lambda_{\min}^p}.
$$

Selecting $0 < \gamma \leq \frac{1}{\sqrt{LA_1 \lambda_{\max}^p T}}$, and using $(1 + \gamma^2 \lambda_{\max}^p LA_1)^T \leq \exp\left(\gamma^2 \lambda_{\max}^p LA_1 T\right) \leq \exp(1) \leq 3$, we obtain

$$
\sum_{t=0}^{T-1}\frac{w_t}{\mathcal{W}^{T-1}}r^t \leq \frac{6\delta^0}{\gamma T \lambda_{\min}^p} + \gamma LC_1 \frac{\lambda_{\max}^p}{\lambda_{\min}^p}.
$$

$\square$

Next we show convergence of Bernoulli-LoRA-SGD under additional Assumption 6.

### H.2.2 CONVERGENCE UNDER POLYAK-ŁOJASIEWICZ CONDITION

**Theorem 12.** *Let Assumptions 2, 3, 4, and 6 hold, and stepsize satisfy*

$0 < \gamma \leq \min\left\{\frac{\mu \lambda_{\min}^p}{2LA_1 \lambda_{\max}^p}, \frac{2}{\mu \lambda_{\min}^p}, \frac{1}{LB_1}\left(\frac{\lambda_{\max}^p}{\lambda_{\min}^p}\right)^{-1}\right\}$. *Then iterates generated by* Bernoulli-LoRA-SGD *(Algorithm 4) satisfy*

$$
\mathbb{E}\left[f(W^T) - f^*\right] \leq \left(1 - \frac{1}{2}\gamma \mu \lambda_{\min}^p\right)^T\left(f(W^0) - f^*\right) + \frac{\gamma LC_1}{\mu}\cdot\frac{\lambda_{\max}^p}{\lambda_{\min}^p},
$$

*where* $\lambda_{\min}^p := p\lambda_{\min}^{H_B} + (1-p)\lambda_{\min}^{H_A}$, $\lambda_{\max}^p := p\lambda_{\max}^{H_B} + (1-p)\lambda_{\max}^{H_A}$.

*Proof.* We start our proof with inequality 106. Using PL-inequality (see Assumption 6), we have

$$
\begin{aligned}
\mathbb{E}\left[f(W^{t+1}) - f^*\right] &\leq \left(1 + \gamma^2 \lambda_{\max}^p LA_1\right)\mathbb{E}\left[f(W^t) - f^*\right] - \frac{\gamma \lambda_{\min}^p}{2}\mathbb{E}\left[\left\|\nabla f(W^t)\right\|_F^2\right] + \frac{\gamma^2 \lambda_{\max}^p LC_1}{2} \\
&\leq \left(1 - \gamma \mu \lambda_{\min}^p + \gamma^2 \lambda_{\max}^p LA_1\right)\mathbb{E}\left[f(W^t) - f^*\right] + +\frac{\gamma^2 \lambda_{\max}^p LC_1}{2}.
\end{aligned}
$$

Taking the stepsize as $0 < \gamma \le \min\left\{\frac{\mu\lambda_{\min}^p}{2LA_1\lambda_{\max}^p}, \frac{2}{\mu\lambda_{\min}^p}\right\}$, we obtain

$$
\begin{aligned}
\mathbb{E}\left[f(W^{t+1}) - f^*\right] &\le \left(1 - \frac{1}{2}\gamma\mu\lambda_{\min}^p\right)\mathbb{E}\left[f(W^t) - f^*\right] + \frac{\gamma^2\lambda_{\max}^p LC_1}{2} \\
&\le \left(1 - \frac{1}{2}\gamma\mu\lambda_{\min}^p\right)^{t+1}\mathbb{E}\left[f(W^0) - f^*\right] + \frac{\gamma^2\lambda_{\max}^p LC_1}{2}\sum_{\tau=0}^{t}\left(1 - \frac{1}{2}\gamma\mu\lambda_{\min}^p\right)^{t-\tau} \\
&\le \left(1 - \frac{1}{2}\gamma\mu\lambda_{\min}^p\right)^{t+1}\mathbb{E}\left[f(W^0) - f^*\right] + \frac{\gamma^2\lambda_{\max}^p LC_1}{2}\sum_{\tau=0}^{\infty}\left(1 - \frac{1}{2}\gamma\mu\lambda_{\min}^p\right)^{\tau} \\
&= \left(1 - \frac{1}{2}\gamma\mu\lambda_{\min}^p\right)^{t+1}\mathbb{E}\left[f(W^0) - f^*\right] + \frac{\gamma^2\lambda_{\max}^p LC_1}{\gamma\mu\lambda_{\min}^p},
\end{aligned}
$$

where in the last equation we use the formula of the sum of geometric progression. $\qquad\square$

### H.3 Analysis of Bernoulli-LoRA-MVR

---

**Algorithm 5** Bernoulli-LoRA-MVR

---

1: **Parameters:** pre-trained model $W^0 \in \mathbb{R}^{m \times n}$, $G^0 \in \mathbb{R}^{m \times n}$ rank $r \ll \min\{m,n\}$, scaling factor $\alpha > 0$, chain length $T$, sketch distribution $\mathcal{D}_S^B$ or $\mathcal{D}_S^A$, Bernoulli probability $p$, momentum parameter $b \in [0,1]$
2: **for** $t = 0, 1, \ldots, T-1$ **do**
3:      Sample $c^t \sim \mathrm{Be}(p)$            Bernoulli random variable
4:      **if** $c^t = 1$ **then**
5:          Sample $B_S^t \sim \mathcal{D}_S^B$           Left sketch
6:          $\hat{A}^t = -\eta \left( (B_S^t)^\top B_S^t \right)^\dagger (B_S^t)^\top G^t$
7:          $W^{t+1} = W^t + \frac{\alpha}{r} B_S^t \hat{A}^t$
8:      **else**
9:          Sample $A_S^t \sim \mathcal{D}_S^A$          Right sketch
10:        $\hat{B}^t = -\eta G^t (A_S^t)^\top \left( A_S^t (A_S^t)^\top \right)^\dagger$
11:        $W^{t+1} = W^t + \frac{\alpha}{r} \hat{B}^t A_S^t$
12:      **end if**
13:      Sample $\xi^{t+1} \sim \mathcal{D}$
14:      $G^{t+1} = \nabla f_{\xi^{t+1}}(W^{t+1}) + (1-b)\left( G^t - \nabla f_{\xi^{t+1}}(W^t) \right)$
15: **end for**

---

Recently, there has been a significant surge of interest in variance-reduced methods for addressing finite-sum problems (J Reddi et al., 2015; Shang et al., 2018; Malinovsky et al., 2022; Richtárik et al., 2024). It has gained prominence as a formidable alternative to stochastic gradient descent (SGD) in tackling non-convex optimization problems. Notably, it has been pivotal in introducing the first algorithms capable of surpassing SGD 's convergence rate for locating first-order critical points. Despite these advancements, variance reduction methods often come with challenges, including the necessity for meticulously tuned learning rates and the reliance on overly large batch sizes to realize their benefits. To address some of these limitations, Momentum Variance Reduction (MVR) was proposed specifically for server-only stochastic non-convex optimization (Cutkosky & Orabona, 2019). This approach leverages a modified form of momentum to achieve variance reduction while eliminating the dependence on large batch sizes. A proof on MVR technique with better dependence on momentum parameter was obtained by Tyurin & Richtárik (2023). In the context of Federated Learning, Karagulyan et al. (2024) proposed the SPAM method. On the server side, MVR is utilized to enhance optimization efficiency, while the client side incorporates the Stochastic Proximal Point Method updates. This section is devoted to Bernoulli-LoRA-MVR, a method, designed in the scope of Bernoulli-LoRA framework, based on the MVR technique.

To show convergence guarantees for Bernoulli-LoRA-MVR, the iterates of the method can be rewritten in following way

$$W^{t+1} \;=\; W^t - \gamma \hat{G}^t, \quad \text{where} \quad \hat{G}^t = \begin{cases} H_B^t G^t, & \text{with probability } p \\ G^t H_A^t, & \text{with probability } 1-p \end{cases} \tag{107}$$

$$G^{t+1} \;=\; \nabla f_{\xi^{t+1}}(W^{t+1}) + (1-b)\left( G^t - \nabla f_{\xi^{t+1}}(W^t) \right). \tag{108}$$

First of all, we reprove descent lemma from the paper of Li et al. (2021) for generic gradient step (107).

**Lemma 12.** *Let Assumptions 1, 3 hold. Then, iterates defined as* (107) *satisfy*

$$\mathbb{E}\left[ f(W^{t+1}) - f^* \mid W^t \right] \;\leq\; f(W^t) - f^* - \frac{\gamma \lambda_{\min}^p}{2} \left\| \nabla f(W^t) \right\|_{\mathrm{F}}^2$$
$$+ \frac{\gamma \lambda_{\max}^p}{2} \left\| G^t - \nabla f(W^t) \right\|_{\mathrm{F}}^2 - \left( \frac{1}{2\gamma} - \frac{L}{2} \right) \mathbb{E}\left[ \left\| W^{t+1} - W^t \right\|_{\mathrm{F}}^2 \mid W^t \right].$$

*Proof.* By Assumption 3, we have

$$f(W^{t+1}) \leq f(W^t) + \langle \nabla f(W^t), W^{t+1} - W^t \rangle_F + \frac{L}{2} \|W^{t+1} - W^t\|_F^2$$

$$= f(W^t) - \gamma \langle \nabla f(W^t), \hat{G}^t \rangle_F + \frac{L}{2} \|W^{t+1} - W^t\|_F^2. \qquad (109)$$

To continue our proof, we need to bound the second term from (109). Taking conditional expectation by $H^t, W^t$, we obtain

$$\mathbb{E}\left[ \langle \nabla f(W^t), \hat{G}^t \rangle_F \mid H^t, W^t \right] \overset{(107)}{=} p \langle \nabla f(W^t), H_B^t G^t \rangle_F + (1-p) \langle \nabla f(W^t), G^t H_A^t \rangle_F$$

$$= p \langle H_B^t \nabla f(W^t), H_B^t G^t \rangle_F + (1-p) \langle \nabla f(W^t) H_A^t, G^t H_A^t \rangle_F$$

$$= \frac{p}{2} \left( \|H_B^t \nabla f(W^t)\|_F^2 + \|H_B^t G^t\|_F^2 - \|H_B^t G^t - H_B^t \nabla f(W^t)\|_F^2 \right)$$

$$+ \frac{1-p}{2} \left( \|\nabla f(W^t) H_A^t\|_F^2 + \|G^t H_A^t\|_F^2 - \|G^t H_A^t - \nabla f(W^t) H_A^t\|_F^2 \right)$$

$$\geq \frac{1}{2} \left( p \|H_B^t \nabla f(W^t)\|_F^2 + (1-p) \|\nabla f(W^t) H_A^t\|_F^2 \right) + \frac{1}{2} \mathbb{E}\left[ \|\hat{G}^t\|_F^2 \mid H^t, W^t \right]$$

$$- \frac{1}{2} \left( p \|H_B^t G^t - H_B^t \nabla f(W^t)\|_F^2 + (1-p) \|G^t H_A^t - \nabla f(W^t) H_A^t\|_F^2 \right).$$

Taking conditional expectation by $W^t$, we have

$$\mathbb{E}\left[ \langle \nabla f(W^t), \hat{G}^t \rangle_F \mid W^t \right] \geq \frac{1}{2} \left( p \mathbb{E}\left[ \|H_B^t \nabla f(W^t)\|_F^2 \mid W^t \right] + (1-p) \mathbb{E}\left[ \|\nabla f(W^t) H_A^t\|_F^2 \mid W^t \right] \right) + \frac{1}{2} \mathbb{E}\left[ \|\hat{G}^t\|_F^2 \mid W^t \right]$$

$$- \frac{1}{2} \left( p \mathbb{E}\left[ \|H_B^t G^t - H_B^t \nabla f(W^t)\|_F^2 \mid W^t \right] + (1-p) \mathbb{E}\left[ \|G^t H_A^t - \nabla f(W^t) H_A^t\|_F^2 \mid W^t \right] \right)$$

$$\overset{(*)}{\geq} \frac{1}{2} \underbrace{\left( p \lambda_{\min}(\mathbb{E}[H_B^t]) + (1-p) \lambda_{\min}(\mathbb{E}[H_A^t]) \right)}_{:= \lambda_{\min}^p} \|\nabla f(W^t)\|_F^2 + \frac{1}{2} \mathbb{E}\left[ \|\hat{G}^t\|_F^2 \mid W^t \right]$$

$$- \frac{1}{2} \underbrace{\left( p \lambda_{\max}(\mathbb{E}[H_B^t]) + (1-p) \lambda_{\max}(\mathbb{E}[H_A^t]) \right)}_{:= \lambda_{\max}^p} \|G^t - \nabla f(W^t)\|_F^2$$

$$\overset{(107)}{=} \frac{\lambda_{\min}^p}{2} \|\nabla f(W^t)\|_F^2 + \frac{1}{2\gamma^2} \mathbb{E}\left[ \|W^{t+1} - W^t\|_F^2 \mid W^t \right] - \frac{\lambda_{\max}^p}{2} \|G^t - \nabla f(W^t)\|_F^2, \qquad (110)$$

where in $(*)$ we used the following inequalities for any matrix $V \in \mathbb{R}^{m \times n}$

$$\mathbb{E}\left[ \|H_B^t V\|_F^2 \right] = \mathbb{E}\left[ \langle H_B^t V, H_B^t V \rangle_F \right] = \langle \mathbb{E}[H_B^t] V, V \rangle_F \geq \lambda_{\min}(\mathbb{E}[H_B^t]) \|V\|_F^2,$$

$$\mathbb{E}\left[ \|H_B^t V\|_F^2 \right] \leq \lambda_{\max}(\mathbb{E}[H_B^t]) \|V\|_F^2,$$

$$\mathbb{E}\left[ \|V H_A^t\|_F^2 \right] = \mathbb{E}\left[ \langle V H_A^t, V H_A^t \rangle_F \right] = \langle V \mathbb{E}[H_A^t], V \rangle_F \geq \lambda_{\min}(\mathbb{E}[H_A^t]) \|V\|_F^2,$$

$$\mathbb{E}\left[ \|V H_A^t\|_F^2 \right] \leq \lambda_{\max}(\mathbb{E}[H_A^t]) \|V\|_F^2.$$

Plugging in (110) into (109), we get

$$\mathbb{E}\left[ f(W^{t+1}) \mid W^t \right] \leq f(W^t) - \frac{\gamma \lambda_{\min}^p}{2} \|\nabla f(W^t)\|_F^2 - \frac{1}{2\gamma} \mathbb{E}\left[ \|W^{t+1} - W^t\|_F^2 \mid W^t \right]$$

$$+ \frac{\gamma \lambda_{\max}^p}{2} \|G^t - \nabla f(W^t)\|_F^2 + \frac{L}{2} \mathbb{E}\left[ \|W^{t+1} - W^t\|_F^2 \mid W^t \right]. \qquad \square$$

**Lemma 13.** *Let Assumptions 3, 5 hold. Then, iterates generated by* Bernoulli-LoRA-MVR *(Algorithm 5) satisfy*

$$\mathbb{E}\left[ \|G^{t+1} - \nabla f(W^{t+1})\|_F^2 \right] \leq (1-b)^2 \mathbb{E}\left[ \|G^t - \nabla f(W^t)\|_F^2 \right] + 2(1-b)^2 L^2 \mathbb{E}\left[ \|W^{t+1} - W^t\|_F^2 \right] + 2b^2 \sigma^2$$

$$(111)$$

*Proof.* Taking conditional expectation by $\mathcal{F}^{t+1} = \{W^{t+1}, G^t\}$, we obtain

$$\mathbb{E}\left[\left\|G^{t+1} - \nabla f(W^{t+1})\right\|_F^2 | \mathcal{F}^{t+1}\right] \stackrel{(108)}{=} \mathbb{E}\left[\left\|\nabla f_{\xi^{t+1}}(W^{t+1}) - \nabla f(W^{t+1}) + (1-b)\left(G^t - \nabla f_{\xi^{t+1}}(W^t)\right)\right\|_F^2 | \mathcal{F}^{t+1}\right]$$

$$\stackrel{(13)}{=} (1-b)^2 \left\|G^t - \nabla f(W^t)\right\|_F^2$$

$$+ \mathbb{E}\left[\left\|\nabla f_{\xi^{t+1}}(W^{t+1}) - \nabla f(W^{t+1}) + (1-b)\left(\nabla f(W^t) - \nabla f_{\xi^{t+1}}(W^t)\right)\right\|_F^2 | \mathcal{F}^{t+1}\right]$$

$$\leq (1-b)^2 \left\|G^t - \nabla f(W^t)\right\|_F^2 + 2b^2 \mathbb{E}\left[\left\|\nabla f_{\xi^{t+1}}(W^{t+1}) - \nabla f(W^{t+1})\right\|_F^2 | \mathcal{F}^{t+1}\right]$$

$$+ 2(1-b)^2 \mathbb{E}\left[\left\|\nabla f_{\xi^{t+1}}(W^{t+1}) - \nabla f_{\xi^{t+1}}(W^t) - \nabla f(W^{t+1}) + \nabla f(W^t)\right\|_F^2 | \mathcal{F}^{t+1}\right]$$

$$\leq (1-b)^2 \left\|G^t - \nabla f(W^t)\right\|_F^2 + 2b^2 \mathbb{E}\left[\left\|\nabla f_{\xi^{t+1}}(W^{t+1}) - \nabla f(W^{t+1})\right\|_F^2 | \mathcal{F}^{t+1}\right]$$

$$+ 2(1-b)^2 \mathbb{E}\left[\left\|\nabla f_{\xi^{t+1}}(W^{t+1}) - \nabla f_{\xi^{t+1}}(W^t)\right\|_F^2 | \mathcal{F}^{t+1}\right]$$

$$\leq (1-b)^2 \left\|G^t - \nabla f(W^t)\right\|_F^2 + 2(1-b)^2 L^2 \left\|W^{t+1} - W^t\right\|_F^2 + 2b^2\sigma^2,$$

where in the last inequality we used smoothness of $f_\xi$ and bounded variance assumption. Taking math expectation, we conclude the proof. $\qquad\square$

### H.3.1 CONVERGENCE FOR SMOOTH NON-CONVEX FUNCTIONS

**Theorem 13.** *Let Assumptions 1, 2, 3, and 5 hold, and let the stepsize satisfy $0 < \gamma \leq \frac{1}{L\left(1+\sqrt{\frac{2\lambda_{\max}^p(1-b)^2}{b}}\right)}$. Then the iterates of* Bernoulli-LoRA-MVR *(Algorithm 5) satisfy*

$$\mathbb{E}\left[\left\|\nabla f(\widetilde{W}^T)\right\|_F^2\right] \leq \frac{2(f(W^0) - f^*)}{\lambda_{\min}^p \gamma T} + \frac{\left\|G^0 - \nabla f(W^0)\right\|_F^2}{b(2-b)T} \cdot \frac{\lambda_{\max}^p}{\lambda_{\min}^p} + \frac{2b\sigma^2}{2-b} \cdot \frac{\lambda_{\max}^p}{\lambda_{\min}^p}, \quad (112)$$

*where $\lambda_{\min}^p := p\lambda_{\min}^{H_B} + (1-p)\lambda_{\min}^{H_A}$, $\lambda_{\max}^p := p\lambda_{\max}^{H_B} + (1-p)\lambda_{\max}^{H_A}$, $\widetilde{W}^T$ is drawn uniformly at random from the iterate sequence $\{W^0, W^1, \ldots, W^{T-1}\}$.*

*Proof.* Denote Lyapunov function $\Phi_t$ as follows

$$\Phi_t = f(W^t) - f^* + \frac{\gamma\lambda_{\max}^p}{2b(2-b)}\left\|G^t - \nabla f(W^t)\right\|_F^2. \quad (113)$$

By Lemma 12 and Lemma 13, we have

$$\mathbb{E}[\Phi_{t+1}] \leq \mathbb{E}[f(W^t)] - f^* - \frac{\gamma\lambda_{\min}^p}{2}\mathbb{E}\left[\left\|\nabla f(W^t)\right\|_F^2\right] - \left(\frac{1}{2\gamma} - \frac{L}{2}\right)\mathbb{E}\left[\left\|W^{t+1} - W^t\right\|_F^2\right]$$

$$+ \frac{\gamma\lambda_{\max}^p}{2}\mathbb{E}\left[\left\|G^t - \nabla f(W^t)\right\|_F^2\right] + \frac{\gamma(1-b)^2\lambda_{\max}^p}{2b(2-b)}\mathbb{E}\left[\left\|G^t - \nabla f(W^t)\right\|_F^2\right]$$

$$+ \frac{\gamma(1-b)^2 L^2\lambda_{\max}^p}{2b(2-b)}\mathbb{E}\left[\left\|W^{t+1} - W^t\right\|_F^2\right] + \frac{\gamma\lambda_{\max}^p b\sigma^2}{2-b}$$

$$\leq \mathbb{E}[\Phi_t] - \frac{\gamma\lambda_{\min}^p}{2}\mathbb{E}\left[\left\|\nabla f(W^t)\right\|_F^2\right] + \frac{\gamma\lambda_{\max}^p b\sigma^2}{2-b}$$

$$- \left(\frac{1}{2\gamma} - \frac{L}{2} - \frac{\gamma(1-b)^2 L^2\lambda_{\max}^p}{2b(2-b)}\right)\mathbb{E}\left[\left\|W^{t+1} - W^t\right\|_F^2\right].$$

Selecting $0 < \gamma \leq \frac{1}{L\left(1+\sqrt{\frac{(1-b)^2}{b(2-b)}\lambda_{\max}^p}\right)}$, we obtain

$$\mathbb{E}[\Phi_{t+1}] \leq \mathbb{E}[\Phi_t] - \frac{\gamma\lambda_{\min}^p}{2}\mathbb{E}\left[\left\|\nabla f(W^t)\right\|_F^2\right] + \frac{\gamma\lambda_{\max}^p b\sigma^2}{2-b}.$$

Summing over $t$ from 0 to $T-1$, we get

$$\frac{\gamma\lambda_{\min}^p}{2}\sum_{t=0}^{T-1}\mathbb{E}\left[\left\|\nabla f(W^t)\right\|_F^2\right] \leq \mathbb{E}[\Phi_0] - \mathbb{E}[\Phi_T] + \frac{\gamma\lambda_{\max}^p b\sigma^2}{2-b}T.$$

Finally, dividing both sides by $\frac{\gamma\lambda_{\min}^p}{2}$ yields

$$\mathbb{E}\left[\left\|\nabla f(\widetilde{W}^T)\right\|_F^2\right] \leq \frac{2\Phi_0}{\lambda_{\min}^p \gamma T} + \frac{2b\sigma^2}{2-b} \cdot \frac{\lambda_{\max}^p}{\lambda_{\min}^p},$$

where $\widetilde{W}^T$ is drawn uniformly at random from the iterate sequence $\{W^0, W^1, \dots, W^{T-1}\}$. $\qquad\square$

Next we show convergence guarantee for Bernoulli-LoRA-MVR, supposing additionally Assumption 6 holds.

### H.3.2 CONVERGENCE UNDER POLYAK-ŁOJASIEWICZ CONDITION

**Theorem 14.** *Let Assumptions 1, 2, 3, 5, and 6 hold, and let the stepsize satisfy*

$$0 < \gamma \leq \min\left\{ \frac{1}{L\left(1 + \sqrt{\frac{2(1-b)^2}{b(2-b)}\lambda_{\max}^p}\right)}, \frac{b}{2\mu\lambda_{\min}^p} \right\}.$$

*Then the iterates of Bernoulli-LoRA-MVR (Algorithm 5) satisfy*

$$\mathbb{E}\left[f(W^T) - f^*\right] \leq (1 - \gamma\mu\lambda_{\min}^p)^T \Phi_0 + \frac{b\sigma^2}{(2-b)\mu} \cdot \frac{\lambda_{\max}^p}{\lambda_{\min}^p}, \tag{114}$$

*where $\lambda_{\min}^p := p\lambda_{\min}^{H_B} + (1-p)\lambda_{\min}^{H_A}$, $\lambda_{\max}^p := p\lambda_{\max}^{H_B} + (1-p)\lambda_{\max}^{H_A}$, and $\Phi_0 = f(W^0) - f^* + \frac{\gamma\lambda_{\max}^p}{b(2-b)}\left\|G^0 - \nabla f(W^0)\right\|_{\mathrm{F}}^2$.*

*Proof.* Denote Lyapunov function $\Phi_t$ as follows

$$\Phi_t = f(W^t) - f^* + \frac{\gamma\lambda_{\max}^p}{b(2-b)}\left\|G^t - \nabla f(W^t)\right\|_{\mathrm{F}}^2. \tag{115}$$

By Lemma 12 and Lemma 13, we have

$$
\begin{aligned}
\mathbb{E}\left[\Phi_{t+1}\right] &\leq \mathbb{E}\left[f(W^t)\right] - f^* - \frac{\gamma\lambda_{\min}^p}{2}\mathbb{E}\left[\left\|\nabla f(W^t)\right\|_{\mathrm{F}}^2\right] - \left(\frac{1}{2\gamma} - \frac{L}{2}\right)\mathbb{E}\left[\left\|W^{t+1} - W^t\right\|_{\mathrm{F}}^2\right] \\
&\quad + \frac{\gamma\lambda_{\max}^p}{2}\mathbb{E}\left[\left\|G^t - \nabla f(W^t)\right\|_{\mathrm{F}}^2\right] + \frac{\gamma(1-b)^2\lambda_{\max}^p}{b(2-b)}\mathbb{E}\left[\left\|G^t - \nabla f(W^t)\right\|_{\mathrm{F}}^2\right] \\
&\quad + \frac{\gamma(1-b)^2 L^2\lambda_{\max}^p}{b(2-b)}\mathbb{E}\left[\left\|W^{t+1} - W^t\right\|_{\mathrm{F}}^2\right] + \frac{\gamma\lambda_{\max}^p b\sigma^2}{2-b} \\
&\leq \max\left\{1 - \gamma\mu\lambda_{\min}^p, 1 - \frac{b}{2}\right\}\mathbb{E}\left[\Phi_t\right] + \frac{\gamma\lambda_{\max}^p b\sigma^2}{2-b} \\
&\quad - \left(\frac{1}{2\gamma} - \frac{L}{2} - \frac{\gamma(1-b)^2 L^2\lambda_{\max}^p}{b(2-b)}\right)\mathbb{E}\left[\left\|W^{t+1} - W^t\right\|_{\mathrm{F}}^2\right],
\end{aligned}
$$

where in the last inequality we used Assumption 6. Selecting positive stepsize $\gamma$ satisfying the upper bound assumed in the theorem statement, we obtain

$$
\begin{aligned}
\mathbb{E}\left[\Phi_{t+1}\right] &\leq (1 - \gamma\mu\lambda_{\min}^p)\mathbb{E}\left[\Phi_t\right] + \frac{\gamma\lambda_{\max}^p b\sigma^2}{2-b} \\
&\leq (1 - \gamma\mu\lambda_{\min}^p)^{t+1}\mathbb{E}\left[\Phi_0\right] + \frac{\gamma\lambda_{\max}^p b\sigma^2}{2-b}\sum_{\tau=0}^{t}(1 - \gamma\mu\lambda_{\min}^p)^{t-\tau} \\
&\leq (1 - \gamma\mu\lambda_{\min}^p)^{t+1}\mathbb{E}\left[\Phi_0\right] + \frac{\gamma\lambda_{\max}^p b\sigma^2}{2-b}\sum_{\tau=0}^{\infty}(1 - \gamma\mu\lambda_{\min}^p)^{\tau} \\
&= (1 - \gamma\mu\lambda_{\min}^p)^{t+1}\mathbb{E}\left[\Phi_0\right] + \frac{\gamma\lambda_{\max}^p b\sigma^2}{(2-b)\gamma\mu\lambda_{\min}^p},
\end{aligned}
$$

where, in the last equation, we used the formula for the sum of a geometric progression.

$\qquad\square$

## H.4 ANALYSIS OF BERNOULLI-LORA-PAGE

---

**Algorithm 6** Bernoulli-LoRA-PAGE

1: **Parameters:** pre-trained model $W^0 \in \mathbb{R}^{m \times n}$, a vector $G^0 \in\in \mathbb{R}^{m \times n}$, rank $r \ll \min\{m,n\}$, scaling factor $\alpha > 0$, chain length $T$, sketch distribution $\mathcal{D}_S^B$ or $\mathcal{D}_S^A$, Bernoulli probability $p$, probability $q$
2: **for** $t = 0, 1, \ldots, T-1$ **do**
3:    Sample $c^t \sim \text{Be}(p)$          Bernoulli random variable
4:    **if** $c^t = 1$ **then**
5:       Sample $B_S^t \sim \mathcal{D}_S^B$          Left sketch
6:       $\hat{A}^t = -\eta \left( (B_S^t)^\top B_S^t \right)^\dagger (B_S^t)^\top G^t$
7:       $W^{t+1} = W^t + \frac{\alpha}{r} B_S^t \hat{A}^t$
8:    **else**
9:       Sample $A_S^t \sim \mathcal{D}_S^A$          Right sketch
10:       $\hat{B}^t = -\eta g(W^t) (A_S^t)^\top \left( A_S^t (A_S^t)^\top \right)^\dagger A_S^t$
11:       $W^{t+1} = W^t + \frac{\alpha}{r} \hat{B}^t A_S^t$
12:    **end if**
13:    Sample $i_{t+1}$ uniformly at random from $[n]$
14:    $G^{t+1} = \begin{cases} \nabla f(W^{t+1}), & \text{with probability } q \\ G^t + \left( \nabla f_{i_{t+1}}(W^{t+1}) - \nabla f_{i_{t+1}}(W^t) \right), & \text{with probability } 1-q \end{cases}$
15: **end for**

---

There exist several optimal methods for solving a general non-convex optimization problem, e.g. SPIDER (Fang et al., 2018) and SARAH (Pham et al., 2020). However, the known lower bound used to establish their optimality works only in the small data regime. ProbAbilistic Gradient Estimator (PAGE) (Li et al., 2021) is a very simple and easy to implement algorithm, known for achieving optimal convergence results in non-convex optimization. PAGE uses the full gradient update with probability $q_t$, or reuses the previous gradient with a small adjustment (at a low computational cost) with probability $1 - q_t$. A general version of PAGE on Riemannian manifolds is considered in (Demidovich et al., 2024a). In this section we present Bernoulli-LoRA-PAGE, a new method within Bernoulli-LoRA framework, based on PAGE algorithm.

Notice, that the iterates of Bernoulli-LoRA-PAGE (Algorithm 6) can be rewritten in the following simple way

$$W^{t+1} = W^t - \gamma \hat{G}^t, \quad \text{where} \quad \hat{G}^t = \begin{cases} H_B^t G^t, & \text{with probability } p \\ G^t H_A^t, & \text{with probability } 1-p \end{cases} \tag{116}$$

$$G^{t+1} = \begin{cases} \nabla f(W^{t+1}), & \text{with probability } q \\ G^t + \left( \nabla f_{i_{t+1}}(W^{t+1}) - \nabla f_{i_{t+1}}(W^t) \right), & \text{with probability } 1-q \end{cases} \tag{117}$$

**Lemma 14.** *Let Assumption 3 hold. Then, iterates generated by* Bernoulli-LoRA-PAGE

$$\mathbb{E}\left[ \left\| G^{t+1} - \nabla f(W^{t+1}) \right\|_F^2 \right] \leq (1-q)\mathbb{E}\left[ \left\| G^t - \nabla f(W^t) \right\|_F^2 \right] + (1-q)L^2 \mathbb{E}\left[ \left\| W^{t+1} - W^t \right\|_F^2 \right]. \tag{118}$$

*Proof.* Taking the full mathematical expectation, we obtain

$$\mathbb{E}\left[\left\|G^{t+1} - \nabla f(W^{t+1})\right\|_{\mathrm{F}}^2\right] \overset{(117)}{=} (1-q)\mathbb{E}\left[\left\|G^t - \nabla f(W^{t+1}) + \left(\nabla f_{i_{t+1}}(W^{t+1}) - \nabla f_{i_{t+1}}(W^t)\right)\right\|_{\mathrm{F}}^2\right]$$

$$\overset{(13)}{=} (1-q)\mathbb{E}\left[\left\|G^t - \nabla f(W^t)\right\|_{\mathrm{F}}^2\right]$$

$$+ (1-q)\mathbb{E}\left[\left\|\left(\nabla f_{i_{t+1}}(W^{t+1}) - \nabla f_{i_{t+1}}(W^t)\right) - \left(\nabla f(W^{t+1}) - \nabla f(W^t)\right)\right\|_{\mathrm{F}}^2\right]$$

$$\leq (1-q)\mathbb{E}\left[\left\|G^t - \nabla f(W^t)\right\|_{\mathrm{F}}^2\right]$$

$$+ (1-q)\mathbb{E}\left[\left\|\nabla f_{i_{t+1}}(W^{t+1}) - \nabla f_{i_{t+1}}(W^t)\right\|_{\mathrm{F}}^2\right]$$

$$\leq (1-q)\mathbb{E}\left[\left\|G^t - \nabla f(W^t)\right\|_{\mathrm{F}}^2\right] + (1-q)L^2\mathbb{E}\left[\left\|W^{t+1} - W^t\right\|_{\mathrm{F}}^2\right],$$

where in the last inequality we used smoothness of each $f_i$. □

### H.4.1 CONVERGENCE FOR SMOOTH NON-CONVEX FUNCTIONS

**Theorem 15.** *Let Assumptions 1, 2, and 3 hold, and let the stepsize satisfy*

$$0 < \gamma \leq \frac{1}{L\left(1 + \sqrt{\frac{1-q}{q}\lambda_{\max}^p}\right)}.$$

*Then the iterates of* PAGE-Bernoulli-LoRA *(Algorithm 6) satisfy*

$$\mathbb{E}\left[\left\|\nabla f(\widetilde{W}^T)\right\|_{\mathrm{F}}^2\right] \leq \frac{2(f(W^0) - f^*)}{\lambda_{\min}^p \gamma T} + q\frac{\left\|G^0 - \nabla f(W^0)\right\|_{\mathrm{F}}^2}{T} \cdot \frac{\lambda_{\max}^p}{\lambda_{\min}^p}, \quad (119)$$

*where $\lambda_{\min}^p := p\lambda_{\min}^{H_B} + (1-p)\lambda_{\min}^{H_A}$, $\lambda_{\max}^p := p\lambda_{\max}^{H_B} + (1-p)\lambda_{\max}^{H_A}$, $\widetilde{W}^T$ is drawn uniformly at random from the iterate sequence $\{W^0, W^1, \ldots, W^{T-1}\}$.*

*Proof.* Denote Lyapunov function $\Phi_t$ as follows

$$\Phi_t = f(W^t) - f^* + \frac{\gamma\lambda_{\max}^p}{2q}\left\|G^t - \nabla f(W^t)\right\|_{\mathrm{F}}^2. \quad (120)$$

By Lemma 12 and Lemma 14, we have

$$\mathbb{E}\left[\Phi_{t+1}\right] \leq \mathbb{E}\left[f(W^t)\right] - f^* - \frac{\gamma\lambda_{\min}^p}{2}\mathbb{E}\left[\left\|\nabla f(W^t)\right\|_{\mathrm{F}}^2\right]$$

$$- \left(\frac{1}{2\gamma} - \frac{L}{2}\right)\mathbb{E}\left[\left\|W^{t+1} - W^t\right\|_{\mathrm{F}}^2\right] + \frac{\gamma\lambda_{\max}^p}{2}\mathbb{E}\left[\left\|G^t - \nabla f(W^t)\right\|_{\mathrm{F}}^2\right]$$

$$+ \frac{\gamma\lambda_{\max}^p(1-q)}{2q}\mathbb{E}\left[\left\|G^t - \nabla f(W^t)\right\|_{\mathrm{F}}^2\right] + \frac{\gamma\lambda_{\max}^p(1-q)L^2}{2q}\mathbb{E}\left[\left\|W^{t+1} - W^t\right\|_{\mathrm{F}}^2\right]$$

$$\leq \mathbb{E}\left[\Phi_t\right] - \frac{\gamma\lambda_{\min}^p}{2}\mathbb{E}\left[\left\|\nabla f(W^t)\right\|_{\mathrm{F}}^2\right] - \left(\frac{1}{2\gamma} - \frac{L}{2} - \frac{\gamma(1-q)L^2\lambda_{\max}^p}{2q}\right)\mathbb{E}\left[\left\|W^{t+1} - W^t\right\|_{\mathrm{F}}^2\right].$$

Selecting $0 < \gamma \leq \frac{1}{L\left(1+\sqrt{\frac{1-q}{q}\lambda_{\max}^p}\right)}$, we obtain

$$\mathbb{E}\left[\Phi_{t+1}\right] \leq \mathbb{E}\left[\Phi_t\right] - \frac{\gamma\lambda_{\min}^p}{2}\mathbb{E}\left[\left\|\nabla f(W^t)\right\|_{\mathrm{F}}^2\right].$$

Summing over $t$ from 0 to $T-1$, we get

$$\frac{\gamma\lambda_{\min}^p}{2}\sum_{t=0}^{T-1}\mathbb{E}\left[\left\|\nabla f(W^t)\right\|_{\mathrm{F}}^2\right] \leq \mathbb{E}\left[\Phi_0\right] - \mathbb{E}\left[\Phi_T\right].$$

Finally, dividing both sides by $\frac{\gamma\lambda_{\min}^p}{2}$ yields

$$\mathbb{E}\left[\left\|\nabla f(\widetilde{W}^T)\right\|_{\mathrm{F}}^2\right] \leq \frac{2\Phi_0}{\gamma\lambda_{\min}^p T}.$$

where $\widetilde{W}^T$ is drawn uniformly at random from the iterate sequence $\{W^0, W^1, \ldots, W^{T-1}\}$. □

### H.4.2 CONVERGENCE UNDER POLYAK-ŁOJASIEWICZ CONDITION

**Theorem 16.** *Let Assumptions 1, 2, 3, and 6 hold, and let the stepsize satisfy*

$$0 < \gamma \le \min\left\{ \frac{1}{L\left(1 + 2\sqrt{\frac{1-q}{q}\lambda_{\max}^p}\right)}, \frac{q}{2\mu\lambda_{\min}^p} \right\}.$$

*Then the iterates of* Bernoulli-LoRA-PAGE *(Algorithm 6) satisfy*

$$\mathbb{E}\left[f(W^T) - f^*\right] \le (1 - \gamma\mu\lambda_{\min}^p)^T \Phi_0, \tag{121}$$

*where* $\lambda_{\min}^p := p\lambda_{\min}^{H_B} + (1-p)\lambda_{\min}^{H_A}$, *and* $\Phi_0 = f(W^0) - f^* + \frac{\gamma\lambda_{\max}^p}{q}\left\|G^0 - \nabla f(W^0)\right\|_{\mathrm{F}}^2$.

*Proof.* Denote Lyapunov function $\Phi_t$ as follows

$$\Phi_t = f(W^t) - f^* + \frac{\gamma\lambda_{\max}^p}{q}\left\|G^t - \nabla f(W^t)\right\|_{\mathrm{F}}^2. \tag{122}$$

By Lemma 12 and Lemma 14, we have

$$\begin{aligned}
\mathbb{E}\left[\Phi_{t+1}\right] &\le& \mathbb{E}\left[f(W^t)\right] - f^* - \frac{\gamma\lambda_{\min}^p}{2}\mathbb{E}\left[\left\|\nabla f(W^t)\right\|_{\mathrm{F}}^2\right] - \left(\frac{1}{2\gamma} - \frac{L}{2}\right)\mathbb{E}\left[\left\|W^{t+1} - W^t\right\|_{\mathrm{F}}^2\right] \\
&& + \frac{\gamma\lambda_{\max}^p}{2}\mathbb{E}\left[\left\|G^t - \nabla f(W^t)\right\|_{\mathrm{F}}^2\right] + \frac{\gamma(1-q)\lambda_{\max}^p}{q}\mathbb{E}\left[\left\|G^t - \nabla f(W^t)\right\|_{\mathrm{F}}^2\right] \\
&& + \frac{\gamma(1-q)L^2\lambda_{\max}^p}{q}\mathbb{E}\left[\left\|W^{t+1} - W^t\right\|_{\mathrm{F}}^2\right] \\
&\le& (1 - \gamma\mu\lambda_{\min}^p)\mathbb{E}\left[f(W^t) - f^*\right] + \left(1 - \frac{q}{2}\right)\frac{\gamma\lambda_{\max}^p}{q}\mathbb{E}\left[\left\|G^t - \nabla f(W^t)\right\|_{\mathrm{F}}^2\right] \\
&& - \left(\frac{1}{2\gamma} - \frac{L}{2} - \frac{\gamma(1-q)L^2\lambda_{\max}^p}{q}\right)\mathbb{E}\left[\left\|W^{t+1} - W^t\right\|_{\mathrm{F}}^2\right],
\end{aligned}$$

where in the last inequality we used Assumption 6. Selecting $0 < \gamma \le \min\left\{ \frac{1}{L\left(1 + 2\sqrt{\frac{1-q}{q}\lambda_{\max}^p}\right)}, \frac{q}{2\mu\lambda_{\min}^p} \right\}$, we obtain

$$\mathbb{E}\left[\Phi_{t+1}\right] \le (1 - \gamma\mu\lambda_{\min}^p)\mathbb{E}\left[\Phi_t\right].$$

Unrolling the recursion, we obtain

$$\mathbb{E}\left[\Phi_T\right] \le (1 - \gamma\mu\lambda_{\min}^p)^T \Phi_0.$$

$\square$

# I    PROOFS FOR FEDERATED LEARNING EXTENSIONS

In recent years, distributed optimization problems and algorithms have become a focal point in the Machine Learning (ML) community. This surge in interest is driven by the need to train modern deep neural networks, which often involve billions of parameters and massive datasets (Brown et al., 2020; Kolesnikov et al., 2020). To achieve practical training times (Li, 2020), parallelizing computations, such as stochastic gradient evaluations, has emerged as a natural solution, leading to the widespread adoption of distributed training algorithms (Goyal et al., 2017; You et al., 2019; Le Scao et al., 2023). Additionally, distributed methods are crucial when data is inherently distributed across multiple devices or clients, often accompanied by privacy constraints—a common scenario in Federated Learning (FL) (Konečný et al., 2016; McMahan et al., 2016; Kairouz et al., 2019; Demidovich et al., 2024b; Sadiev et al., 2024; Yi et al., 2024).

We develop several FL methods within the Bernoulli-LoRA framework and provide a convergence analysis for them.

## I.1    ANALYSIS OF FED-BERNOULLI-LORA-QGD

---

**Algorithm 7** Fed-Bernoulli-LoRA-QGD

1: **Parameters:** pre-trained model $W^0 \in \mathbb{R}^{m \times n}$, rank $r \ll \min\{m,n\}$, scaling factor $\alpha > 0$, chain length $T$, sketch distribution $\mathcal{D}_S^B$ or $\mathcal{D}_S^A$, Bernoulli probabilities $p$ and $q$
2: **for** $t = 0, 1, \ldots, T-1$ **do**
3:     **for** any client $l \in [M]$ in parallel **do**
4:         Compute gradient $\nabla f_l(W^{t+1})$ and send compressed version $G_l^t = \mathcal{Q}_l^t\left(\nabla f_l(W^{t+1})\right)$ to the server
5:     **end for**
6:     $G^t = \frac{1}{M} \sum\limits_{l=1}^{M} G_l^t$
7:     Sample $c^t \sim \mathrm{Be}(p)$                                            Bernoulli random variable
8:     **if** $c^t = 1$ **then**
9:         Sample $B_S^t \sim \mathcal{D}_S^B$                                             Left sketch
10:         $\hat{A}^t = -\eta \left(\left(B_S^t\right)^\top B_S^t\right)^\dagger \left(B_S^t\right)^\top G^t$
11:         $W^{t+1} = W^t + \frac{\alpha}{r} B_S^t \hat{A}^t$
12:     **else**
13:         Sample $A_S^t \sim \mathcal{D}_S^A$                                           Right sketch
14:         $\hat{B}^t = -\eta G^t \left(A_S^t\right)^\top \left(A_S^t \left(A_S^t\right)^\top\right)^\dagger$
15:         $W^{t+1} = W^t + \frac{\alpha}{r} \hat{B}^t A_S^t$
16:     **end if**
17:     Broadcast $W^{t+1}$ to each client $l \in [M]$
18: **end for**

---

Parallel implementations of SGD have become a prominent area of study due to their impressive scalability. However, one of the primary challenges in parallelizing SGD lies in the substantial communication overhead required to exchange gradient updates across nodes. To address this, numerous lossy compression techniques have been developed, enabling nodes to transmit quantized gradients instead of full gradients. While these methods often work well in practice, they are not universally reliable and may fail to ensure convergence.

To overcome these limitations, Quantized SGD (QSGD) by Alistarh et al. (2017) introduces a family of compression techniques that provide both theoretical convergence guarantees and strong empirical performance. QSGD offers a flexible mechanism for balancing communication bandwidth and convergence speed. By adjusting the number of bits transmitted per iteration, nodes can reduce bandwidth usage, albeit at the potential cost of increased variance in the gradient estimates. Different variants of QSGD were considered by Horváth et al. (2022); Wen et al. (2017); Panferov et al. (2024).

We consider the following distributed optimization problem:

$$\min_{W \in \mathbb{R}^{m \times n}} \frac{1}{M} \sum_{l=1}^{M} f_l(W),$$

where $M$ represents the number of clients. In Federated Learning, a primary bottleneck is the communication overhead between clients and the central server. A common approach to mitigate this issue is communication compression.

**Definition 2.** *A randomized operator $\mathcal{Q} : \mathbb{R}^{m \times n} \to \mathbb{R}^{m \times n}$ is called an unbiased compression operator (or compressor) if there exists a constant $\omega > 0$ such that, for any matrix $W \in \mathbb{R}^{m \times n}$, the following conditions hold:*

$$\mathbb{E}[\mathcal{Q}(W)] = W, \quad and \quad \mathbb{E}\left[\|\mathcal{Q}(W) - W\|_{\mathrm{F}}^2\right] \le \omega \|W\|_{\mathrm{F}}^2. \tag{123}$$

To analyze the optimization process, we introduce the following assumption regarding function dissimilarity:

**Assumption 11.** *Let $f^* := \inf_W f(W)$ and $f_l^* := \inf_W f_l$ for each $l \in [M]$. In the non-convex case, the difference at the optimum is defined as:*

$$\Delta^* := f^* - \frac{1}{M} \sum_{l=1}^{M} f_l^* \ge 0. \tag{124}$$

This assumption quantifies the discrepancy between the global optimal function value and the average of the local optimal function values between the clients.

To start convergence analysis, we rewrite the updates for $W^t$ and $G^t$ generated by Fed-Bernoulli-LoRA-QGD (Algorithm 7) as follows

$$G^t = \frac{1}{M} \sum_{l=1}^{M} \mathcal{Q}_l^t \left( \nabla f_l(W^t) \right); \tag{125}$$

$$W^{t+1} = W^t - \gamma \hat{G}^t, \quad \text{where} \quad \hat{G}^t = \begin{cases} H_B^t G^t, & \text{with probability } p \\ G^t H_A^t, & \text{with probability } 1 - p \end{cases}. \tag{126}$$

To establish the convergence guarantee for Fed-Bernoulli-LoRA-QGD (Algorithm 7), we first demonstrate that the gradient estimator $G^t$ satisfies Assumption 4. Once this is verified, the convergence rate follows directly using the same reasoning as in the proof of Theorem 2.

**Lemma 15.** *Let Assumptions 2, 3, and 11 hold. Then, $G^t$ defined in Algorithm 7 (see (125)) satisfies Assumption 4 with the following constants:*

$$A_1 = \frac{L\omega}{M}, \quad B_1 = 1, \quad C_1 = 2\frac{L\omega\Delta^*}{M}.$$

*Proof.* First, we show $G^t$ is an unbiased estimator of $\nabla f(W^t)$:

$$\mathbb{E}\left[G^t | W^t\right] = \frac{1}{M} \sum_{l=1}^{M} \mathbb{E}\left[\mathcal{Q}_l^t \left(\nabla f_l(W^t)\right) | W^t\right] \overset{(123)}{=} \frac{1}{M} \sum_{l=1}^{M} \nabla f_l(W^t) = \nabla f(W^t).$$

Now we establish that $G^t$ satisfies Assumption 4. Taking the conditional expectation with respect to $W^t$, we have

$$
\begin{aligned}
\mathbb{E}\left[\|G^t\|_F^2 \,|W^t\right] &= \mathbb{E}\left[\left\|\frac{1}{M}\sum_{l=1}^M \mathcal{Q}_l^t\left(\nabla f_l(W^t)\right) - \nabla f(W^t) + \nabla f(W^t)\right\|_F^2 \,|W^t\right] \\
&\overset{(13)}{=} \mathbb{E}\left[\left\|\frac{1}{M}\sum_{l=1}^M \mathcal{Q}_l^t\left(\nabla f_l(W^t)\right) - \nabla f(W^t)\right\|_F^2 \,|W^t\right] + \|\nabla f(W^t)\|_F^2 \\
&= \frac{1}{M^2}\sum_{l=1}^M \mathbb{E}\left[\left\|\mathcal{Q}_l^t\left(\nabla f_l(W^t)\right) - \nabla f_l(W^t)\right\|_F^2 \,|W^t\right] + \|\nabla f(W^t)\|_F^2 \\
&\overset{(123)}{\leq} \frac{\omega}{M^2}\sum_{l=1}^M \|\nabla f_l(W^t)\|_F^2 + \|\nabla f(W^t)\|_F^2 \\
&\overset{(*)}{\leq} \frac{2L\omega}{M^2}\sum_{l=1}^M \left(f_l(W^t) - f_l^*\right) + \|\nabla f(W^t)\|_F^2 \\
&= 2\frac{L\omega}{M}\left(f(W^t) - f^*\right) + \|\nabla f(W^t)\|_F^2 + 2\frac{L\omega}{M}\underbrace{\left(f^* - \frac{1}{M}\sum_{l=1}^M f_l^*\right)}_{:=\Delta^*},
\end{aligned}
$$

where in $(*)$ we used smoothness of each $f_l$ Thus, we have shown that $G^t$ satisfies Assumption 4 with following constants

$$
A_1 = \frac{L\omega}{M}, \quad B_1 = 1, \quad C_1 = 2\frac{L\omega\Delta^*}{M}.
$$

$\square$

### I.1.1 CONVERGENCE FOR SMOOTH NON-CONVEX FUNCTIONS

**Theorem 17.** *Let Assumptions 1 2, and 3 hold, and stepsize satisfy*

$$
0 < \gamma \leq \min\left\{\frac{1}{L\sqrt{\frac{\omega}{M}\lambda_{\max}^p T}}, \frac{1}{L}\left(\frac{\lambda_{\max}^p}{\lambda_{\min}^p}\right)^{-1}\right\}.
$$

*Then iterates generated by Fed-Bernoulli-LoRA-QGD (Algorithm 7) satisfy*

$$
\mathbb{E}\left[\left\|\nabla f(\widetilde{W}^T)\right\|_F^2\right] \leq \frac{6(f(W^0) - f^*)}{\gamma\lambda_{\min}^p T} + \frac{2\gamma L\omega\Delta^*}{M}\frac{\lambda_{\max}^p}{\lambda_{\min}^p},
$$

*where $\lambda_{\min}^p := p\lambda_{\min}^{H_B} + (1-p)\lambda_{\min}^{H_A}$, $\lambda_{\max}^p := p\lambda_{\max}^{H_B} + (1-p)\lambda_{\max}^{H_A}$, and $\widetilde{W}^T$ is chosen at random from $\left\{W^0, W^1, \ldots, W^{T-1}\right\}$ with probabilities $\left\{\frac{w_t}{\mathcal{W}_{T-1}}\right\}_{t=0}^{T-1}$, where $w_t = \frac{w_{t-1}}{(1+\gamma^2 L^2\lambda_{\max}^p\omega/M)}$, $\mathcal{W}_{T-1} = \sum_{t=0}^{T-1} w_t$, and $w^{-1} > 0$.*

*Proof.* By Lemma 15, and Theorem 2, we directly obtain the statement of the theorem. $\square$

### I.1.2 CONVERGENCE UNDER POLYAK-ŁOJASIEWICZ CONDITION

**Theorem 18.** *Let Assumptions 1, 2, 3, and 6 hold, and stepsize satisfy*

$$
0 < \gamma \leq \min\left\{\frac{\mu}{2L^2\omega/M}\left(\frac{\lambda_{\max}^p}{\lambda_{\min}^p}\right)^{-1}, \frac{2}{\mu\lambda_{\min}^p}, \frac{1}{L}\left(\frac{\lambda_{\max}^p}{\lambda_{\min}^p}\right)^{-1}\right\}.
$$

*Then iterates generated by Fed-Bernoulli-LoRA-QGD (Algorithm 7) satisfy*

$$
\mathbb{E}\left[f(W^T) - f^*\right] \leq \left(1 - \frac{1}{2}\gamma\mu\lambda_{\min}^p\right)^T\left(f(W^0) - f^*\right) + \frac{2\gamma L^2}{\mu}\cdot\frac{\omega}{M}\cdot\frac{\lambda_{\max}^p}{\lambda_{\min}^p},
$$

*where $\lambda_{\min}^p := p\lambda_{\min}^{H_B} + (1-p)\lambda_{\min}^{H_A}$, $\lambda_{\max}^p := p\lambda_{\max}^{H_B} + (1-p)\lambda_{\max}^{H_A}$.*

*Proof.* By Lemma 15, and Theorem 12, we directly obtain the statement of the theorem. $\square$

## I.2 ANALYSIS OF FED-BERNOULLI-LORA-MARINA

---

**Algorithm 8** Fed-Bernoulli-LoRA-MARINA

---

1: **Parameters:** pre-trained model $W^0 \in \mathbb{R}^{m \times n}$, $\{G_l^0\}_{l \in [M]} \in \mathbb{R}^{m \times n}$ rank $r \ll \min\{m,n\}$, scaling factor $\alpha > 0$, chain length $T$, sketch distribution $\mathcal{D}_S^B$ or $\mathcal{D}_S^A$, Bernoulli probabilities $p$ and $q$
2: **for** $t = 0, 1, \ldots, T-1$ **do**
3:     Sample $c^t \sim \text{Be}(p)$                             Bernoulli random variable
4:     **if** $c^t = 1$ **then**
5:         Sample $B_S^t \sim \mathcal{D}_S^B$                          Left sketch
6:         $\hat{A}^t = -\eta \left( (B_S^t)^\top B_S^t \right)^\dagger (B_S^t)^\top G^t$
7:         $W^{t+1} = W^t + \frac{\alpha}{r} B_S^t \hat{A}^t$
8:     **else**
9:         Sample $A_S^t \sim \mathcal{D}_S^A$                        Right sketch
10:        $\hat{B}^t = -\eta G^t (A_S^t)^\top \left( A_S^t (A_S^t)^\top \right)^\dagger$
11:        $W^{t+1} = W^t + \frac{\alpha}{r} \hat{B}^t A_S^t$
12:     **end if**
13:     Broadcast $W^{t+1}$ to each client $l \in [M]$
14:     Sample $s^t \sim \text{Be}(q)$
15:     **for** any client $l \in [M]$ in parallel **do**
16:         Compute gradient $\nabla f_l(W^{t+1})$
17:         $G_l^{t+1} = \begin{cases} \nabla f_l(W^{t+1}), & \text{with probability } q \\ G_l^t + \mathcal{Q}_l^t \left( \nabla f_l(W^{t+1}) - \nabla f_l(W^t) \right), & \text{with probability } 1-q \end{cases}$
18:         Send $G_l^{t+1}$ to the server
19:     **end for**
20:     $G^{t+1} = \frac{1}{M} \sum_{l=1}^M G_l^{t+1}$
21: **end for**

---

MARINA (Gorbunov et al., 2021) is an advanced method that significantly enhances communication efficiency in non-convex distributed learning across heterogeneous datasets. Its core innovation lies in a communication reduction mechanism that compresses the differences between gradients. The communication complexity bounds for MARINA are known to be better than those of all previous first-order methods. Non-smooth convex analysis of MARINA with different stepsize strategies can be found in (Sokolov & Richtárik, 2024). This section is devoted to Fed-Bernoulli-LoRA-MARINA (Algorithm 8), a method within the Bernoulli-LoRA framework, based on MARINA algorithm.

In order to start convergence analysis, we rewrite the updates $W^t, G^t$ generated by Fed-Bernoulli-LoRA-MARINA (Algorithm 8):

$$W^{t+1} = W^t - \gamma \hat{G}^t, \quad \text{where} \quad \hat{G}^t = \begin{cases} H_B^t G^t, & \text{with probability } p \\ G^t H_A^t, & \text{with probability } 1-p \end{cases} \tag{127}$$

$$G_l^{t+1} = \begin{cases} \nabla f_l(W^{t+1}), & \text{with probability } q \\ G_l^t + \mathcal{Q}_l^t \left( \nabla f_l(W^{t+1}) - \nabla f_l(W^t) \right), & \text{with probability } 1-q \end{cases} \tag{128}$$

$$G^{t+1} = \frac{1}{M} \sum_{l=1}^M G_l^{t+1}. \tag{129}$$

**Lemma 16.** *Let Assumption 3 hold. Then iterates generated by* Fed-Bernoulli-LoRA-MARINA *satisfy*

$$\mathbb{E}\left[ \left\| G^{t+1} - \nabla f(W^{t+1}) \right\|_F^2 \right] \leq (1-q)\mathbb{E}\left[ \left\| G^t - \nabla f(W^t) \right\|_F^2 \right] + (1-q)\frac{\omega L^2}{M} \mathbb{E}\left[ \left\| W^{t+1} - W^t \right\|_F^2 \right]. \tag{130}$$

*Proof.* Taking the conditional expectation with respect to $W^{t+1}$ and defining $D_l^{t+1} := \nabla f_l(W^{t+1}) - \nabla f_l(W^t)$, $D^{t+1} = \frac{1}{M} \sum_{l=1}^M D_l^{t+1}$, we obtain

$$
\begin{aligned}
\mathbb{E}\left[\left\|G^{t+1} - \nabla f(W^{t+1})\right\|_{\mathrm{F}}^2 | W^{t+1}\right] &= (1-q)\mathbb{E}\left[\left\|G^t - \nabla f(W^t) + \frac{1}{M}\sum_{l=1}^M \mathcal{Q}_l^t\left(\nabla f_l(W^{t+1}) - \nabla f_l(W^t)\right)\right\|_{\mathrm{F}}^2 | W^{t+1}\right] \\
&\stackrel{(13)}{=} (1-q)\left\|G^t - \nabla f(W^t)\right\|_{\mathrm{F}}^2 + (1-q)\mathbb{E}\left[\left\|\frac{1}{M}\sum_{l=1}^M \mathcal{Q}_l^t\left(D_l^{t+1}\right) - D^{t+1}\right\|_{\mathrm{F}}^2 | W^{t+1}\right] \\
&= (1-q)\left\|G^t - \nabla f(W^t)\right\|_{\mathrm{F}}^2 + \frac{1-q}{M^2}\sum_{m=1}^M \mathbb{E}\left[\left\|\mathcal{Q}_l^t\left(D_l^{t+1}\right) - D_l^{t+1}\right\|_{\mathrm{F}}^2 | W^{t+1}\right] \\
&\stackrel{(123)}{\leq} (1-q)\left\|G^t - \nabla f(W^t)\right\|_{\mathrm{F}}^2 + \frac{(1-q)\omega}{M^2}\sum_{l=1}^M \left\|\nabla f_l(W^{t+1}) - \nabla f_l(W^t)\right\|_{\mathrm{F}}^2 \\
&\leq (1-q)\left\|G^t - \nabla f(W^t)\right\|_{\mathrm{F}}^2 + \frac{(1-q)\omega L^2}{M}\left\|W^{t+1} - W^t\right\|_{\mathrm{F}}^2,
\end{aligned}
$$

where in the last inequality we used that the gradient of each $f_l$ is Lipschitz continuous. $\square$

### I.2.1 CONVERGENCE FOR SMOOTH NON-CONVEX FUNCTIONS

**Theorem 19.** *Let Assumptions 1, 2, 3, and hold, and let the stepsize satisfy*

$$
0 < \gamma \leq \frac{1}{L\left(1 + \sqrt{\lambda_{\max}^p \frac{1-q}{q} \cdot \frac{\omega}{M}}\right)}.
$$

*Then the iterates of* Fed-Bernoulli-LoRA-MARINA *(Algorithm 8) satisfy*

$$
\mathbb{E}\left[\left\|\nabla f(\widetilde{W}^T)\right\|_{\mathrm{F}}^2\right] \leq \frac{2\left(f(W^0) - f^*\right)}{\gamma\lambda_{\min}^p T} + \frac{\left\|G^0 - \nabla f(W^0)\right\|_{\mathrm{F}}^2}{qT} \cdot \frac{\lambda_{\max}^p}{\lambda_{\min}^p}, \tag{131}
$$

*where $\lambda_{\min}^p := p\lambda_{\min}^{H_B} + (1-p)\lambda_{\min}^{H_A}$, $\lambda_{\max}^p := p\lambda_{\max}^{H_B} + (1-p)\lambda_{\max}^{H_A}$, and $\widetilde{W}^T$ is drawn uniformly at random from the iterate sequence $\{W^0, W^1, \ldots, W^{T-1}\}$.*

*Proof.* Denote Lyapunov function $\Phi_t$ as follows

$$
\Phi_t = f(W^t) - f^* + \frac{\gamma\lambda_{\max}^p}{2q}\left\|G^t - \nabla f(W^t)\right\|_{\mathrm{F}}^2. \tag{132}
$$

By Lemma 12 and Lemma 16, we have

$$
\begin{aligned}
\mathbb{E}\left[\Phi_{t+1}\right] &\leq \mathbb{E}\left[f(W^t)\right] - f^* - \frac{\gamma\lambda_{\min}^p}{2}\mathbb{E}\left[\left\|\nabla f(W^t)\right\|_{\mathrm{F}}^2\right] - \left(\frac{1}{2\gamma} - \frac{L}{2}\right)\mathbb{E}\left[\left\|W^{t+1} - W^t\right\|_{\mathrm{F}}^2\right] \\
&\quad + \frac{\gamma\lambda_{\max}^p}{2}\mathbb{E}\left[\left\|G^t - \nabla f(W^t)\right\|_{\mathrm{F}}^2\right] + \frac{\gamma(1-q)\lambda_{\max}^p}{2q}\mathbb{E}\left[\left\|G^t - \nabla f(W^t)\right\|_{\mathrm{F}}^2\right] \\
&\quad + \frac{\gamma(1-q)L^2\omega\lambda_{\max}^p}{2qM}\mathbb{E}\left[\left\|W^{t+1} - W^t\right\|_{\mathrm{F}}^2\right] \\
&\leq \mathbb{E}\left[\Phi_t\right] - \frac{\gamma\lambda_{\min}^p}{2}\mathbb{E}\left[\left\|\nabla f(W^t)\right\|_{\mathrm{F}}^2\right] - \left(\frac{1}{2\gamma} - \frac{L}{2} - \frac{\gamma(1-q)L^2\omega\lambda_{\max}^p}{2qM}\right)\mathbb{E}\left[\left\|W^{t+1} - W^t\right\|_{\mathrm{F}}^2\right].
\end{aligned}
$$

Selecting $0 < \gamma \leq \frac{1}{L\left(1 + \sqrt{\lambda_{\max}^p \frac{1-q}{q} \cdot \frac{\omega}{M}}\right)}$, we obtain

$$
\mathbb{E}\left[\Phi_{t+1}\right] \leq \mathbb{E}\left[\Phi_t\right] - \frac{\gamma\lambda_{\min}^p}{2}\mathbb{E}\left[\left\|\nabla f(W^t)\right\|_{\mathrm{F}}^2\right].
$$

Summing over, we get

$$
\frac{\gamma\lambda_{\min}^p}{2}\sum_{t=0}^{T-1}\mathbb{E}\left[\left\|\nabla f(W^t)\right\|_{\mathrm{F}}^2\right] \leq \mathbb{E}\left[\Phi_0\right] - \mathbb{E}\left[\Phi_T\right].
$$

Finally, we derive

$$
\mathbb{E}\left[\left\|\nabla f(\widetilde{W}^T)\right\|_{\mathrm{F}}^2\right] \leq \frac{2\Phi_0}{\lambda_{\min}^p \gamma T}.
$$

where $\widetilde{W}^T$ is drawn uniformly at random from the iterate sequence $\{W^0, W^1, \ldots, W^{T-1}\}$. $\square$

### I.2.2 CONVERGENCE UNDER POLYAK-ŁOJASIEWICZ CONDITION

**Theorem 20.** *Let Assumptions 1, 2, 3, and 6 hold, and let the stepsize satisfy*

$$0 < \gamma \leq \min \left\{ \frac{1}{L \left( 1 + \sqrt{2\lambda_{\max}^p \frac{1-q}{q} \cdot \frac{\omega}{M}} \right)}, \frac{q}{2\mu \lambda_{\min}^p} \right\}.$$

*Then the iterates of* Fed-Bernoulli-LoRA-MARINA *(Algorithm 8) satisfy*

$$\mathbb{E}\left[ f(W^T) - f^* \right] \leq (1 - \gamma\mu\lambda_{\min}^p)^T \Phi_0, \tag{133}$$

*where* $\lambda_{\min}^p := p\lambda_{\min}^{H_B} + (1-p)\lambda_{\min}^{H_A}$, $\lambda_{\max}^p := p\lambda_{\max}^{H_B} + (1-p)\lambda_{\max}^{H_A}$, *and* $\Phi_0 = f(W^0) - f^* + \frac{\gamma\lambda_{\max}^p}{q} \left\| G^0 - \nabla f(W^0) \right\|_{\mathrm{F}}^2$.

*Proof.* Denote Lyapunov function $\Phi_t$ as follows

$$\Phi_t = f(W^t) - f^* + \frac{\gamma\lambda_{\max}^p}{q} \left\| G^t - \nabla f(W^t) \right\|_{\mathrm{F}}^2. \tag{134}$$

By Lemma 12 and Lemma 14, we have

$$\begin{aligned}
\mathbb{E}\left[ \Phi_{t+1} \right] &\leq \mathbb{E}\left[ f(W^t) \right] - f^* - \frac{\gamma\lambda_{\min}^p}{2} \mathbb{E}\left[ \left\| \nabla f(W^t) \right\|_{\mathrm{F}}^2 \right] - \left( \frac{1}{2\gamma} - \frac{L}{2} \right) \mathbb{E}\left[ \left\| W^{t+1} - W^t \right\|_{\mathrm{F}}^2 \right] \\
&\quad + \frac{\gamma\lambda_{\max}^p}{2} \mathbb{E}\left[ \left\| G^t - \nabla f(W^t) \right\|_{\mathrm{F}}^2 \right] + \frac{\gamma(1-q)\lambda_{\max}^p}{q} \mathbb{E}\left[ \left\| G^t - \nabla f(W^t) \right\|_{\mathrm{F}}^2 \right] \\
&\quad + \frac{\gamma(1-q)L^2\lambda_{\max}^p}{q} \cdot \frac{\omega}{M} \mathbb{E}\left[ \left\| W^{t+1} - W^t \right\|_{\mathrm{F}}^2 \right] \\
&\leq (1 - \gamma\mu\lambda_{\min}^p)\mathbb{E}\left[ f(W^t) - f^* \right] + \left( 1 - \frac{q}{2} \right) \frac{\gamma\lambda_{\max}^p}{q} \mathbb{E}\left[ \left\| G^t - \nabla f(W^t) \right\|_{\mathrm{F}}^2 \right] \\
&\quad - \left( \frac{1}{2\gamma} - \frac{L}{2} - \frac{\gamma(1-q)L^2\lambda_{\max}^p}{q} \cdot \frac{\omega}{M} \right) \mathbb{E}\left[ \left\| W^{t+1} - W^t \right\|_{\mathrm{F}}^2 \right],
\end{aligned}$$

where in the last inequality we used Assumption 6. Selecting $0 < \gamma \leq \min \left\{ \frac{1}{L \left( 1 + \sqrt{\frac{2(1-q)\omega}{qM} \lambda_{\max}^p} \right)}, \frac{q}{2\mu\lambda_{\min}^p} \right\}$, we obtain

$$\mathbb{E}\left[ \Phi_{t+1} \right] \leq (1 - \gamma\mu\lambda_{\min}^p)\mathbb{E}\left[ \Phi_t \right].$$

Taking recursion, we have

$$\mathbb{E}\left[ \Phi_T \right] \leq (1 - \gamma\mu\lambda_{\min}^p)^T \Phi_0.$$

$\square$

## I.3   ANALYSIS OF FED-BERNOULLI-LORA-EF21

---

**Algorithm 9** Fed-Bernoulli-LoRA-EF21

---

1: **Parameters:** pre-trained model $W^0 \in \mathbb{R}^{m \times n}$, $\{G_l^0\}_{l \in [M]} \in \mathbb{R}^{m \times n}$ rank $r \ll \min\{m,n\}$, scaling factor $\alpha > 0$, chain length $T$, sketch distribution $\mathcal{D}_S^B$ or $\mathcal{D}_S^A$, Bernoulli probability $p$
2: **for** $t = 0, 1, \ldots, T-1$ **do**
3:     Sample $c^t \sim \text{Be}(p)$                                                          Bernoulli random variable
4:     **if** $c^t = 1$ **then**
5:         Sample $B_S^t \sim \mathcal{D}_S^B$                                                      Left sketch
6:         $\hat{A}^t = -\eta \left( \left(B_S^t\right)^\top B_S^t \right)^\dagger \left(B_S^t\right)^\top G^t$
7:         $W^{t+1} = W^t + \frac{\alpha}{r} B_S^t \hat{A}^t$
8:     **else**
9:         Sample $A_S^t \sim \mathcal{D}_S^A$                                                      Right sketch
10:         $\hat{B}^t = -\eta G^t \left(A_S^t\right)^\top \left( A_S^t \left(A_S^t\right)^\top \right)^\dagger$
11:         $W^{t+1} = W^t + \frac{\alpha}{r} \hat{B}^t A_S^t$
12:     **end if**
13:     Broadcast $W^{t+1}$ to each client $l \in [M]$
14:     **for** any client $l \in [M]$ in parallel **do**
15:         Compute gradient $\nabla f_l(W^{t+1})$
16:         $G_l^{t+1} = G_l^t + \mathcal{C}_l^t \left( \nabla f_l(W^{t+1}) - G_l^t \right)$
17:         Send $G_l^{t+1}$ to the server
18:     **end for**
19:     $G^{t+1} = \frac{1}{M} \sum_{l=1}^{M} G_l^{t+1}$
20: **end for**

---

Error Feedback (EF) (Seide et al., 2014; Stich et al., 2018; Alistarh et al., 2018; Richtárik et al., 2021; Fatkhullin et al., 2021; Richtárik et al., 2022; Khirirat et al., 2024), often referred to as error compensation, is an exceptionally influential mechanism for stabilizing convergence in distributed training of supervised machine learning models, particularly when contractive communication compression techniques are employed. We design Fed-Bernoulli-LoRA-EF21 within the Bernoulli-LoRA framework, based on EF-21 method. Our theoretical analysis, built on standard assumptions, applies to distributed training in heterogeneous data settings and achieves the best known convergence rates.

Compared to Fed-Bernoulli-LoRA-MARINA, in this section we work with the wider class of compression operators called contractive.

**Definition 3.** *A randomized operator* $\mathcal{C} : \mathbb{R}^{m \times n} \to \mathbb{R}^{m \times n}$ *is called a contractive compression operator (compressor) if it satisfies the following condition: there exists a constant* $0 < \beta \leq 1$ *such that*

$$\mathbb{E}\left[ \left\| \mathcal{C}(W) - W \right\|_F^2 \right] \leq (1 - \beta) \left\| W \right\|_F^2, \quad \forall W \in \mathbb{R}^{m \times n}. \tag{135}$$

The iterates of Fed-Bernoulli-LoRA-EF21 can be rewritten as follows

$$W^{t+1} = W^t - \gamma \hat{G}^t, \quad \text{where} \quad \hat{G}^t = \begin{cases} H_B^t G^t, & \text{with probability } p \\ G^t H_A^t, & \text{with probability } 1-p \end{cases} \tag{136}$$

$$G_l^{t+1} = G_l^t + \mathcal{C}_l^t \left( \nabla f_l(W^{t+1}) - G_l^t \right), \quad \forall l \in [M] \tag{137}$$

$$G^{t+1} = \frac{1}{M} \sum_{l=1}^{M} G_l^{t+1}. \tag{138}$$

**Lemma 17.** *Let Assumption 3 hold. Then for the iterates generated by* Fed-Bernoulli-LoRA-EF21 *(Algorithm 9)satisfy*

$$\mathbb{E}\left[ \left\| G_l^{t+1} - \nabla f_l(W^{t+1}) \right\|_F^2 \right] \leq \sqrt{1-\beta}\, \mathbb{E}\left[ \left\| G_l^t - \nabla f_l(W^t) \right\|_F^2 \right] + \frac{(1-\beta)L^2}{1-\sqrt{1-\beta}} \mathbb{E}\left[ \left\| W^{t+1} - W^t \right\|_F^2 \right]$$

*Proof.* For each $l \in [M]$ we have

$$\mathbb{E}\left[\left\|G_l^{t+1} - \nabla f_l(W^{t+1})\right\|_F^2\right] \overset{(137),(138)}{=} \mathbb{E}\left[\mathbb{E}\left[\left\|\mathcal{C}_l^t\left(\nabla f_l(W^{t+1}) - G_l^t\right) - \left(\nabla f_l(W^{t+1}) - G_l^t\right)\right\|_F^2 \mid G_l^{t+1}, W^{t+1}\right]\right]$$

$$\overset{(135)}{\leq} (1-\beta)\,\mathbb{E}\left[\left\|G_l^t - \nabla f_l(W^{t+1})\right\|_F^2\right]$$

$$\leq (1-\beta)(1+\theta)\,\mathbb{E}\left[\left\|G_l^t - \nabla f_l(W^t)\right\|_F^2\right]$$

$$+ (1-\beta)\left(1+\frac{1}{\theta}\right)\mathbb{E}\left[\left\|\nabla f_l(W^{t+1}) - \nabla f_l(W^t)\right\|_F^2\right],$$

where in the last inequality we used $\|U+V\|_F^2 \leq (1+\theta)\|U\|_F^2 + \left(1+\frac{1}{\theta}\right)\|V\|_F^2$ for any constant $\theta > 0$, and matrices $U, V \in \mathbb{R}^{m \times n}$. Taking $\theta = \frac{1}{\sqrt{1-\beta}} - 1$, we acquire

$$\mathbb{E}\left[\left\|G_l^{t+1} - \nabla f_l(W^{t+1})\right\|_F^2\right] \leq \sqrt{1-\beta}\,\mathbb{E}\left[\left\|G_l^t - \nabla f_l(W^t)\right\|_F^2\right] + \frac{1-\beta}{1-\sqrt{1-\beta}}\mathbb{E}\left[\left\|\nabla f_l(W^{t+1}) - \nabla f_l(W^t)\right\|_F^2\right]$$

$$\leq \sqrt{1-\beta}\,\mathbb{E}\left[\left\|G_l^t - \nabla f_l(W^t)\right\|_F^2\right] + \frac{(1-\beta)L^2}{1-\sqrt{1-\beta}}\mathbb{E}\left[\left\|W^{t+1} - W^t\right\|_F^2\right],$$

where in the last inequality we used that the gradient of each $f_l$ is Lipschitz continuous. Summing over $l$ from 1 to $M$, we finish the proof. $\square$

### I.3.1 CONVERGENCE FOR SMOOTH NON-CONVEX FUNCTIONS

**Theorem 21.** *Let Assumptions 1, 2, and 3 hold, and let the stepsize satisfy*

$$0 < \gamma \leq \frac{1}{L\left(1 + \frac{\sqrt{\lambda_{\max}^p(1-\beta)}}{1-\sqrt{1-\beta}}\right)}.$$

*Then the iterates of* Fed-Bernoulli-LoRA-EF21 *(Algorithm 9) satisfy*

$$\mathbb{E}\left[\left\|\nabla f(\widetilde{W}^T)\right\|_F^2\right] \leq \frac{2(f(W^0) - f^*)}{\gamma \lambda_{\min}^p T} + \frac{\mathcal{G}^0}{(1-\sqrt{1-\beta})T} \cdot \frac{\lambda_{\max}^p}{\lambda_{\min}^p}, \tag{139}$$

*where* $\lambda_{\min}^p := p\lambda_{\min}^{H_B} + (1-p)\lambda_{\min}^{H_A}$, *and* $\lambda_{\max}^p := p\lambda_{\max}^{H_B} + (1-p)\lambda_{\max}^{H_A}$, $\widetilde{W}^T$ *is drawn uniformly at random from the iterate sequence* $\{W^0, W^1, \ldots, W^{T-1}\}$, *and* $\mathcal{G}^0 := \frac{1}{M}\sum_{l=1}^M \left\|G_l^0 - \nabla f_l(W^0)\right\|_F^2$.

*Proof.* Denote Lyapunov function $\Phi_t$ as follows

$$\Phi_t = f(W^t) - f^* + \frac{\gamma \lambda_{\max}^p}{2(1-\sqrt{1-\beta})} \cdot \frac{1}{M}\sum_{l=1}^M \left\|G_l^t - \nabla f_l(W^t)\right\|_F^2. \tag{140}$$

By Lemma 12 and Lemma 17, we have

$$\mathbb{E}\left[\Phi_{t+1}\right] \leq \mathbb{E}\left[f(W^t)\right] - f^* - \frac{\gamma \lambda_{\min}^p}{2}\mathbb{E}\left[\left\|\nabla f(W^t)\right\|_F^2\right] - \left(\frac{1}{2\gamma} - \frac{L}{2}\right)\mathbb{E}\left[\left\|W^{t+1} - W^t\right\|_F^2\right]$$

$$+ \frac{\gamma \lambda_{\max}^p}{2}\mathbb{E}\left[\left\|G^t - \nabla f(W^t)\right\|_F^2\right] + \frac{\gamma \lambda_{\max}^p \sqrt{1-\beta}}{2(1-\sqrt{1-\beta})} \cdot \frac{1}{M}\sum_{l=1}^M \mathbb{E}\left[\left\|G_l^t - \nabla f_l(W^t)\right\|_F^2\right]$$

$$+ \frac{\gamma \lambda_{\max}^p L^2(1-\beta)}{2(1-\sqrt{1-\beta})^2}\mathbb{E}\left[\left\|W^{t+1} - W^t\right\|_F^2\right]$$

$$\leq \mathbb{E}\left[\Phi_t\right] - \frac{\gamma \lambda_{\min}^p}{2}\mathbb{E}\left[\left\|\nabla f(W^t)\right\|_F^2\right] - \left(\frac{1}{2\gamma} - \frac{L}{2} - \frac{\gamma \lambda_{\max}^p L^2(1-\beta)}{2(1-\sqrt{1-\beta})^2}\right)\mathbb{E}\left[\left\|W^{t+1} - W^t\right\|_F^2\right].$$

Selecting $0 < \gamma \leq \frac{1}{L\left(1 + \frac{\sqrt{\lambda_{\max}^p(1-\beta)}}{1-\sqrt{1-\beta}}\right)}$, we obtain

$$\mathbb{E}\left[\Phi_{t+1}\right] \leq \mathbb{E}\left[\Phi_t\right] - \frac{\gamma \lambda_{\min}^p}{2}\mathbb{E}\left[\left\|\nabla f(W^t)\right\|_F^2\right].$$

Summing over $t$ from 0 to $T-1$, we get

$$\frac{\gamma \lambda_{\min}^p}{2}\sum_{t=0}^{T-1}\mathbb{E}\left[\left\|\nabla f(W^t)\right\|_F^2\right] \leq \mathbb{E}\left[\Phi_0\right] - \mathbb{E}\left[\Phi_T\right].$$

Finally, dividing both sides by $\frac{\gamma \lambda_{\min}^p}{2}$ yields

$$\mathbb{E}\left[\left\|\nabla f(\widetilde{W}^T)\right\|_F^2\right] \leq \frac{2\Phi_0}{\gamma \lambda_{\min}^p T}.$$

where $\widetilde{W}^T$ is drawn uniformly at random from the iterate sequence $\{W^0, W^1, \ldots, W^{T-1}\}$. $\qquad\square$

### I.3.2 Convergence under Polyak-Łojasiewicz Condition

**Theorem 22.** *Let Assumptions 1, 2, 3, and 6 hold, and let the stepsize satisfy*

$$0 < \gamma \leq \min\left\{ \frac{1}{L\left(1 + \frac{\sqrt{2\lambda_{\max}^p(1-\beta)}}{1-\sqrt{1-\beta}}\right)}, \frac{1+\sqrt{1-\beta}}{2\mu\lambda_{\min}^p} \right\}$$

*. Then the iterates of Fed-Bernoulli-LoRA-EF21 (Algorithm 9) satisfy*

$$\mathbb{E}\left[f(W^T) - f^*\right] \leq (1 - \gamma\mu\lambda_{\min}^p)^T \Phi_0, \tag{141}$$

*where $\lambda_{\min}^p := p\lambda_{\min}^{H_B} + (1-p)\lambda_{\min}^{H_A}$, $\lambda_{\max}^p := p\lambda_{\max}^{H_B} + (1-p)\lambda_{\max}^{H_A}$, and $\Phi_0 = f(W^0) - f^* + \frac{\gamma\lambda_{\max}^p}{1-\sqrt{1-\beta}} \frac{1}{M} \sum_{l=1}^M \left\|G_l^0 - \nabla f_l(W^0)\right\|_F^2$.*

*Proof.* Denote Lyapunov function $\Phi_t$ as follows

$$\Phi_t = f(W^t) - f^* + \frac{\gamma\lambda_{\max}^p}{1-\sqrt{1-\beta}} \cdot \frac{1}{M} \sum_{l=1}^M \left\|G_l^t - \nabla f_l(W^t)\right\|_F^2. \tag{142}$$

By Lemma 12 and Lemma 17, we have

$$\mathbb{E}[\Phi_{t+1}] \leq \mathbb{E}\left[f(W^t)\right] - f^* - \frac{\gamma\lambda_{\min}^p}{2}\mathbb{E}\left[\left\|\nabla f(W^t)\right\|_F^2\right] - \left(\frac{1}{2\gamma} - \frac{L}{2}\right)\mathbb{E}\left[\left\|W^{t+1} - W^t\right\|_F^2\right]$$

$$+ \frac{\gamma\lambda_{\max}^p}{2} \cdot \mathbb{E}\left[\left\|G^t - \nabla f(W^t)\right\|_F^2\right] + \frac{\gamma\lambda_{\max}^p\sqrt{1-\beta}}{1-\sqrt{1-\beta}} \cdot \frac{1}{M} \sum_{l=1}^M \mathbb{E}\left[\left\|G_l^t - \nabla f_l(W^t)\right\|_F^2\right]$$

$$+ \frac{\gamma\lambda_{\max}^p(1-\beta)L^2}{(1-\sqrt{1-\beta})^2}\mathbb{E}\left[\left\|W^{t+1} - W^t\right\|_F^2\right]$$

$$\leq (1 - \gamma\mu\lambda_{\min}^p)\mathbb{E}\left[f(W^t) - f^*\right] + \frac{\gamma\lambda_{\max}^p\left(1 + \sqrt{1-\beta}\right)}{2(1-\sqrt{1-\beta})} \cdot \frac{1}{M} \sum_{l=1}^M \mathbb{E}\left[\left\|G_l^t - \nabla f_l(W^t)\right\|_F^2\right]$$

$$- \left(\frac{1}{2\gamma} - \frac{L}{2} - \frac{\gamma\lambda_{\max}^p(1-\beta)L^2}{(1-\sqrt{1-\beta})^2}\right)\mathbb{E}\left[\left\|W^{t+1} - W^t\right\|_F^2\right],$$

where in the last inequality we used Assumption 6. Selecting $0 < \gamma \leq$ $\min\left\{\frac{1}{L\left(1 + \frac{\sqrt{2\lambda_{\max}^p(1-\beta)}}{1-\sqrt{1-\beta}}\right)}, \frac{1+\sqrt{1-\beta}}{2\mu\lambda_{\min}^p}\right\}$, we obtain

$$\mathbb{E}[\Phi_{t+1}] \leq (1 - \gamma\mu\lambda_{\min}^p)\mathbb{E}[\Phi_t].$$

Taking the recursion, we have

$$\mathbb{E}[\Phi_T] \leq (1 - \gamma\mu\lambda_{\min}^p)^T \Phi_0.$$

$\qquad\square$

complete it was that from new reps

## J    EXPERIMENTS: MISSING DETAILS

In this section, we provide additional details regarding the experimental setting from Section 7.

### J.1    LINEAR REGRESSION WITH NON-CONVEX REGULARIZATION

**Full gradient setting.** We begin by evaluating these methods in a standard optimization setting where full gradients are computed at each iteration. In this regime, we compare Bernoulli-LoRA-GD and RAC-LoRA-GD.

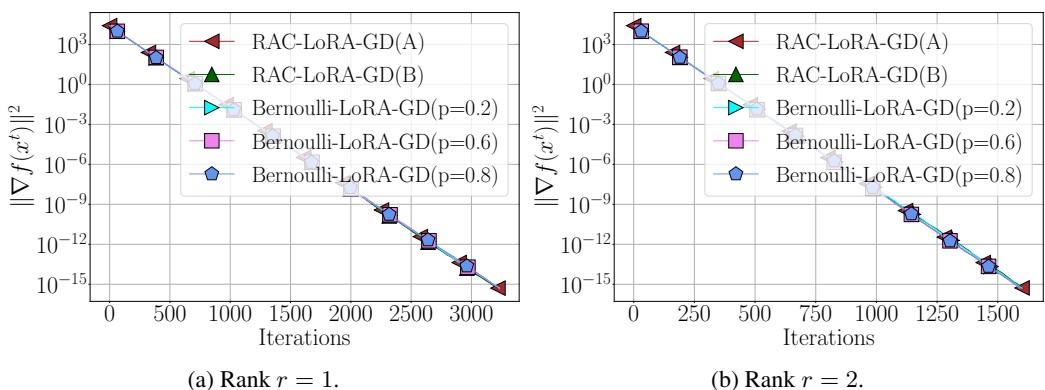

(a) Rank $r = 1$.                          (b) Rank $r = 2$.

Figure 2: Comparison of RAC-LoRA-GD and Bernoulli-LoRA-GD on linear regression fine-tuning. Curves with $p = 0.01, 0.2, \ldots$ indicate Bernoulli-LoRA-GD sampling parameters. RAC-LoRA-GD(A) trains $B$ after resampling $A$, while RAC-LoRA-GD(B) does the reverse. All methods use $\gamma = c/\hat{L}$ with $c \in \{1, 2\}$ tuned individually.

Figure 2 shows that, across all tested probabilities, Bernoulli-LoRA-GD and both variants of RAC-LoRA-GD exhibit similar convergence on the linear regression task. This numerical stability suggests that the ratio of updates between $A$ and $B$ has little effect on the performance for this problem. We also observe that higher ranks $r$ produce faster convergence, which aligns with the theoretical $r/n$ factor in our analysis.

**Hardware and Software.**    All algorithms were implemented in Python 3.10 and executed on three different CPU cluster node types:

1. AMD EPYC 7702 64-Core,
2. Intel(R) Xeon(R) Gold 6148 CPU @ 2.40GHz,
3. Intel(R) Xeon(R) Gold 6248 CPU @ 2.50GHz.

**Implementation Details.**    For each method, we set the stepsize to $\gamma = c/\hat{L}$, where $c$ is a constant multiplier tuned individually for every algorithm. Convergence was monitored by computing the squared norm of the full gradient at each iteration. The algorithms terminated when either a maximum iteration limit was reached or the criterion $\|\nabla f(x^t)\|_2^2 \leq 5 \times 10^{-16}$ was satisfied. To ensure reliability, each method was run 20 times using different random seeds, and all figures show the median performance over these trials.

**Datasets.**    The synthetic pre-training dataset $(\widetilde{D}, \widetilde{b})$ was generated using

$$\texttt{sklearn.datasets.make\_regression}$$

with moderate noise and a controlled rank structure:

```
1  wt_D, wt_b = make_regression(n_samples=90000, n_features=4096,
2                               n_informative=4096, noise=20.0,
3                               bias=0.0, tail_strength=0.8,
4                               effective_rank=64, random_state=42)
```

followed by standard scaling. The fine-tuning dataset $(\hat{D}, \hat{b})$ was produced similarly:

```
1  h_D, h_b = make_regression(n_samples=10000, n_features=4096,
2                             n_informative=4096//2, noise=50.0,
3                             bias=10.0, tail_strength=0.9,
4                             effective_rank=32, random_state=84)
```

and subsequently adjusted with a biased scaling (mean 1, standard deviation 2).

## LLM USE ACKNOWLEDGMENT

In this paper, we used large language models (LLMs) to assist with grammar and wording during the preparation of the manuscript. We did not use LLMs to derive convergence theorems, generate empirical plots, or search for citations. This usage is in accordance with two primary LLM-related policies.

