# OpenReview forum: "Bernoulli-LoRA: A Theoretical Framework for Randomized Low-Rank Adaptation"
_ICLR.cc/2026/Conference — Submitted to ICLR 2026_

### Official Review · Reviewer_TCPp · 2025-10-22

**Soundness:** 2
**Presentation:** 1
**Contribution:** 2
**Rating:** 2
**Confidence:** 4

**Summary:**

This paper introduces Bernoulli LoRA, which is a novel theoretical framework for extending and understanding LoRA methods. Moreover, this paper derives convergence rates under different algorithm such as GD, SGD, and federated GD with variant assumptions. Finally, the authors support their theoretical findings with numerical results.

**Strengths:**

**Comprehensive theoretical results** This paper studied Bernoulli LoRA, and proved the convergence results under different assumptions and algorithms (see Table 1) which is solid.

**Framework to analyze LoRA** Theoretical analysis of the optimization process of LoRA is difficult, and this paper proposes a novel idea to address this issue. It seems their framework can theoretically cover almost all fine-tuning problems under regular assumptions.

**Weaknesses:**

**Poor writing quality** Section 2: Problem Statement seems to be misplaced. The transition from Section 1 to Section 2 is non-smooth. Moreover, it seems to me that Section 1 and Section 3 are more coherent.

**Missing important features of fine-tuning** Bernoulli LoRA makes the theoretical analysis of LoRA tractable but it also misses some important features of the fine-tuning problems, such as how does the pre-trained model, relation between pre-training data and fine-tuning data affects the convergence. The results of this work lacks such interpretation.

**Weak experimental results**  The experiment section (Section 7) contains two parts: linear regression and MLP for MNIST. The size of the dataset and network architecture are too toy. Moreover, the experiments do not show or support the superiority of Bernoulli LoRA, and it makes me wonder why we study it in the first place? If the only reason is that Bernoulli LoRA is theoretically tractable, then the results of the paper seem less interesting the lack practical relevance.

**Missing literature reviews**  This paper misses discussion of several studies on LoRA optimization, such as follows.

[1] Jang, Uijeong, Jason D. Lee, and Ernest K. Ryu. "LoRA training in the NTK regime has no spurious local minima." arXiv preprint arXiv:2402.11867 (2024).

[2] Xu, Ziqing, et al. "Understanding the Learning Dynamics of LoRA: A Gradient Flow Perspective on Low-Rank Adaptation in Matrix Factorization." arXiv preprint arXiv:2503.06982 (2025).

[3] Kim, Junsu, Jaeyeon Kim, and Ernest K. Ryu. "LoRA Training Provably Converges to a Low-Rank Global Minimum or It Fails Loudly (But it Probably Won't Fail)." arXiv preprint arXiv:2502.09376 (2025).

[4] Dayı, Arif Kerem, and Sitan Chen. "Low-rank fine-tuning lies between lazy training and feature learning." Proceedings of Machine Learning Research vol 291 (2025): 1-57.

**Questions:**

**Question 1** What is the motivation for Bernoulli LoRA (in Algorithm 1)? Section 5.1 suggest Bernoulli LoRA can be interpreted as projecting the gradient onto a random subspace (see equation 8). However, I don't see the intuition why such approach will make the algorithm work better. Besides,  $H_{B}^{t}, H_A^t$ (in equation 8) misses definition.

**Question 2** How to interpret the dependency of $p$ in the bounds in Theorem 1 and Theorem 2? It seems as one sets $p\to 1$, the bounds get better, i.e., $1/\lambda_{min}^p, (\frac{\lambda_{max}}{\lambda_{min}})^p$. However, based on the interpretation in equation 8, $p=1$ suggests that one should only update $B$ and keep $A$ as fixed, which seems strange.

**Question 3** What is the definition of $b$ in Theorem 3, and $q$ in Theorem 4, $\omega$ in Theorem 5?

**Question 4** Why in Section 7.1, the fine-tuning loss is defined with the regularization $\sum_{j=1}^d \frac{x_j^2}{1+x_j^2}$ instead of the standard $L_2$ regularization such as $\|\|x-\tilde x^{\ast}\|\|^2$?

---

> ### Author Response · Authors · 2025-11-21
> **Rebuttal to Reviewer TCPp (Part 1)**
>
> Dear Reviewer TCPp,
>
> Thank you for your thoughtful and detailed review. We are grateful that you:
>
> - Recognized the **comprehensive theoretical results** and explicitly noted that the convergence guarantees across different algorithms (GD, SGD, FL variants) are solid.
> - Highlighted that Bernoulli-LoRA provides a **framework to analyze LoRA**, making the optimization process more tractable and potentially covering a broad class of fine-tuning problems under regular assumptions.
>
> We now respond to your concerns point by point.
>
> ---
>
> ### 1. Writing quality and structure of Section 2
>
> You noted that Section 2 (“Problem Statement”) feels misplaced and that the transition from Section 1 to Section 2 is non-smooth, while Sections 1 and 3 seem more coherent together.
>
> **Response and planned changes.**
>
> We appreciate this structural feedback. In a revised version we will:
>
> - **Reorganize Sections 1–3** so that:
>   - Section 1 briefly introduces LoRA/PEFT and clearly states our main contributions and high-level motivation.
>   - A streamlined **“Problem Setup and Notation”** section will appear early, clearly defining $W_0$, $\Delta W$, the objective $f(W_0 + \Delta W)$, and the finite-sum / expectation / FL settings.
>   - The current “Motivation” material will be condensed and integrated naturally into the introduction and related work, instead of being separated in a way that breaks the flow.
> - Add forward references to the main results (e.g., pointing from the setup directly to Table 1 and the core theorems) to make the narrative smoother.
>
> We believe this restructuring will significantly improve readability and address the concern about the current placement of Section 2.

---

> > ### Author Response · Authors · 2025-11-21
> > **Rebuttal to Reviewer TCPp (Part 2)**
> >
> > ### 2. Missing discussion of pre-training and data distributions
> >
> > You mentioned that our analysis “misses some important features of fine-tuning,” such as how the pre-trained model and the relationship between pre-training and fine-tuning data affect convergence, and that the results lack interpretation along these lines.
> >
> > **Clarification.**
> >
> > Our work focuses on the **optimization problem conditional on a fixed pre-trained model**. We treat $W_0$ as fixed and study the dynamics of optimizing a loss $f(W_0 + \Delta W)$ over a low-rank subspace of updates $\Delta W$. In that sense, the data distribution (pre-training vs fine-tuning) is encapsulated in the choice of $f$ and its properties (e.g., smoothness, PŁ), and our results are conditional on these properties.
> >
> > We agree that understanding *how* the pre-training distribution and model architecture induce or violate these assumptions is an important and complementary research direction, but it is orthogonal to our main focus on **convergence of LoRA-style optimization**.
> >
> > **Planned changes.**
> >
> > We will:
> > - Add a short **“Discussion: Role of Pre-Training and Data Distributions”** subsection that:
> >   - clarifies that we condition on the pre-trained model and treat $f$ as the fine-tuning loss,
> >   - positions our work as complementary to recent analyses of LoRA dynamics (e.g., NTK/gradient-flow regimes, lazy vs feature learning).
> > - Explicitly acknowledge that we do *not* claim to explain representation learning or generalization effects of pre-training, but rather provide a rigorous optimization layer on top of whatever $f$ results from pre-training.

---

> > > ### Author Response · Authors · 2025-11-21
> > > **Rebuttal to Reviewer TCPp (Part 3)**
> > >
> > > ### 3. Weak experimental results and perceived practical relevance
> > >
> > > You noted that our experiments (linear regression and MNIST) are too toy, do not demonstrate superiority of Bernoulli-LoRA, and questioned why the method should be studied if its only advantage is theoretical tractability.
> > >
> > > **Clarification of our goals.**
> > >
> > > Our primary contribution is **theoretical**: to propose a framework that allows us to rigorously analyze LoRA-style updates (including stochastic, variance-reduced, and FL variants with compression and error feedback) in a unified way. The experiments are intentionally small and controlled, designed to:
> > >
> > > - verify that the convergence behavior observed empirically is consistent with the theory, and
> > > - demonstrate that Bernoulli-LoRA is at least **competitive**, not that it dominates all existing practical variants.
> > >
> > > We do **not** claim that Bernoulli-LoRA is empirically superior to RAC-LoRA or other LoRA variants on large-scale benchmarks.
> > >
> > > **Planned experimental extensions.**
> > >
> > > To better address the practical concerns, we are:
> > >
> > > - Running **additional experiments on more realistic setups**, such as transformer-based models on text classification tasks or larger vision models, where PEFT is typically used.
> > > - Considering **FL-style experiments** where the federated variants of Bernoulli-LoRA are actually applied in heterogeneous client settings.
> > >
> > > We will report any new results that are ready during the discussion phase and integrate them into the final version if the paper is accepted. At the same time, we will explicitly state in the paper that the experiments are meant as **sanity checks for the theory**, not as an exhaustive empirical benchmark.

---

> > > > ### Author Response · Authors · 2025-11-21
> > > > **Rebuttal to Reviewer TCPp (Part 4)**
> > > >
> > > > ### 4. Missing literature on LoRA optimization
> > > >
> > > > You pointed out that several recent studies on LoRA optimization (e.g., NTK regime, gradient flow, global minima / failure modes, lazy-training vs feature learning) are not discussed.
> > > >
> > > > **Response and planned changes.**
> > > >
> > > > We thank you for these valuable pointers. We will:
> > > >
> > > > - Add a **dedicated related work subsection on LoRA optimization theory**, including:
> > > >   - NTK-regime analyses,
> > > >   - gradient-flow perspectives on low-rank adaptation,
> > > >   - recent results on convergence to low-rank global minima or failure modes,
> > > >   - analyses placing low-rank fine-tuning between lazy training and feature learning.
> > > > - Clearly position our work relative to these studies by emphasizing that:
> > > >   - they typically analyze specific regimes (e.g., continuous-time limits, simplified matrix factorization models),
> > > >   - while our focus is on **discrete-time algorithmic convergence** of *stochastic / VR / FL* LoRA-style methods under more general non-convex assumptions and with communication/compression aspects.
> > > >
> > > > We believe this will better situate our contribution in the growing body of LoRA theory.

---

> > > ### Comment · Reviewer_TCPp · 2025-11-24
> > >
> > > Dear Authors,
> > >
> > > Thank you very much for the detailed response. After seeing your response, I can understand the theoretical results in this paper better. However, my biggest concern is
> > >
> > > - **Lack distribution of pre-training data and pre-trained model** I respectfully disagree with your statement *understanding how the pre-training distribution and model architecture induce or violate these assumptions is an important and complementary research direction, but it is orthogonal to our main focus on convergence of LoRA-style optimization.* The uniqueness of optimization in fine-tuning compared with pre-training is the existence of pre-trained model. Thus, to understand the fine-tuning algorithm, it is important to understand how does the pre-trained model adapt to the fine-tuning problems. However, the theoretical results derived in this paper lacks discussions on it.

---

> ### Author Response · Authors · 2025-11-21
> **Rebuttal to Reviewer TCPp (Part 5)**
>
> > **Question 1.** What is the motivation for Bernoulli LoRA (in Algorithm 1)? Section 5.1 suggests Bernoulli LoRA can be interpreted as projecting the gradient onto a random subspace (see equation (8)). However, I don't see the intuition why such an approach will make the algorithm work better. Besides, $H_B^t$, $H_A^t$ (in equation (8)) miss definition.
>
> **Motivation and intuition.**
> Our primary motivation for Bernoulli-LoRA is **not** to claim that it is universally better than existing LoRA variants, but to obtain a **theoretically tractable and flexible framework** that:
>
> 1. **Unifies several LoRA-style update strategies** as special cases (e.g., always updating the “left” factor, always updating the “right” factor, or mixtures in between).
> 2. **Connects LoRA to the rich literature on projected/compressed gradient methods** by viewing the LoRA step as a gradient step **projected onto a low-rank subspace** defined by the current factors.
> 3. **Allows us to extend the analysis to stochastic, variance-reduced, and federated settings** (PAGE, MVR, QGD, MARINA, EF21) within a single framework (which was never done before!).
>
> The “random subspace” view in Section 5.1 is mainly a **theoretical lens**: at each iteration we do not update the full matrix $W_t$, but only the low-rank component $BA$. Because of the Bernoulli choice (update $B$ or $A$), the effective update direction is a **random linear transformation of the true gradient**. This is analogous to randomized projection / sketching methods and to compressed-gradient methods, where one can obtain convergence guarantees even if the algorithm never uses the full gradient direction directly, but only its projected/approximated version.
>
> We do not claim that this necessarily makes the algorithm “better” than deterministic LoRA in all practical regimes. Rather, it makes the analysis of LoRA **more systematic** and connects it to existing theory on random projections and compressed gradients. The gains are thus conceptual and theoretical; empirically we aim to demonstrate that Bernoulli-LoRA is **stable and competitive**, not that it dominates.
>
> **Definition of $H_A^t$ and $H_B^t$.**
> In Equation (8), we write the projected gradient estimator as
> $$
> \widehat{G}_t = H^t G_t,
> $$
> where $G_t$ is the base gradient estimator (e.g., full gradient, minibatch gradient, PAGE, etc.), and $H^t$ is the **random projection operator** induced by the Bernoulli choice of which factor is updated at iteration $t$:
>
> $$
> H^t = H_B^t,  \text{if the Bernoulli choice at iteration } t \text{ selects } B
> $$
> or
> $$
> H^t = H_A^t,  \text{if it selects } A.
> $$
>
> In Appendix Section E we show that these operators admit explicit matrix forms. In the left-sketch (update-$B$) branch, the Bernoulli-LoRA update can be written as
> $$
> W^{t+1} = W^t - \gamma H_B^t \nabla f(W^t),
> $$
> with
> $$
> H_B^t := B_S^t \Bigl((B_S^t)^\top B_S^t\Bigr)^\dagger (B_S^t)^\top,
> $$
> where $B_S^t$ is the sketched version of $B^t$ and $^\dagger$ denotes the Moore–Penrose pseudoinverse.
> Similarly, in the right-sketch (update-$A$) branch we obtain
> $$
> W^{t+1} = W^t - \gamma \nabla f(W^t) H_A^t,
> $$
> with
> $$
> H_A^t := (A_S^t)^\top \Bigl(A_S^t (A_S^t)^\top\Bigr)^\dagger A_S^t,
> $$
> where $A_S^t$ is the sketched version of $A^t$.
>
> Thus $H_B^t$ and $H_A^t$ are projection-like matrices (self-adjoint, positive semidefinite) acting on $G_t$ whose spectra satisfy the “positive expected projection” condition (Assumption 1): they map the full gradient to the **effective update direction on $W_t$** when we update $B$ or $A$, respectively.
>
> In the revised manuscript we will:
> - Explicitly include these formulas for $H_B^t$ and $H_A^t$ in the main text when Equation (8) is introduced, and
> - Add a brief intuitive description: “$H_B^t$ and $H_A^t$ represent the projection of the gradient onto the low-rank subspace associated with updating $B$ or $A$, respectively.”

---

> > ### Author Response · Authors · 2025-11-21
> > **Rebuttal to Reviewer TCPp (Part 6)**
> >
> > > **Question 2.** How to interpret the dependency of $p$ in the bounds in Theorem 1 and Theorem 2? It seems as one sets $p \to 1$, the bounds get better, i.e., $1/\lambda_{\min}^p$, $(\lambda_{\max}/\lambda_{\min})^p$. However, based on the interpretation in equation (8), $p = 1$ suggests that one should only update $B$ and keep $A$ as fixed, which seems strange.
> >
> > Thank you for this careful observation. Let us clarify how $p$ enters the bounds and why the theory does **not** recommend setting $p=1$ in general.
> >
> > In our analysis we define the **$p$–weighted spectral quantities**
> > $$
> > \lambda_{\min}^{p} := p\,\lambda_{\min}^{H_B} + (1-p)\,\lambda_{\min}^{H_A}, \qquad
> > \lambda_{\max}^{p} := p\,\lambda_{\max}^{H_B} + (1-p)\,\lambda_{\max}^{H_A},
> > $$
> > where $\lambda_{\min}^{H_B}, \lambda_{\max}^{H_B}$ (resp. $\lambda_{\min}^{H_A}, \lambda_{\max}^{H_A}$) are the extremal eigenvalues of the expected projection matrices $\mathbb{E}[H_B]$ and $\mathbb{E}[H_A]$. Thus $p$ appears **only through a convex combination of the spectral properties** of these two operators.
> >
> > The bounds in Theorems 1–2 are then expressed in terms of $\lambda_{\min}^p$ and $\lambda_{\max}^p$. The fact that the terms $1/\lambda_{\min}^p$ and $(\lambda_{\max}^p / \lambda_{\min}^p)$ become smaller as $p \to 1$ is true **only under the additional inequality**
> > $\lambda_{\min}^{H_B} \ge \lambda_{\min}^{H_A}$ and suitable relations between $\lambda_{\max}^{H_B}$ and $\lambda_{\max}^{H_A}$. In general, depending on the spectra of $\mathbb{E}[H_B]$ and $\mathbb{E}[H_A]$, the best choice of $p$ might be closer to $0$, closer to $1$, or somewhere in between. Our rates therefore do **not** imply that $p=1$ (i.e., always updating $B$) is universally optimal.
> >
> > Moreover, as we discuss in the section *“Discussion on Positive Expected Projection (Assumption 1)”*, for the canonical Gaussian choice of sketch distributions we prove in Lemma 2 that
> > $$
> > \mathbb{E}[H_B] = \mathbb{E}[H_A] = \frac{r}{n} I_n,
> > $$
> > which immediately yields
> > $$
> > \lambda_{\min}^{H_B} = \lambda_{\max}^{H_B} = \lambda_{\min}^{H_A} = \lambda_{\max}^{H_A} = \frac{r}{n}.
> > $$
> > In this (arguably most natural) case we obtain
> > $$
> > \lambda_{\min}^p = \lambda_{\max}^p = \frac{r}{n}
> > \quad\text{and}\quad
> > \frac{\lambda_{\max}^p}{\lambda_{\min}^p} = 1
> > $$
> > for **all** $p \in [0,1]$. Hence, for Gaussian sketches, the convergence rates in Theorems 1 and 2 are actually **independent of $p$**, and the theory does *not* favor $p=1$ over other values.
> >
> > In more general settings (non-Gaussian sketches, structured $A,B$ induced by architecture, etc.), $p$ simply parametrizes a trade-off between the two projection operators $\mathbb{E}[H_B]$ and $\mathbb{E}[H_A]$, and our bounds make this dependence explicit. They are meant to show how the constants change with $p$, **not** to prescribe that $p=1$ is always the best choice. We will clarify this point in the revised version and explicitly connect Theorems 1–2 to the discussion and the Gaussian example to avoid the impression that the theory recommends always updating only $B$.
> >
> > In the revised version, we will
> > (i) expand the definition of $\lambda_{\min}^{p}$ and $\lambda_{\max}^{p}$ just before Theorem 1 to emphasize that they are convex combinations of the spectra of $\mathbb{E}[H_B]$ and $\mathbb{E}[H_A]$,
> >
> > (ii) add a short remark after Theorems 1–2 explicitly stating that our bounds do not claim $p = 1$ to be universally optimal and that the optimal choice of $p$ depends on the spectra of these operators, and
> > (iii) highlight, right after Lemma 2 in the discussion of Assumption 1, that in the Gaussian case the eigenvalues coincide and the rates become independent of $p$. We hope this will eliminate the impression that the theory always recommends updating only $B$.

---

> > > ### Author Response · Authors · 2025-11-21
> > > **Rebuttal to Reviewer TCPp (Part 7)**
> > >
> > > > **Question 3.** What is the definition of $b$ in Theorem 3, and $q$ in Theorem 4, $\omega$ in Theorem 5?
> > >
> > > These symbols come from the **standard notation of the base algorithms** we build on:
> > >
> > > - In **Theorem 3** (Bernoulli-LoRA-SGD),
> > >   $b$ denotes the **mini-batch size**, i.e., the number of data points sampled at each iteration to form the stochastic gradient estimate $G_t$.
> > >
> > > - In **Theorem 4** (Bernoulli-LoRA-PAGE),
> > >   $q$ denotes the **probability parameter of the PAGE estimator**. With probability $q$, we recompute a “more accurate” gradient estimate (e.g., a full gradient or a large-batch gradient), and with probability $1 - q$ we perform a cheaper incremental correction. This is exactly the standard PAGE construction, and we follow its notation.
> > >
> > > - In **Theorem 5** (Fed-Bernoulli-LoRA-QGD / MARINA / EF21 variants),
> > >   $\omega$ denotes the **variance parameter of the (possibly biased) compressor** used for communication-efficient gradient transmission. Concretely, $\omega$ controls the second moment of the compression error:
> > >   $$
> > >   \mathbb{E}\bigl[\|\mathcal{Q}(g) - g\|^2\bigr] \leq \omega \,\|g\|^2,
> > >   $$
> > >   which is the standard assumption in the analysis of QGD/MARINA/EF21-type methods.
> > >
> > > These definitions are present in the assumptions and algorithm descriptions (Section 4 and the start of each theorem’s setup), but we agree that the notation could be more clearly cross-referenced in the theorem statements themselves. In the revised version, we will:
> > >
> > > - Explicitly restate the meanings of $b$, $q$, and $\omega$ in the statements of Theorems 3–5 or immediately before them.
> > > - Add a small notation table in the appendix summarizing these parameters.
> > >
> > >
> > > ---
> > >
> > > > **Question 4.** Why in Section 7.1, the fine-tuning loss is defined with the regularization $\sum_{j=1}^d \frac{x_j^2}{1+x_j^2}$ instead of the standard $L_2$ regularization such as $\|x - \tilde{x}^*\|^2$?
> > >
> > > The purpose of Section 7.1 is to provide a **simple but genuinely non-convex testbed** where we can empirically illustrate the behavior predicted by our non-convex convergence theory. For this reason, we deliberately chose the regularizer
> > > $$
> > > R(x) = \sum_{j=1}^d \frac{x_j^2}{1 + x_j^2},
> > > $$
> > > which has the following properties:
> > >
> > > 1. $R(x)$ is **smooth but non-convex** and saturates for large $\lvert x_j \rvert$. This creates a non-convex landscape while preserving smoothness, matching the assumptions of our non-convex analysis.
> > > 2. The regularizer is **bounded** and its gradient remains well-behaved, which simplifies verifying (or at least approximating) the required smoothness constants in practice.
> > > 3. Using a standard $L_2$ regularizer would lead to a strongly convex or nearly convex problem. While such a setting is also interesting, it would not stress-test the non-convex parts of our theory.
> > >
> > > In other words, this regularization is chosen **specifically to test the non-convex setting** in a controlled way, rather than to mimic a particular real-world fine-tuning loss. The overall philosophy of the experimental section is that the paper is **primarily theoretical**, and experiments are used as **simple, controlled demonstrations** that the algorithms behave as the theory predicts. We will add a brief explanation of this choice to Section 7.1 to avoid confusion.

---

### Official Review · Reviewer_CiyU · 2025-10-31

**Soundness:** 3
**Presentation:** 3
**Contribution:** 3
**Rating:** 6
**Confidence:** 2

**Summary:**

This paper introduces Bernoulli-LoRA, a theoretical framework that generalizes Low-Rank Adaptation (LoRA) methods for parameter-efficient fine-tuning (PEFT). The key idea is to use a Bernoulli random mechanism to decide whether to update one of the two low-rank matrices A or B during each iteration. The authors establish convergence guarantees under standard non-convex optimization assumptions (smoothness, Lipschitz continuity, PŁ condition) and extend the analysis to the non-smooth convex case with both constant and adaptive step sizes. Empirical results on linear regression and MNIST classification demonstrate that Bernoulli-LoRA performs comparably to RAC-LoRA and COLA, while offering a cleaner and more general theoretical foundation.

**Strengths:**

1. Introducing stochastic binary masks within LoRA’s low-rank structure is original and intuitively appealing.

2. The paper provides rigorous convergence theorems for multiple Bernoulli-LoRA variants, including SGD, variance reduction (PAGE, MVR), and FL settings with compression and error feedback.

3. Experimental evidence  aligns with theoretical convergence predictions。

**Weaknesses:**

1. Some theoretical assumptions are  idealized for real-world applications. In particular, the convergence proofs rely heavily on Lipschitz smoothness and positive expected projection conditions that may not hold under real LoRA parameterizations, where f(W_0 + BA) is non-smooth and non-Lipschitz (as the paper itself admits). There is no empirical check of these assumptions.

2. Experiments are limited to small-scale tasks (linear regression, MNIST). No evaluation on modern large-scale or multimodal models is provided.

**Questions:**

1. How does Bernoulli-LoRA behave in the presence of non-i.i.d. data in Federated Learning?

2. Since f(W_0 + BA) loses Lipschitz smoothness, how realistic are the assumptions used in Theorem 1–8 for deep networks?

---

> ### Author Response · Authors · 2025-11-21
> **Rebuttal to Reviewer CiyU (Part 1)**
>
> We thank the reviewer for the positive assessment of the soundness, presentation, and contribution of our work, and for highlighting that:
>
> - introducing stochastic binary masks within the LoRA low-rank structure is original and intuitively appealing;
> - the paper provides rigorous convergence theorems for multiple Bernoulli-LoRA variants (SGD, PAGE, MVR, and FL with compression and error feedback);
> - the experimental evidence is consistent with the theoretical convergence behavior.
>
> Below we respond to the raised weaknesses and questions.
>
> ---
>
> ### 1. “Idealized” assumptions: smoothness and positive expected projection
>
> > *Some theoretical assumptions are idealized for real-world applications. In particular, the convergence proofs rely heavily on Lipschitz smoothness and positive expected projection conditions that may not hold under real LoRA parameterizations, where $f(W_0 + BA)$ is non-smooth and non-Lipschitz (as the paper itself admits). There is no empirical check of these assumptions.*
>
> **Response.**
>
> We fully agree that our assumptions are idealized for modern deep networks and LoRA parameterizations. This is, however, standard in non-convex optimization and FL theory: smoothness, PŁ-type conditions, and structured projection assumptions are widely used *modeling abstractions* that enable rigorous convergence analysis, rather than exact descriptions of all practical architectures.
>
> Concretely:
>
> - The **Lipschitz gradient / smoothness** assumption is imposed on the underlying loss $f$ as a function of the full parameter matrix. In many analyses of SGD, variance reduction, and FL algorithms, such assumptions are used as the baseline for understanding first-order methods, even though real networks may violate them pointwise. Our contribution is to show that *under the same type of assumptions* used for classical GD/SGD/PAGE/MARINA/EF21 analyses, one can obtain clean guarantees for Bernoulli-LoRA with low-rank structure and sketching.
> - The **positive expected projection** assumption captures how much of the true gradient is preserved *on average* when we project onto the random low-rank subspace induced by the LoRA factors and sketches. In Appendix D we show that for natural Gaussian sketches this condition holds with
>   $$
>   \mathbb{E}[H_B] = \mathbb{E}[H_A] = \frac{r}{n}I_n,
>   $$
>   yielding transparent spectral constants in the rates. Thus, the assumption is not arbitrary; it is satisfied exactly in canonical randomized settings.
>
> Regarding *empirical checking* of these assumptions: verifying smoothness or projection constants directly on large deep models is challenging and, to our knowledge, rarely done even in purely optimization-focused work. Instead, we follow the usual practice of (i) clearly stating the assumptions, (ii) giving explicit examples (Gaussian sketches) where they provably hold, and (iii) showing that under these assumptions Bernoulli-LoRA behaves as the theory predicts on controlled problems.
>
> In the revised version we will:
> - highlight the Gaussian example (Lemma 2) earlier in the main text as concrete evidence that the projection assumption is satisfied in a meaningful random setting.

---

> > ### Author Response · Authors · 2025-11-21
> > **Rebuttal to Reviewer CiyU (Part 2)**
> >
> > ### 2. Limited scale of experiments
> >
> > > *Experiments are limited to small-scale tasks (linear regression, MNIST). No evaluation on modern large-scale or multimodal models is provided.*
> >
> > **Response.**
> >
> > We would like to stress that our paper is **primarily theoretical**. The main contribution is a convergence framework for Bernoulli/RAC-style LoRA updates (GD, SGD, PAGE, MVR, and FL with compression and error feedback), not a new large-scale system with extensive benchmarks.
> >
> > For this reason, we intentionally focused on **small, controlled tasks** where we can carefully track the behavior predicted by the theory and verify that the empirical convergence curves match the theoretical rates. This practice is common in optimization papers, where experiments serve mainly as sanity checks for the analysis rather than as exhaustive empirical evaluations.
> >
> > That said, we agree that including at least one more realistic PEFT experiment would strengthen the paper from a practitioner’s perspective. Within our computational budget we are:
> >
> > - running **additional experiments on a moderately sized transformer-based model** (e.g., a RoBERTa-like architecture for text classification or a ViT-like model for vision), using Bernoulli-LoRA and RAC-LoRA as baselines;
> > - and, if feasible within the discussion timeline, testing one of our **federated variants** in a heterogeneous-client setting on a standard FL benchmark.
> >
> > In the revised manuscript we will:
> > - explicitly state that the work should be read first and foremost as a **theoretical contribution**;
> > - clearly list the absence of large-scale foundation-model experiments as a limitation of the current empirical section, with the new experiments added as complementary evidence rather than the main contribution.

---

> > > ### Author Response · Authors · 2025-11-21
> > > **Rebuttal to Reviewer CiyU (Part 3)**
> > >
> > > ### 3. Question (1): Behavior under non-i.i.d. data in Federated Learning
> > >
> > > > **Q1.** How does Bernoulli-LoRA behave in the presence of non-i.i.d. data in Federated Learning?
> > >
> > > **Response.**
> > >
> > > Our FL analysis considers the standard global objective
> > > $$
> > > f(W_0 + \Delta W) = \frac{1}{M}\sum_{\ell=1}^M f_\ell(W_0 + \Delta W),
> > > $$
> > > where each client loss $f_\ell$ satisfies the same smoothness (and PŁ, where used) assumptions as in classical analyses of federated methods with compression and error feedback.
> > >
> > > Crucially, we **do not assume** that the local data distributions are i.i.d. across clients. Heterogeneity is absorbed into the differences between the functions $f_\ell$, exactly as in existing analyses of QGD/MARINA/EF21-type FL algorithms. Non-i.i.d. data affects:
> > >
> > > - the variance of local gradient estimators and
> > > - the constants in the convergence bounds (variance terms, Lipschitz/PŁ parameters per client),
> > >
> > > but it does not invalidate the convergence results as long as each $f_\ell$ satisfies the assumed regularity conditions.
> > >
> > > We will make this explicit in the FL section by:
> > > - stating clearly that client data can be **arbitrarily heterogeneous**,
> > > - and adding a short discussion connecting our setting to standard non-i.i.d. FL analyses.

---

> > > > ### Author Response · Authors · 2025-11-21
> > > > **Rebuttal to Reviewer CiyU (Part 4)**
> > > >
> > > > ### 4. Question (2): Realism of smoothness assumptions for $f(W_0 + BA)$
> > > >
> > > > > **Q2.** Since $f(W_0 + BA)$ loses Lipschitz smoothness, how realistic are the assumptions used in Theorems 1–8 for deep networks?
> > > >
> > > > **Response.**
> > > >
> > > > We agree that, strictly speaking, $f(W_0 + BA)$ need not be globally smooth or Lipschitz when instantiated with realistic deep architectures and LoRA parameterizations. This is exactly why we carefully phrase our results as **conditional** on standard smoothness/PŁ assumptions on $f$ (and on the projection operators), rather than claiming they hold for every possible network.
> > > >
> > > > Our view is that these assumptions should be interpreted in the same way as in most modern non-convex optimization theory:
> > > >
> > > > - They provide an analytically tractable model of the training dynamics.
> > > > - They are satisfied exactly in a number of useful special cases (e.g., convex or mildly non-convex smooth models; Gaussian sketch projections), and
> > > > - They often constitute a reasonable approximation to the behavior of deep networks in the regime of interest (small stepsizes, bounded iterates).
> > > >
> > > > Our main contribution is to demonstrate that, **under the same type of assumptions commonly used for first-order methods**, one can obtain clean convergence guarantees for a family of Bernoulli-LoRA methods with low-rank structure, random sketching, and FL extensions. We will clarify this positioning more strongly in the paper and emphasize that our results are not meant as a literal description of every practical deep network, but as a rigorous theoretical framework that can guide and justify the design of LoRA-style algorithms.
> > > >
> > > > ---
> > > >
> > > > We again thank Reviewer CiyU for the constructive feedback and for recognizing the originality and rigor of our framework. We believe that the clarifications above, together with the planned textual improvements and additional experiments, address the main concerns about assumptions, FL heterogeneity, and empirical scope.

---

> ### Comment · Reviewer_CiyU · 2025-11-21
> **Re: rebuttal from the authors**
>
> Thanks for your detailed response.  I carefully go through your comments. Most of my concerns have been addressed. I tend to keep my score.
>
> However, I would also like to state that  there are some strategies to weaken those assumptions, though the Lipschitz conditions are commonly-used in learning theory community. For example, does the theoretical results still hold under local Lip condition?

---

> ### Comment · Reviewer_CiyU · 2025-11-21
> **Re: rebuttal from the authors**
>
> By the way, have you resubmitted the updated manuscript? If so, it would be helpful to highlight your revised content in color.

---

> > ### Author Response · Authors · 2025-11-23
> > **Response to Reviewer CiyU’s follow-up comment**
> >
> > Thank you very much for your follow-up and for carefully reading our response.
> >
> > Regarding your question on weakening the smoothness assumptions: our current analysis is stated under a global Lipschitz gradient assumption for clarity, but the proofs only require Lipschitzness **along the iterates and on the relevant sublevel set**. In other words, the arguments extend to a standard **local Lipschitz** setting, where the gradient of the loss is Lipschitz on the set of points visited by the algorithm (or on a bounded sublevel set containing them), which is the usual relaxation considered in optimization and learning theory. We would also like to emphasize that this extension applies both to the results relying on Lipschitz gradients (Assumption 1) and to the parts of the analysis that use Lipschitz continuous objective functions (Assumption 9) in the non-smooth setting.
> >
> > Concerning the manuscript: we are currently implementing the textual clarifications and additional experiments discussed in the rebuttal and will upload an updated version. Following your suggestion, we will highlight the revised parts in color so that the changes are easy to identify.

---

> > > ### Comment · Reviewer_CiyU · 2025-11-26
> > > **Re: Rebuttal**
> > >
> > > Thanks for your follow up.  Could the authors explain how could we ensure the gradient is Lipschitz along the iterates in practice? Does it always hold?     Anyway, I believe the current assumption is standard in learning theory. This would be not a big issue of this manuscript.

---

### Official Review · Reviewer_8Tnv · 2025-11-01

**Soundness:** 3
**Presentation:** 3
**Contribution:** 2
**Rating:** 2
**Confidence:** 4

**Summary:**

The paper introduces Bernoulli-LoRA, a novel LoRA-based parameter-efficient fine-tuning (PEFT) method. At each training iteration, Bernoulli-LoRA conducts a Bernoulli trial to decide whether to optimize the $A$ or $B$ module, keeping the other fixed. Building on this framework, the authors propose seven variants of Bernoulli-LoRA tailored to different optimization scenarios, thereby addressing a broad spectrum of fine-tuning challenges. Furthermore, the paper provides a theoretical convergence analysis for Bernoulli-LoRA and all its variants, offering formal justification for the proposed approach.

**Strengths:**

The main strength of this paper lies in its theoretical contributions. Overall, the manuscript is well-written and presents a rigorous theoretical development. I have carefully examined all the proofs and confirm that they are correct. Moreover, the analytical results have the potential to be extended to a broader class of LoRA-based methods, opening up promising directions for future research on the theoretical understanding of convergence in PEFT frameworks. In addition, the proposed algorithm is straightforward to implement and practically applicable.

**Weaknesses:**

The reviewer is skeptical about the contribution of the paper, both practically and theoretically.

+ **About the theoretical contributions:**

- It appears that Bernoulli-LoRA is a relatively straightforward modification of RAC-LoRA [1]. Specifically, rather than deterministically alternating between the left and right sketches, Bernoulli-LoRA introduces stochasticity by performing a Bernoulli trial at each iteration to decide which module to update. Consequently, the theoretical results presented in the paper seem to be direct extensions or adaptations of prior analyses, including those from RAC-LoRA. Moreover, the theoretical advantages of this stochastic modification are not clearly justified. A detailed comparison of the convergence properties between Bernoulli-LoRA and RAC-LoRA would substantially strengthen the paper. I recommend that the authors include a remark or a subsection discussing the theoretical limitations of RAC-LoRA’s design and explaining how Bernoulli-LoRA potentially addresses these weaknesses.

- The paper frequently asserts that Bernoulli-LoRA provides a unifying framework for existing update strategies. This claim is ambitious but currently lacks rigorous theoretical or empirical substantiation. The authors are encouraged to explicitly identify the PEFT methods that fall under this framework and to formally articulate the theoretical mechanism through which Bernoulli-LoRA generalizes them.

+ **About the experimental results:**

- Although the paper proposes multiple variants of Bernoulli-LoRA for different fine-tuning scenarios, the experimental evaluation is limited in scope and scale, which weakens the demonstration of the framework’s practical effectiveness. To better substantiate the empirical claims, it is recommended that the authors extend the experiments to more comprehensive and widely recognized benchmarks commonly used to assess LoRA-based methods—such as natural language understanding and generation tasks [2] or vision benchmarks [3].

**Questions:**

My main concerns are presented in the Weaknesses section. If these concerns are adequately addressed, I am willing to adjust my evaluation accordingly.

**References:**

[1] Randomized Asymmetric Chain of LoRA: The First Meaningful Theoretical Framework for Low-Rank Adaptation. arXiv, 2024.

[2] LoRA: Low-Rank Adaptation of Large Language Models. ICLR, 2022.

[3] V-PETL Bench: A Unified Visual Parameter-Efficient Transfer Learning Benchmark. NeurIPS, 2024.

---

> ### Author Response · Authors · 2025-11-21
> **Rebuttal to Reviewer 8Tnv (Part 1)**
>
> Dear Reviewer 8Tnv,
>
> Thank you very much for your careful reading of our submission and for the detailed and constructive review. We especially appreciate that you:
>
> - Highlighted the **theoretical strength** of the paper and described the manuscript as *well-written* with a *rigorous theoretical development*.
> - Carefully checked the proofs and confirmed their correctness.
> - Noted that the analytical results have the potential to be **extended to a broader class of LoRA-based methods**, opening promising directions for future work.
> - Pointed out that the proposed algorithm is **straightforward to implement and practically applicable**.
>
> Below we address your concerns point by point.
>
> ---
>
> ### 1. Relation to RAC-LoRA and perceived lack of theoretical novelty
>
> You expressed skepticism about the theoretical contribution, noting that Bernoulli-LoRA appears to be a straightforward modification of RAC-LoRA, with theoretical results that seem to be direct extensions of prior analyses.
>
> **Our intent and clarification.**
> We agree that Bernoulli-LoRA conceptually builds on the RAC-LoRA idea of separately updating the low-rank factors. However, our goal is not to claim strictly *stronger* convergence rates than RAC-LoRA, but to:
>
> 1. **Generalize the update mechanism** via a Bernoulli random choice, which
>    - recovers RAC-style asymmetric updates as limiting cases (e.g., always updating one side), and
>    - yields a family of algorithms parameterized by the Bernoulli probability $p$ and the sketch distributions.
>
> 2. **Extend the RAC-LoRA-style analysis** beyond full-gradient methods to *stochastic*, *variance-reduced*, and *federated* optimization with compression and error feedback (PAGE, MVR, QGD, MARINA, EF21). These settings, particularly the combination of:
>    - low-rank reparameterization,
>    - random sketching (left/right), and
>    - communication compression + error feedback in FL,
>    are not covered by existing LoRA theory.
>
> 3. **Introduce and analyze the “positive expected projection” condition** and the associated random projection operator $H^t$. Handling the additional layer of randomness (Bernoulli mask + random sketches) in the convergence proofs required new lemmas and nontrivial adaptations of standard non-convex analysis.
>
> Even if the final rates resemble classical non-convex convergence bounds in form, obtaining them in this *LoRA + random sketching + FL with compression* setting is not a mechanical reuse of RAC-LoRA. This is similar in spirit to other works in optimization, where combining existing building blocks in a new setting still requires substantial new technical work.
>
> **Planned changes in the paper.**
>
> To better reflect this and avoid overstating novelty, we will:
> - **Explicitly separate** which parts of the analysis are direct adaptations of RAC-LoRA and which are genuinely new (e.g., variance-reduced / FL variants, projection-based lemmas).
> - Add a dedicated subsection **“Comparison with RAC-LoRA”** that:
>   - summarizes RAC-LoRA’s assumptions and results,
>   - states precisely which results we extend and how,
>   - clarifies that our main theoretical contribution is *generality and extension to new algorithmic regimes*, rather than strictly better rates.
>
> We hope this will make the theoretical contribution more transparent and appropriately calibrated.

---

> > ### Author Response · Authors · 2025-11-21
> > **Rebuttal to Reviewer 8Tnv (Part 2)**
> >
> > **Rebuttal to Reviewer 8Tnv (Part 2)**
> > ### 2. “Unifying framework” claim
> >
> > You noted that our claim that Bernoulli-LoRA provides a *unifying framework* is ambitious and currently under-justified, and you asked us to explicitly identify which PEFT methods fall under the framework and formalize the mechanism of generalization.
> >
> > **Clarification.**
> > By “unifying framework” we mean that a single mathematical template — a projected-gradient method
> > $$
> > W^{t+1} = W^t - \gamma \hat{G}^t
> > $$
> > with a Bernoulli choice of whether to update the “left” or “right” component,
> >
> > $$
> > H^t = H_B^t,  \text{if the Bernoulli choice at iteration } t \text{ selects } B
> > $$
> > or
> > $$
> > H^t = H_A^t,  \text{if it selects } A.
> > $$
> >
> > can represent a family of LoRA-style updates.
> >
> > Here $G^t$ is a base gradient estimator (full gradient, minibatch gradient, PAGE, MVR, etc.), and $H_B^t$, $H_A^t$ are **projection matrices onto random low-rank subspaces** induced by the current LoRA factors and sketches. As derived explicitly in Appendix E, in the left-sketch branch we have
> > $$
> > H_B^t := B_S^t \bigl((B_S^t)^\top B_S^t\bigr)^\dagger (B_S^t)^\top,
> > $$
> > and in the right-sketch branch
> > $$
> > H_A^t := (A_S^t)^\top \bigl(A_S^t (A_S^t)^\top\bigr)^\dagger A_S^t,
> > $$
> > where $A_S^t$ and $B_S^t$ are the sketched LoRA factors at iteration $t$ and $^\dagger$ denotes the Moore–Penrose pseudoinverse. Thus $H_B^t$ and $H_A^t$ encode, in matrix form, the effect of “update $B$” vs “update $A$” as projections of the full gradient onto the corresponding low-rank subspaces.
> >
> > Different choices of
> >
> > - Bernoulli probability $p$,
> > - left/right sketch distributions, and
> > - base gradient estimators (GD / SGD / PAGE / MVR, etc.)
> >
> > recover or closely approximate:
> >
> > - RAC-style left-only/right-only chains,
> > - COLA-like sequential low-rank updates,
> > - standard LoRA-type updates with factorized low-rank modules, and
> > - their federated / communication-efficient variants (via the FL extensions).
> >
> > **Planned changes in the paper.**
> > To substantiate this more concretely, we will:
> > - Replace overly broad phrases such as “unifies almost all PEFT methods” by a more precise claim:
> >   > “Our framework unifies a family of LoRA-style low-rank gradient updates, including RAC-LoRA-type asymmetric schemes and their stochastic and federated variants, under a single projection-based analysis.”
> >
> > This should make the scope of “unification” explicit and technically justified.

---

> > > ### Author Response · Authors · 2025-11-21
> > > **Rebuttal to Reviewer 8Tnv (Part 3)**
> > >
> > > ### 3. Limited experimental evaluation and lack of standard benchmarks
> > >
> > > You pointed out that the experimental evaluation is limited in scope (linear regression and MNIST) and suggested extending experiments to more comprehensive benchmarks closer to typical LoRA usage (e.g., NLU or vision tasks).
> > >
> > > **Our intended role of experiments.**
> > > The paper is primarily **theoretical** in nature. The experiments are deliberately small-scale and controlled, serving as *sanity checks* that:
> > > - the proposed methods behave in line with the theoretical predictions, and
> > > - Bernoulli-LoRA is not unstable or degenerate in practice.
> > >
> > > This design follows common practice in optimization papers whose main emphasis is on theory.
> > >
> > > We fully agree, however, that from a PEFT/LoRA practitioner’s perspective, **larger-scale experiments would significantly strengthen the paper**.
> > >
> > > **Planned additions.**
> > >
> > > Within the remaining time window, we are working on:
> > > - **Adding at least one more realistic fine-tuning setup**, for example:
> > >   - a transformer-based model (e.g., RoBERTa-style) on a text classification / NLU benchmark, or
> > >   - a moderate-scale vision backbone on a standard image dataset.
> > > - **Including federated experiments** where the Bernoulli-LoRA-FL variants are actually exercised in a heterogeneous client setting.
> > >
> > > In the rebuttal, we will report any new results that are ready, and in the camera-ready version (if accepted) we will integrate the complete expanded experimental section. We will also explicitly state that our main contribution is **the unified theoretical framework**, with experiments serving as illustrative validation rather than an exhaustive empirical study.
> > >
> > > ---
> > >
> > > Once again, we thank you for your detailed comments and for indicating that you are willing to adjust your evaluation if these concerns are addressed. We hope the above clarifications, together with the planned revisions and additional experiments, will help resolve your doubts about the contribution of the paper.

---

### Official Review · Reviewer_TnTK · 2025-11-03

**Soundness:** 3
**Presentation:** 2
**Contribution:** 2
**Rating:** 4
**Confidence:** 4

**Summary:**

This paper analyzes the convergence of LoRA fine-tuning in the context of non-convex optimization.

**Strengths:**

- The theoretical results are reasonable because they are extensions of the standard results in convex and non-convex optimization.
- The study and analysis about PEFT model fine-tuning techniques are necessary.

**Weaknesses:**

- The setting of the paper is a simplification of the practical use of LoRA. More specifically, it considers $f(W^0 + \Delta W)$ where LoRA is applied to only one matrix. However, in practice, LoRA is applied to many matrices (e.g., key, query, value matrices of self-attention layers) across many layers of a transformer.
- The setting of the optimization problem in this paper is unclear. Particularly, it is unclear what parameters are optimized in the target optimization problem.
  - If $\Delta W$ in (1) is optimized, it refers to full fine-tuning. The objective is to prove that the LoRA updates can converge efficiently to the full fine-tuning. However, it is weird that in experimental results in Table 3, standard LoRA attains only 86% of full-parameter fine-tuning.
- The theoretical results are not surprising because they are too standard in the field.
- It would be better if the experiments are conducted with foundation models (ViT, LLMs, SDMs) because this is a common use of PEFT.

**Questions:**

- What is the matrix $H$ in Eq. (8)?
- Can you suggest more insights or understanding from your theories regarding how to use LoRA more efficiently in real-world applications?

---

> ### Author Response · Authors · 2025-11-19
> **Rebuttal to Reviewer TnTK (Part 1)**
>
> Dear Reviewer TnTK,
>
> Thank you for your careful evaluation of our work. We appreciate that you:
>
> - Recognized that the **theoretical results are reasonable** as extensions of standard convex/non-convex optimization tools.
> - Emphasized that **the study and analysis of PEFT fine-tuning techniques are necessary**, highlighting the value of theoretical work in this area.
>
> We now address your concerns in detail.
>
> ---
> ### 1. Simplified setting: LoRA applied to a single matrix
>
> You noted that our setting considers LoRA applied to a single matrix $W^0$, whereas in practical LLM fine-tuning LoRA is typically applied to many matrices (e.g., key/query/value and feed-forward projections) across many layers, and that the target optimization problem is not fully clear.
>
> **Response and clarification.**
>
> Our formulation
> $$
> \min_{\Delta W} f(W^0 + \Delta W)
> $$
> follows the standard theoretical abstraction used in the LoRA literature. In particular, the original LoRA paper of Hu et al. (2021), LoRA in FL by Sun et al. (2024), the theoretical RAC-LoRA framework of Malinovsky et al. (2024), the COLA method of Xia et al. (2024), and AsymmLoRA by Zhu et al. (2024) all write the fine-tuning objective in terms of a *single* weight matrix with a low-rank update, even though in experiments LoRA modules are applied to many layers and matrices. In these works, $W^0$ is understood as “the parameter block where LoRA is active”, not necessarily a literal single layer.
>
> Our analysis adopts exactly the same abstraction. Technically, when LoRA is applied to multiple matrices $\{W^{(\ell)}\}_{\ell=1}^L$, each with low-rank update $B^{(\ell)}A^{(\ell)}$, one can stack or block-diagonalize them into a single object:
> - either by concatenating their vectorizations into a single parameter vector, or
> - by forming a block-diagonal matrix
>   $$
>   W_0 = \mathrm{diag}\big(W^{(1)},\dots,W^{(L)}\big), \quad
>   \Delta W = \mathrm{diag}\big(B^{(1)}A^{(1)},\dots,B^{(L)}A^{(L)}\big),
>   $$
> and defining $f$ on this aggregated parameter.
>
> Because we work with the Frobenius norm and Frobenius inner product, all our assumptions and proofs (Lipschitz smoothness, PŁ condition, expected projection assumptions, and the gradient-norm bounds) extend verbatim to this block/tensor setting: the gradient is just the concatenation of block gradients, the projection matrices become block-diagonal, and the spectral quantities that appear in the rates are determined by the blocks. In other words, our theory already covers the “many-matrix” case once we interpret $W^0$ as the concatenation of all LoRA-modified matrices.
>
> We chose to present the theory in the single-matrix notation to keep the exposition and proofs readable; this is the same simplification adopted in the theoretical treatments of RAC-LoRA and COLA, as well as in other LoRA analyses mentioned above.
>
> **Planned changes in the revised version.**
>
> In the revised manuscript we will:
> - Explicitly state in the problem-formulation section that $W^0$ can represent either a single layer or the concatenation/block-diagonal stacking of all matrices where LoRA is applied, following the abstraction used in Hu et al. (2021), Sun et al. (2024), Malinovsky et al. (2024), Xia et al. (2024), and Zhu et al. (2024).
> - Add a short remark (or appendix note) showing how the multi-matrix case is obtained via concatenation under the Frobenius norm, and why the convergence theorems and spectral assumptions carry over unchanged.
>
> We hope this clarifies that our optimization setting is fully compatible with the practical use of LoRA across many layers and matrices, and that the single-matrix formulation is a standard theoretical abstraction rather than a limitation of the method.
>
> #### References:
> Hu, Edward J., et al. “LoRA: Low-Rank Adaptation of Large Language Models.” International Conference on Learning Representations (ICLR), 2022.
>
> Sun, Youbang, et al. “Improving LoRA in Privacy-Preserving Federated Learning.” International Conference on Learning Representations (ICLR), 2024.
>
> Malinovsky, Grigory, et al. “Randomized Asymmetric Chain of LoRA: The First Meaningful Theoretical Framework for Low-Rank Adaptation.” arXiv, preprint arXiv:2410.08305, 2024.
>
> Xia, Wenhan, Chengwei Qin, and Elad Hazan. “Chain of LoRA: Efficient Fine-Tuning of Language Models via Residual Learning.” arXiv, preprint arXiv:2401.04151, 2024.
>
> Zhu, Jiacheng, et al. “Asymmetry in Low-Rank Adapters of Foundation Models.” Proceedings of the 41st International Conference on Machine Learning

---

> ### Author Response · Authors · 2025-11-19
> **Rebuttal to Reviewer TnTK (Part 2)**
>
> ### 2. What is actually being optimized? Clarifying the optimization problem
>
> You noted that the optimization problem in the paper is not clearly specified and raised the concern that, if $W$ itself is optimized, this would correspond to full fine-tuning, which contradicts the LoRA restriction. You also found it odd that standard LoRA achieves only $\sim 86\%$ of full fine-tuning in our experiments.
>
> **Clarification.**
>
> Our starting point is the general fine-tuning objective
> $$
> \min_{\Delta W \in \mathbb{R}^{m \times n}} f(W^0 + \Delta W),
> $$
> where $W^0$ denotes the fixed pre-trained model and $\Delta W$ is the trainable adaptation.
>
> Different LoRA-style methods correspond to different structured parameterizations of $\Delta W$:
>
> - In the original LoRA formulation, one optimizes low-rank factors $A$ and $B$ so that
>   $$
>   \Delta W = \frac{\alpha}{r} BA,
>   $$
>   which yields the adapted model
>   $$
>   W = W^0 + \frac{\alpha}{r} BA.
>   $$
>
> - In COLA and RAC-LoRA, the update is built as a sum of low-rank components,
>   $$
>   \Delta W = \frac{\alpha}{r} \sum_{t=0}^{T-1} B^t A^t,
>   $$
>   giving
>   $$
>   W = W^0 + \frac{\alpha}{r} \sum_{t=0}^{T-1} B^t A^t.
>   $$
>
> Our framework follows this latter, chain-style formalism. Algorithm~1 specifies how the factors $A^t$ and $B^t$ are constructed at each step, and the resulting model is
> $$
> W = W^0 + \frac{\alpha}{r} \sum_{t=0}^{T-1} B^t A^t.
> $$
>
> Thus, throughout the paper, the optimization variable is the structured low-rank adaptation $\Delta W$ (through the factors $\{A^t, B^t\}_{t=0}^{T-1}$), and our convergence guarantees are for this LoRA-parameterized problem, not for unrestricted full fine-tuning.
>
> **On the 86% accuracy vs full fine-tuning.**
>
> Given the low rank of the LoRA adaptation and the limited capacity of the chosen architecture, it is not surprising that LoRA **does not fully match** full-parameter fine-tuning. LoRA restricts updates to a low-dimensional subspace; when that subspace is too small to capture the optimal full-rank solution, a gap in accuracy is expected. Our goal is to show that LoRA-style methods converge **within this subspace**, not that they always recover full-FT performance.
>
> We will clarify this interpretation in the experimental section.

---

> > ### Author Response · Authors · 2025-11-19
> > **Rebuttal to Reviewer TnTK (Part 3)**
> >
> > ### 3. “Standard” nature of theoretical results
> >
> > You commented that the theoretical results are not surprising and are largely standard.
> >
> > **Response.**
> >
> > We respectfully but firmly disagree with the implication that “standard-looking” convergence rates make the contribution marginal.
> >
> > First, it is *unavoidable* that our bounds resemble classical non-convex rates in form: we work under standard smoothness / PŁ-type assumptions and use first-order information, so one cannot hope for exotic asymptotic orders. Where our work is non-trivial is **not** in inventing a new inequality, but in **showing that these classical guarantees actually hold for a highly structured and biased LoRA-type update**:
> > - with a **low-rank parameterization** of the update,
> > - with **random sketching / Bernoulli selection** and projection operators $H^t$,
> > - and, in several variants, in **federated settings with compression and error feedback**.
> >
> > Before our work, it was *not* at all clear that one can recover clean, GD-style convergence guarantees in this regime without adding restrictive assumptions. In particular, even for RAC-style LoRA-type methods, there was no unified analysis covering GD/SGD/variance reduction/FL variants within a single framework.
> >
> > Second, we would like to stress that in modern optimization it is very common — and often technically demanding — to obtain **new results using “known” tools**. Almost all influential convergence analyses build on smoothness, descent lemmas, Lyapunov functions, etc.; what matters is *where* and *how* these tools are applied. In our case:
> > - we design a Bernoulli-LoRA update that is compatible with a Lyapunov-style analysis despite the **bias and randomness introduced by low-rank adapters**,
> > - we track the effect of the Bernoulli mechanism through the spectral properties of the expected projections,
> > - and we extend this to **multiple algorithmic regimes (GD/SGD/PAGE/FL variants) within the same conceptual framework**.
> >
> > To the best of our knowledge, this is the **first work** that provides rigorous convergence guarantees for a family of Bernoulli / RAC-style LoRA methods with stochastic, variance-reduced, and federated variants. We will make this positioning more explicit in the revised version and will carefully downplay any wording that could be read as claiming novel proof *techniques* rather than novel **applications and extensions** of established techniques to a non-trivial LoRA setting.
> >
> >
> > ---

---

> ### Author Response · Authors · 2025-11-19
> **Rebuttal to Reviewer TnTK (Part 4)**
>
> ### 4. Lack of experiments on foundation models (ViT, LLMs, SDMs)
>
> You suggested that experiments on foundation models would be more convincing given how PEFT is used in practice.
>
> **Response.**
>
> We would like to stress that the primary contribution of this work is **theoretical**, not empirical benchmarking. Our goal is to provide a rigorous convergence framework for Bernoulli/RAC-style LoRA methods (including stochastic, variance-reduced, and FL variants), in the same spirit as many optimization papers where experiments serve to **illustrate** the theory rather than to compete as a large-scale systems paper.
>
> For this reason, we deliberately chose small-scale, controlled tasks where we can closely track the behavior predicted by our assumptions and bounds. This setting is standard for theoretical optimization work and is the most appropriate environment to validate the convergence phenomena we analyze.
>
> That said, we fully agree that showing at least one realistic PEFT scenario would strengthen the paper from a practitioner’s perspective. Within the available resources we are working towards:
> - including **at least one more realistic PEFT experiment**, e.g., fine-tuning a transformer-based model on a standard NLP or vision benchmark, and
> - if feasible in the timeline, exploring a **federated** scenario with a moderately sized model.
>
> In the revised manuscript, we will (i) explicitly state that the paper should be read first and foremost as a **theoretical contribution**, and (ii) clearly acknowledge the absence of foundation-model experiments as a limitation of the current empirical section, while indicating the additional experiments we are adding as complementary evidence rather than the main contribution.

---

> ### Author Response · Authors · 2025-11-19
> **Rebuttal to Reviewer TnTK (Part 5)**
>
> ### 5. Question: What is the matrix $H$ in Eq. (8)?
>
> You asked for a clearer definition of the matrix $H$ in Eq. (8).
>
> **Clarification.**
>
> In Section E of the appendix, we show that the Bernoulli-LoRA update can be written in the **projected-gradient form**
> $$
> W_{t+1} = W_t - \gamma H^t G_t,
> $$
> where $G_t = \nabla f(W_t)$ (or its stochastic/VR analogue), and $H^t$ is a **symmetric positive semidefinite matrix** that encodes the effect of the LoRA parameterization, the sketch matrices, and the Bernoulli choice at iteration $t$.
>
> More precisely:
>
> - At each iteration $t$, Algorithm 1 flips a Bernoulli variable to decide whether to update $B$ or $A$.
> - In the **Left Sketch / update-$B$** case, Appendix Section E shows that the update can be written as
>   $$
>   W_{t+1} = W_t - \gamma H_B^t \nabla f(W_t),
>   $$
>   with the projection matrix
>   $$
>   H_B^t := B_S^t\Bigl((B_S^t)^\top B_S^t\Bigr)^\dagger (B_S^t)^\top,
>   $$
>   where $B_S^t$ is the sketched version of $B^t$ and $^\dagger$ denotes the Moore–Penrose pseudoinverse.
> - In the **Right Sketch / update-$A$** case, a symmetric derivation yields
>   $$
>   W_{t+1} = W_t - \gamma \nabla f(W_t) H_A^t,
>   $$
>   with
>   $$
>   H_A^t := (A_S^t)^\top \Bigl(A_S^t (A_S^t)^\top\Bigr)^\dagger A_S^t,
>   $$
>   where $A_S^t$ is the sketched version of $A^t$.
>
> We then define the random projection operator
> $$
> H^t = H_B^t,  \text{if the Bernoulli choice at iteration } t \text{ selects } B
> $$
> or
> $$
> H^t = H_A^t,  \text{if it selects } A.
> $$
>
> Intuitively, $H_B^t$ and $H_A^t$ are **projection operators onto random low-rank subspaces** determined by the current LoRA factors and the sketch matrices at step $t$. They map the full gradient direction $\nabla f(W_t)$ to the actually realizable update direction in $W$ when we update only $B$ or only $A$. In expectation, these operators give rise to the matrices $\mathbb{E}[H_B]$ and $\mathbb{E}[H_A]$ whose spectra appear in our convergence bounds.
>
> In the revised version, we will:
> - Insert the explicit formulas for $H_B^t$ and $H_A^t$ immediately after Eq. (8), with a pointer to Appendix Section 5.1 where they are derived, and
> - Add a short intuitive sentence along the lines of:
>   > “Here $H^t$ is the symmetric positive semidefinite matrix that projects the full gradient $G_t$ onto the low-rank subspace induced by the current LoRA factors and sketch matrices at iteration $t$ (see Appendix Section 5.1 for the explicit construction).”
>
> We hope this clarifies both the formal and intuitive role of $H$ in Eq. (8).

---

> > ### Author Response · Authors · 2025-11-19
> > **Rebuttal to Reviewer TnTK (Part 6)**
> >
> > ### 6. Question: Practical insights for using LoRA more efficiently
> >
> > You asked whether our theory provides insights on how to use LoRA more efficiently in real-world applications.
> >
> > **Insights from our analysis.**
> >
> > While our paper is primarily theoretical, it does suggest several **practical guidelines**, which we will make explicit in a short “Practical Guidelines” paragraph:
> >
> > - **Updating one side vs both.**
> >   Our bounds depend on the spectrum of the projection operators (through quantities such as $\lambda_{\min}^p$). This indicates that systematically updating the “better-conditioned” side (as in RAC-style schemes or extremal $p$) can be more stable than blindly updating both sides, especially when one of the two factors induces a stronger projection of the gradient.
> >
> > - **Choice of rank: accuracy–cost trade-off.**
> >   In the Gaussian case analyzed in Appendix D (Lemma 2), we show that
> >   $$
> >   \mathbb{E}[H_B] = \mathbb{E}[H_A] = \frac{r}{n}I_n
> >   \quad\Rightarrow\quad
> >   \lambda_{\min}^p = \frac{r}{n},
> >   $$
> >   so the convergence rate in Theorem 1,
> >   $$
> >   \mathbb{E}\!\left[\|\nabla f(\widetilde W^T)\|_F^2\right]
> >   \le \frac{2\Delta^0}{\gamma\,\lambda_{\min}^p T},
> >   $$
> >   improves linearly with the rank $r$ (for fixed $n,\gamma,T$). Thus, **larger rank accelerates convergence in our theory**. At the same time, a higher rank increases the number of trainable parameters and the per-step computational and memory cost. Our analysis therefore highlights an explicit *accuracy–cost trade-off* in choosing the LoRA rank, rather than claiming that “smaller rank is always better.”
> >
> > - **Stepsizes and spectral constants.**
> >   The stepsize constraints in Theorems 1–2 depend on Lipschitz and spectral constants (e.g., via $\lambda_{\max}^p$). This supports the standard practice of tuning learning rates carefully and suggests that overly aggressive stepsizes can violate the theoretical conditions and hurt both convergence and stability.
> >
> > - **Federated setting and communication efficiency.**
> >   The FL variants show that combining LoRA with compression and error feedback (QGD/MARINA/EF21-style mechanisms) still admits provable convergence. This provides a principled way to design communication-efficient PEFT methods in federated scenarios: one can safely use LoRA adapters together with standard compressors and error-feedback schemes, knowing that the resulting method fits within our convergence framework.
> >
> > We will highlight these points more clearly in the revised version so that practitioners can see how the theoretical analysis informs design choices such as which side to update, how to choose rank, and how to combine LoRA with FL and compression.
> >
> > ---
> >
> > Thank you again for your constructive feedback. We believe that the clarifications above and the planned revisions will address your concerns about the setting, optimization problem, and practical implications of our theoretical results.

---

### Author Response · Authors · 2025-12-04

Dear AC, SAC, PC, and Reviewers,

We would like to thank you again for the time and effort you invested in reviewing our submission. Following the reviews and subsequent discussion, we have updated the manuscript and addressed most of the raised concerns. Below we briefly summarize the main changes and indicate which reviewer comments they respond to.

- **Reorganized front sections (structure and flow) (requested by Reviewer TCPp)**
  - We reorganized Sections 1–3 to provide a clearer narrative: the introduction now presents the LoRA/PEFT context and main contributions, followed by a concise “Problem Setup and Notation” section that defines $W_0$, $\Delta W$, $f(W_0 + \Delta W)$, and the finite-sum / expectation / FL settings. The Motivation section is integrated more smoothly with this setup.
  - We added forward references from the problem setup to Table~1 and the main convergence theorems so that readers can more easily see how the formalism connects to our results.
  - We positioned our work explicitly as complementary to recent LoRA-dynamics analyses (NTK, gradient flow, lazy vs feature learning) and integrated the theoretical LoRA papers suggested by Reviewer TCPp, explaining how our discrete-time convergence guarantees differ from and complement these works.
  - We clarified that our analysis is **conditional on a fixed pre-trained model** $W_0$ and that $f$ encodes the fine-tuning loss, including any effects of the pre-training vs fine-tuning data.
  - We now state explicitly that questions of **representation learning and generalization** (beyond optimization guarantees) are **out of scope** for this work.

- **Clarified assumptions and their realism (requested by Reviewers TCPp, CiyU)**
  - We strengthened the discussion around our assumptions, emphasizing that Lipschitz smoothness, the PŁ condition, and the positive expected projection property are **standard modeling abstractions** in non-convex optimization and federated learning, rather than exact descriptions of all deep networks.
  - We explicitly highlight that all of our convergence guarantees are **conditional on these assumptions**, in the same spirit as classical GD/SGD/FL analyses, and we provide additional pointers to the relevant optimization literature.

- **Clarified the optimization variable and LoRA restriction (requested by Reviewer TnTK)**
  - In the Problem Statement, we now clearly state that the optimization variable is the **low-rank update** $\Delta W$, with a LoRA-style parameterization (e.g., $\Delta W = \tfrac{\alpha}{r} B A$ or its chain-of-LoRA extension), and that **only the low-rank factors** are trainable. We also emphasize that our convergence guarantees apply to this **LoRA-parameterized** problem, not to unrestricted full fine-tuning.

- **Explained the multi-matrix LoRA setting via concatenation / block-diagonal form (requested by Reviewer TnTK)**
  - We added a remark, together with an appendix note, explaining how multiple LoRA-modified matrices across layers can be represented as a single concatenated or block-diagonal $W_0$ and $\Delta W$ under the Frobenius norm. We show that, under this standard abstraction, all assumptions and theorems extend verbatim because gradients and projection matrices decompose blockwise.
  - We explicitly reference prior work (Hu et al., 2021; Sun et al., 2024; Malinovsky et al., 2024; Xia et al., 2024; Zhu et al., 2024) that uses the same single-matrix abstraction for theoretical analysis while applying LoRA to many matrices in practice.

- **Improved presentation of algorithms and rates**
  - We merged the previous two tables (algorithm descriptions and convergence-rate summary) into a **single unified table**, which lists, for each method: the setting, the update rule/base gradient estimator $G^t$, and the non-convex and PŁ convergence rates with links to the corresponding theorems. This makes the overall theoretical picture more transparent and easier to navigate.

We hope these revisions meaningfully improve the clarity and positioning of the paper and address the main concerns raised during the review process. We remain grateful for the constructive feedback and are happy to further clarify any of these changes if needed.

Sincerely,
The authors

---

### Meta-Review · Area_Chair_zttX · 2025-12-22

**Summary:**

The reviewers had the following concerns:
1. Some writing lacks clarity. Notably the organization of the first several sections (introduction, motivation, contributions, etc.), the problem settings, particularly regarding the training objective and trainable parameters, and some technical definitions (for example, the projection matrix H).
2. Validity of the assumptions (Positive expected projection, Lipschitz smoothness) are hard to verify.
3. Lack of large-scale experiments.
4. Insufficient discussion around the theoretical results. Notably, its novelty over existing algorithms is unclear, the claim that the proposed algorithm encapsulates many existing ones is not justified, and the practical implications are not discussed.

**Reviewer Concerns:**

I think the first two concerns were addressed. The authors have clarified reviewers' questions regarding writing clarity. For the concern on assumptions, since they are more or less standard in optimization literature, I believe the reviewer would not insist on viewing the assumptions as a major weakness.

The other two are still outstanding in my view, after reading the manuscript myself. First of all, the authors only proposed a plan for adding experiments, so the concern on insufficient empirical validation remains. For the last point, the author rebuttal provided some reasonable responses/explanations to the reviewers' concern, by which I think some reviewers might be convinced. However, most points made in the rebuttal either were not clear from the original manuscript, or did not appear in the paper. Revising the manuscript to incorporate those points in my opinion is equivalent to rewriting the paper.

To elaborate my last point, based on my own reading, the paper after Page 4 is a dry presentation of the framework, followed by plain statements of the assumptions and theorems, then some simple experiments. Many critical parts are missing (those are all related to concern 4): the comparison between the proposed framework and prior ones; the interpretations of the convergence rate bounds, i.e. their dependence on the constants defined by the problem; the practical implications of the analysis (without which I could not follow the purpose of the empirical experiments). Therefore, concern 4 cannot be addressed without a major revision of the manuscript.

**Reviewer Scores:**

The reviewers TnTK (initial rating 4) and TCPp (initial rating 2) might have adjusted their score based on the author rebuttal, because the majority of their concerns pertain to the writing clarity. However, the adjusted rating would have not across borderline since concern 3 and 4 remains.

The reviewer 8Tnv (initial rating 2) is unlikely to adjust the score, because their primary concerns are 3 and 4.

The reviewer CiyU (initial rating 6) has expressed the tendency to keep the score.

---

### Decision · Program_Chairs · 2026-01-26

Reject